# Temporal Geometry of Deep Networks: Hyperbolic Representations of Training Dynamics for Intrinsic Explainability

**Ambarish Moharil**

Data Governance Department

Jheronimus Academy of Data Science

Eindhoven University of Technology

`a.moharil@tue.nl`

## Abstract

Intrinsic explainability remains a challenging problem, particularly in contexts where multilayer perceptrons (MLPs) require dynamic re-training within an optimization environment. This paper investigates how MLPs and their training dynamics can be represented and studied in non-Euclidean spaces; our representation features the Poincaré model of hyperbolic geometry. We aim to capture the geometric evolution of their weighted topology and self-organization over time. Instead of restricting the analysis to single checkpoints—as per established measure-based explainability methods—we construct temporal *parameter graphs*, i.e., snapshots over time $T$ steps of the optimization/training process for MLPs. This reflects the view that neural networks encode information not only in their weights but also in the trajectory traced during training. Drawing on the idea that many complex networks admit embeddings in hidden metric spaces where distances correspond to connection likelihood, we present a geometric and temporal graph-based metalearning framework for obtaining dynamic hyperbolic representations of the underlying neural parameter graphs. Our model embeds temporal parameter graphs in the Poincaré model ball, and learns from them while maintaining equivariance to within-snapshot neuron permutations and invariance to permutations of past snapshots. In doing so, the approach preserves functional equivalence over time and recovers the latent evolving geometry of the network. Experiments on regression and classification tasks with trained MLPs show strong meta-network performance, accompanied by hyperbolic temporal representations. This reveals how the network structure emerges over time under specific training environments, thus providing insights into the network's self-organization.

## 1 Introduction

The central problem that we address in this work is the geometric evolution of neural networks, specifically how the weighted topology of a deep neural network (DNN) model reorganizes during training and how this self-organization shapes the (neural network) function. We argue that studying the weighted topology is key to figure out the role of neural weights, that is the parameters used for any decision, are not just incidental outputs of some optimization process, but also the *structures* through which generalization, compression, and transfer learning arise (Han et al., 2016; Li et al., 2020). Our focus in this work is not only to investigate the performance of hyperbolic and temporal meta-networks, but also to expose how a neural network's structure forms and shifts during training. A temporal geometric view makes it possible to track the emergence of hierarchy, the consolidation of functional modules, and the redistribution of influence across layers by studying the evolution of its parameter graph. This perspective offers intrinsic explanations grounded in the model's own dynamics and self-organization, rather than relying on post-hoc probes for explanation.

A recent approach has been to treat neural networks as data objects *themselves* and to train *meta-networks* that take neural parameters as inputs to predict the generalization performance of the un-

derlying model (accuracy, $R^2$, etc.), classifying the implicit neural representations (INRs) of the underlying neural model or to generate weights for transfer learning across domains (Unterthiner et al., 2021; Schürholt et al., 2021; Dupont et al., 2022; Navon et al., 2024; Zhou et al., 2023b; You et al., 2020; Lim et al., 2023; Kofinas et al., 2024). Early works on the topic flattened weights and biases into a single tensor, but this discarded structural relations and ignored the underlying geometric priors of the weight space, such as functional invariance and equivariance to permutations (Hecht-Nielsen, 1990; Kofinas et al., 2024). Later work enforced data symmetries, such as permutation equivariance, ensuring that reordering the neurons did not affect the network functionality (Zhou et al., 2023a;b; Navon et al., 2024; Hecht-Nielsen, 1990). These methods, although robust, often require architecture-specific tuning and Euclidean representations, making it challenging to interpret the network's internal self-organization.

A complementary direction has leveraged graph-theoretic and network-science perspectives. Here, neurons and biases become network nodes, whereas the model weights are tied to the existence of an edge link between two nodes, with meta-learning being performed on these *parameter graphs* (You et al., 2020; Lim et al., 2023; Kofinas et al., 2024). These methods have performed well for INR classification and generalization prediction tasks, but usually operate on single checkpoints (along the training trajectory) and make zero-shot predictions, leaving the temporal traces of optimization underused (Lim et al., 2023; Kofinas et al., 2024). On the other hand, few studies on complex network graphs in physical domains (Simas & Rocha, 2014; Allard et al., 2017) have considered how geometric priors could inform edge weight dynamics; a link that can be crucial in making deep learning interpretable.

Network science points to two priors of particular relevance: small-world effects and modularity (Song et al., 2005; Kim et al., 2007b; Wei et al., 2013; Fronczak et al., 2024). They also often admit embeddings in metric spaces where the distances reflect the connection probabilities (Allard et al., 2017; Krioukov et al., 2010). If parameter graphs display similar tendencies, embedding them in metric spaces with distance-biased learning can yield compact, interpretable representations. Most existing methods use Euclidean embeddings (Lim et al., 2023; Kofinas et al., 2024), but these require high dimensions and distort structure when projected (Nickel et al., 2014; Nickel & Kiela, 2017). On the other hand, hyperbolic spaces preserve hierarchical relations with low distortion (Gromov, 1987), yet no prior work has embedded neural parameter graphs within hyperbolic spaces while preserving permutation symmetry. The closest example addresses static graphs of multilayer brain networks (Guillemaud et al., 2025).

We close this gap by developing a geometric temporal hyperbolic framework for parameter graphs that treats training itself as an object of study. Conceptually, we view the optimization of a neural network as a trajectory across weighted parameter graphs evolving in a negatively curved space, and we argue that intrinsic explanations should be grounded in this evolution rather than in isolated checkpoints. Methodologically, we (a) construct a temporal hyperbolic meta network (GTH–GMN) that (b) models MLP parameter traces as permutation equivariant parameter graphs, (c) embedding them in the Poincaré ball, and (d) coupling distance-biased hyperbolic attention with meta evolution of the attention kernels over time (Pareja et al., 2019; Li et al., 2024). The architecture is designed to respect the symmetries of weight space: it is equivariant to the neuron permutations within each snapshot and invariant to permutations of past snapshots, so that functionally equivalent networks share the same representation, while their latent geometry can still be recovered. A signed weight regressor links geodesic distance to edge magnitude through a power law prior (Allard et al., 2017), decoupling magnitude from polarity and importing scale free inductive biases from network science (Song et al., 2005; Kim et al., 2007b; Wei et al., 2013; Fronczak et al., 2024). Empirically, we show on INR classification, CIFAR–10 generalization prediction, and sinusoid regression, that this temporal hyperbolic encoder matches or heavily approaches strong Euclidean and tensor-based baselines, while producing compact embeddings whose radial and angular structure tracks the self-organization of the underlying networks. In this way, the method offers a concrete route toward intrinsic explainability through the geometry of parameter graph trajectories (Unterthiner et al., 2021; Schürholt et al., 2021; Dupont et al., 2022; Navon et al., 2024; Zhou et al., 2023b; You et al., 2020; Lim et al., 2023; Kofinas et al., 2024; Allard et al., 2017; Krioukov et al., 2010; Nickel et al., 2014; Nickel & Kiela, 2017; Gromov, 1987; Guillemaud et al., 2025; Pareja et al., 2019; Li et al., 2024).

## 2 RELATED WORK

**Neural Meta Networks.** Neural networks can themselves be treated as data. Early meta-network approaches flattened parameters or relied on simple statistics, which ignored neuron permutation symmetry and had limited cross-architecture generalization (Unterthiner et al., 2021; Eilertsen et al., 2020; Dupont et al., 2022; De Luigi et al., 2023). Subsequent works developed permutation-equivariant meta-networks that respect weight-space symmetries (Schürholt et al., 2021; Navon et al., 2024; Zhou et al., 2023a;b; Maron et al., 2019). Recently, Graph-based approaches have extended this view by modeling parameters as graphs (Lim et al., 2023; Kofinas et al., 2024). Graph Meta Networks (GMN) construct parameter graphs to learn graph neural networks (GNNs) that remain equivariant to neural DAG automorphisms, covering CNNs, Transformers, and multilayer perceptrons (MLPs) (Lim et al., 2023). These graph-based approaches model individual neurons and biases as *parameter graph* nodes, where edge existence depends on the weight between adjacent nodes, enabling GNNs or relational transformers to process architectures with varying depth, width, and skip connections (Lim et al., 2023; Kofinas et al., 2024). Other work has used higher-level graphs for NAS, federated learning, and parameter sharing (Liu et al., 2018; You et al., 2020; Litany et al., 2022), though often at a coarser level of abstraction. GNN provides a foundation of permutation-equivariant operators for nodes and edges (Hamilton, 2020; Maron et al., 2019; Kim et al., 2021; Lim et al., 2023). Our contribution builds on this trajectory but addresses two gaps. First, most approaches rely on single checkpoints and do not exploit the temporal traces generated during optimization. Second, embeddings are almost always Euclidean, which limits their capacity to represent hierarchical or heavy-tailed structures. We therefore focus on *temporal parameter graphs* of MLPs, embedding them in hyperbolic space to capture both the evolving geometry and functional symmetries over the course of training.

**Hyperbolic GNNs.** Poincaré and Lorentz embeddings first established that hierarchical relations admit compact, faithful representations in negatively curved spaces (Nickel & Kiela, 2017; 2018) ( For background on hyperbolic geometry, refer to Appendix B). Building on this insight, hyperbolic neural models extended message passing, attention, and diffusion to non-Euclidean manifolds for static graphs (Ganea et al., 2018; Chami et al., 2019; Zhang et al., 2019; Wen et al., 2024). The key challenge is that hyperbolic spaces are not vector spaces, so Euclidean modules do not apply verbatim (Yang et al., 2023). Two strategies have emerged: design fully manifold counterparts for core layers (Chen et al., 2021), or lift computations to tangent spaces and map back while enforcing model-specific constraints (Ganea et al., 2018; Chami et al., 2019; Zhang et al., 2021). Manifold optimization further supports these pipelines through Riemannian adaptive methods (Sakai & Iiduka, 2020). In our framework, node embeddings are also updated intrinsically: alongside the Euclidean parameter pass for kernels and regressors, we run a dedicated Riemannian optimizer that updates positions $z^{(t)}$ on the Poincaré ball in a geometry-consistent second phase.

**Temporal Learning in Euclidean and Hyperbolic Spaces.** Early approaches to temporal graph learning largely relied on Euclidean neural architectures. Sequence-aware graph models combined recurrent updates with graph convolutions. For example, the spatial operators in convolution-based long-short term memory (LSTM) networks were replaced with Chebyshev filters so that temporal dynamics and topology could be processed jointly (Seo et al., 2016; Shi et al., 2015; Defferrard et al., 2017). Later work introduced uncertainty modeling over time by placing latent distributions at each step (Hajiramezanali et al., 2019), and eventually allowed the graph encoder itself to evolve through recurrent updates, so the parameters of the convolutional layers changed along with the temporal signal (Pareja et al., 2019). However, Euclidean geometry can flatten or obscure the heavy-tailed and hierarchical patterns that naturally arise in many temporal networks (Krioukov et al., 2010; Nickel & Kiela, 2017). This motivated the development of temporal models built directly in hyperbolic space, where message passing and recurrence are designed to respect the curvature. These methods combine hyperbolic graph layers, hyperbolic recurrent units, and temporal attention mechanisms, along with geometric consistency losses that stabilize learning (Yang et al., 2023). Other approaches use hyperbolic diffusion operators, dilated causal convolutions, and curved decoders adapted to long-horizon forecasting (Bai et al., 2023). Hyperbolic attention variants compute similarity using geodesic distances and perform aggregation using Einstein gyromidpoints. Extensions that incorporate temporal context improve multi-step forecasting and link prediction (Li et al., 2024; 2023). Continuous time formulations push this further by combining hyperbolic variational encoders with temporal encodings for event stream modeling (Sun et al., 2021a). *Across these developments, the unifying ideas are operators that preserve the manifold, mappings that respect curvature, and explicit mechanisms that maintain temporal coherence. Despite this progress, difficulties remain in*

*scaling hyperbolic temporal models and in adapting curvature over long sequences* (Zhang et al., 2021; Yang et al., 2023). To address these limitations, we follow the kernel evolution strategy introduced in recurrent graph encoders (Pareja et al., 2019), *which updates the parameters of the attention kernels directly over time.* This avoids storing the entire sequence of hyperbolic embeddings, significantly reduces memory usage, and enables efficient learning on long optimization traces while maintaining temporal smoothness (Pareja et al., 2019) (related work in Appendix C).

## 3 RESEARCH METHOD

Figure 1 provides an overview of our approach. Given a training trace of an MLP, we encode each checkpoint as a parameter graph where nodes represent neurons or biases and edges represent weights. The sequence of graphs is embedded in the Poincaré ball and processed with a hyperbolic graph attention network, while temporal evolution is handled through kernel meta–evolution that updates attention parameters recurrently across snapshots. The model outputs link probabilities and signed weight predictions, and node embeddings are refined by intrinsic Riemannian optimization. This yields compact temporal representations that capture both structural symmetries and geometric self-organization of the network over training.

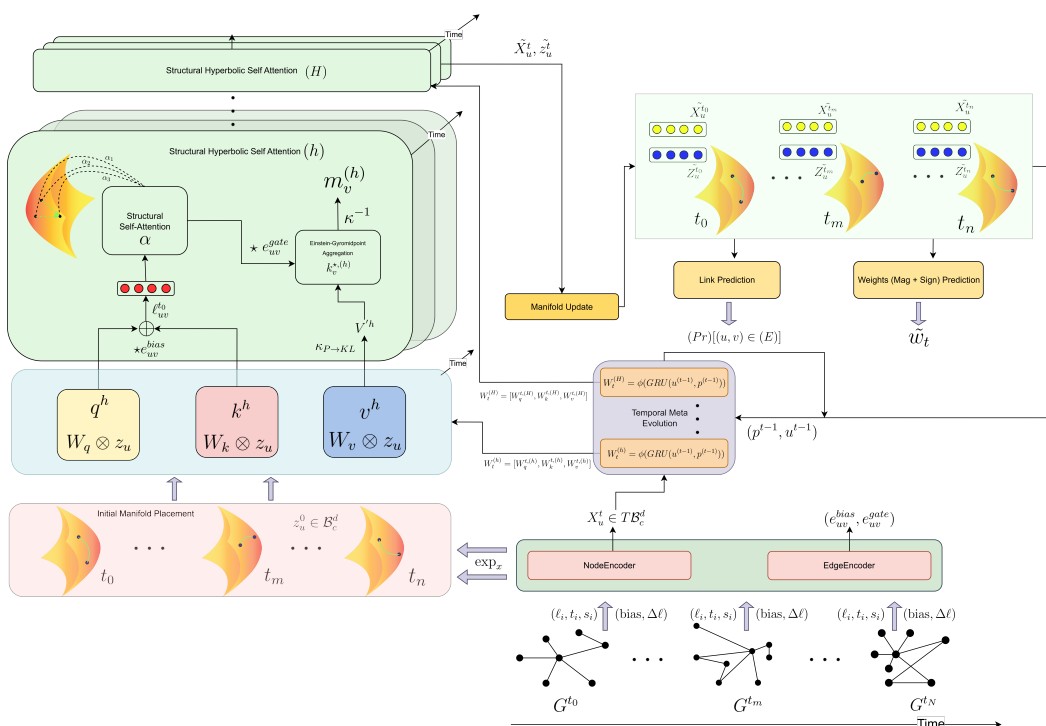

Figure 1: **Architecture of the Geometric Temporal Hyperbolic Graph Meta-Network (GTH-GMN).**

### 3.1 TEMPORAL GRAPH CONSTRUCTION FROM MLP TRACES

We represent the evolving weight matrices of an MLP as a sequence of graph snapshots of parameters, ensuring that the permutation symmetries are respected. At each epoch $t$, individual neurons and biases are represented as nodes ($u \in V_t$), while the parameters define the weighted edges ($\{u, v\} \in E_t$ with signed label $w_{uv,t}^\star$). Nodes with zero weights at epoch $t$ do not register adjacency (Lim et al., 2023; Kofinas et al., 2024). For an MLP with $L$ layers and widths $(n_0, \ldots, n_L)$, this yields a snapshot $G_t = (X_t, E_t, W_t)$. Input and output layers are fixed so that reordering neurons within hidden layers results only in an equivariant graph structure (Lim et al., 2023). The set of nodes includes inputs, hidden neurons, and explicit bias nodes (Lim et al., 2023; Kofinas et al., 2024). Each node feature $(x_{u,t})$ contains a layer label $\ell(u)$ and a type $\tau(u)$. The neuron edges connect successive layers, while the bias edges connect a bias node to its layer. Directed pairs are symmetrized by averaging across antiparallel directions, treating missing partners as zero, so that

the undirected edge set $E_t$ preserves signed magnitudes without spurious artifacts. Each node $(x_{u,t})$ is further endowed with structural signals that remain invariant under permutations. From the edges incident to $u$ we compute absolute and signed strength scores and then standardize them across the graph (Unterthiner et al., 2021). The resulting $z$–scores $(\widetilde{s}^{\mathrm{abs}}(u,t), \widetilde{s}^{\mathrm{sgn}}(u,t))$ quantify how strongly a node is embedded in its neighborhood. Together with $\ell(u)$ and $\tau(u)$, they yield the feature vector $x_{u,t} = \left[\ell(u), \tau(u), \widetilde{s}^{\mathrm{abs}}(u,t), \widetilde{s}^{\mathrm{sgn}}(u,t)\right]$. Edge attributes $(e_{uv,t})$ are deliberately minimal: they encode only whether an edge is a bias connection and whether it spans layers, never leaking functions of $w_{uv,t}^{\star}$ into the model. Repeating this construction produces a temporal sequence $\{G_t\}_{t=1}^{T}$ that captures the self-organization of the MLP while remaining invariant to hidden layer permutations.

All descriptors $(x_{u,t}, e_{uv,t})$ are embedded in the Euclidean tangent space at the origin $T_0 \mathbb{B}_c^d \simeq \mathbb{R}^n$, which provides a stable learning backbone. Discrete labels $(\ell(u), \tau(u))$ are mapped through learned embeddings, while standardized strength features are passed through a small MLP. A residual block refines the result, producing a $n$-dimensional representation $X_{u,t} \in T_0$. For edges, the binary context $e_{uv,t} = (\mathrm{is\_bias}, \Delta\ell)$ is encoded in a bias term $b_{uv,t}$ and a gate of near identity $g_{uv,t}$ using an MLP, which modulates attention scores without altering the geometry of the manifold. Thus, normalization, dropout, and mixing remain stable in Euclidean space, while curvature is introduced only in the subsequent hyperbolic message passing layers (Li et al., 2024; Zhang et al., 2019; Ganea et al., 2018). The final output is a temporal sequence of parameter graphs with node characteristics $(X_{u,t})$ and edge characteristics $(e_{uv,t}, b_{uv,t}, g_{uv,t})$ that respect the permutation symmetry and provide a clean basis for the prediction and regression of semi-supervised link probability.

## 3.2 Hyperbolic Graph Attention Layer: Intuition and Formulation

The goal of this layer is to compute the updated positions of the nodes on the ball by applying manifold-aware structural self-attention, which respects the curvature of the geometry as well as the temporal evolution of parameter graphs. Each parameter graph snapshot $G_t$ lives on the Poincaré ball, which naturally encodes hierarchical and heavy–tailed structure. A hyperbolic attention layer, therefore, cannot simply apply Euclidean linear maps: every projection, comparison, and aggregation must respect the curvature of the underlying manifold. We follow a simple principle throughout: *compute in the tangent space, transport to the ball only when geometry matters, and aggregate in a model where barycenters (a manifold-consistent version of weighted average) are well defined.* This yields a stable attention mechanism whose scores and updates have direct geometric meaning (Li et al., 2024; 2023).

**Hyperbolic Affine Maps (Q/K/V on the Ball).** Each attention head first computes hyperbolic query, key, and value vectors, which act as the basic attention primitives (equation 2): the queries encode how a node seeks information, the keys encode how it is compared to its neighbors in hyperbolic space, and the values carry the features that will be aggregated along curved geodesics. Since linear maps are not directly compatible with curved geometry, we realize them as tangent space projections, followed by exponential lifting, which transports the updated tangent vector back onto the manifold along the geodesic defined at that point (Ganea et al., 2018; Li et al., 2024). Given the Euclidean parameters $W$ and $b$, the hyperbolic affine map ($\Phi$) is defined as

$$\Phi_{W,b}^{(t)}(x) = \exp_{\Phi_W^{(t)}(x)}\left(\mathrm{PT}_{0 \to \Phi_W^{(t)}(x)}(b)\right), \qquad \Phi_W^{(t)}(x) = \exp_0\left(W \log_0(x)\right), \qquad (1)$$

where $\log_0$ and $\exp_0$ move between the tangent space and the ball at the origin, and $\mathrm{PT}$ denotes parallel transport. This construction should be viewed as the hyperbolic analogue of a linear map plus bias: the matrix $W$ acts in the flat tangent space, while $b$ is transported to the tangent space at $\Phi_W^{(t)}(x)$ where it is applied before lifting back onto the ball. Applying $\Phi$ to each node yields the corresponding hyperbolic query, key, and value vectors as follows:

$$q_i^{(r,t)} = \Phi_{W_{\mathrm{q}}^{(r,t)}, b_{\mathrm{q}}^{(r,t)}}^{(t)}(z_i^{(t)}), \quad k_i^{(r,t)} = \Phi_{W_{\mathrm{k}}^{(r,t)}, b_{\mathrm{k}}^{(r,t)}}^{(t)}(z_i^{(t)}), \quad v_i^{(r,t)} = \Phi_{W_{\mathrm{v}}^{(r,t)}, b_{\mathrm{v}}^{(r,t)}}^{(t)}(z_i^{(t)}). \qquad (2)$$

**Attention Logits and Structural Gating.** Attention on the Poincaré ball must measure similarity by geodesic distance, in line with prior hyperbolic graph attention formulations (Zhang et al., 2019; Li et al., 2023; 2024). For each directed edge $(u \to v)$, we therefore score proximity as the negative hyperbolic distance, since smaller geodesic distances correspond to stronger similarity, and using the

negative ensures that nearer nodes receive exponentially larger attention weights under the softmax (equation 3).

$$\theta_{u \to v}^{(r,t)} = - d_c(q_u^{(r,t)}, k_v^{(r,t)}) + b_{uv}^{(t)}, \tag{3}$$

where $b_{uv}^{(t)}$ injects edge context such as bias indicator or layer separation. Softmax normalization produces the structural attention coefficients in the following manner

$$\alpha_{u \to v}^{(r,t)} = \frac{\exp\big(\theta_{u \to v}^{(r,t)}\big)}{\sum_{w:(w \to v) \in E_t} \exp\big(\theta_{w \to v}^{(r,t)}\big)} \quad , \quad \widetilde{\alpha}_{u \to v}^{(r,t)} = \frac{\alpha_{u \to v}^{(r,t)} \, g_{uv}^{(t)}}{\sum_{w:(w \to v) \in E_t} \alpha_{w \to v}^{(r,t)} \, g_{wv}^{(t)}}. \tag{4}$$

Because some edges carry more structural influence (e.g., cross–layer or bias connections), we modulate these coefficients using a gate $g_{uv}^{(t)}$ (right part of equation 4). Renormalizing yields $\widetilde{\alpha}_{u \to v}^{(r,t)}$, which behaves like standard attention but incorporates curvature–aware structural reweighting (Li et al., 2023; 2024), which is achieved by multiplying each coefficient by its gate and normalizing across the gated neighbors so that their relative influence is correctly rescaled.

**Geometric Aggregation via Einstein Gyromidpoints.** Averaging vectors in curved space cannot be done linearly because Euclidean addition does not respect the manifold's geometry: the sum of two points typically lies outside the manifold and does not correspond to any meaningful geodesic midpoint. We therefore use Einstein gyromidpoints, computed in the Klein model where closed–form barycenters exist (Zhang et al., 2019; Li et al., 2024). The Poincaré to Klein map ($\kappa$) and its inverse are

$$\kappa(z) = \frac{2\sqrt{c} \, z}{1 + c\|z\|^2}, \qquad \kappa^{-1}(y) = \frac{1}{\sqrt{c}} \frac{y}{1 + \sqrt{1 - \|y\|^2}}. \tag{5}$$

with Lorentz factor $\gamma(y) = 1/\sqrt{1 - \|y\|^2}$. For node $v$ and head $r$, the Klein barycenter is

$$\kappa\Big(m_v^{(r,t)}\Big) = \frac{\sum_{u:(u \to v) \in E_t} \widetilde{\alpha}_{u \to v}^{(r,t)} \, \gamma(\kappa(v_u^{(r,t)})) \, \kappa(v_u^{(r,t)})}{\sum_{u:(u \to v) \in E_t} \widetilde{\alpha}_{u \to v}^{(r,t)} \, \gamma(\kappa(v_u^{(r,t)}))}. \tag{6}$$

Mapping back and returning to the tangent space gives the multi–head aggregation (right-part equation 7) for computing the node update, which is simply the average of these tangent updates:

$$m_v^{(r,t)} = \kappa^{-1}\Big(\kappa(m_v^{(r,t)})\Big), \qquad \widehat{m}_v^{(r,t)} = \log_0(m_v^{(r,t)}), \quad M_v^{(t)} = \frac{1}{H} \sum_{r=1}^{H} \widehat{m}_v^{(r,t)}. \tag{7}$$

This design has a critical interpretive consequence: attention uses geodesics to decide "who influences whom," while aggregation respects the global curvature, ensuring that hierarchical structure, hub nodes, and boundary effects are preserved. Each node then receives a gated residual update (addition in the tangent space followed by exponential lifting). A projection step enforces a safety margin inside the ball to avoid numerical instability near the boundary (Li et al., 2024).

**Temporal Evolution of Attention Kernels.** Now, to model temporal smoothness across snapshots, we meta–evolve the attention parameters using a GRU, inspired by (Pareja et al., 2019). For each layer $\ell$ and head $r$, we maintain a recurrent state $u_{\ell,r}^{(t)}$ as follows:

$$p_\ell^{(t-1)} = \frac{1}{N} \sum_{i=1}^{N} X_{\ell,i}^{(t-1)}, \quad u_{\ell,r}^{(t)} = \text{GRU}\Big(p_\ell^{(t-1)}, u_{\ell,r}^{(t-1)}\Big). \tag{8}$$

The GRU emits fresh Q/K/V parameters via an MLP:

$$W_{\ell,r,k/q/v}^{(t)} = \phi_{\text{MLP}}\big(W_{\text{out}} u_{\ell,r}^{(t)} + b_{\text{out}}\big) \in \mathbb{R}^{3d^2 + 3d}, \tag{9}$$

which are then used in the hyperbolic affine maps (1)–(2). This mechanism allows the attention kernels to evolve smoothly over time without storing past embeddings, providing temporal consistency while preserving permutation invariance and limiting memory to compact recurrent states.

## 3.3 SIGNED WEIGHT REGRESSION

Before introducing the mathematical details of this module, we outline its role in the architecture. A standard link predictor estimates whether an edge should exist, but in weighted networks, we also need to recover the actual strength and polarity of each connection. Magnitude reflects how strongly two nodes influence each other, while the sign encodes the direction of that influence. In many real networks, these magnitudes follow heavy-tailed distributions (Song et al., 2005; Krioukov et al., 2010; Wei et al., 2013; Allard et al., 2017), and hyperbolic distance offers a natural geometric proxy for such patterns. The signed regression head therefore serves two purposes: it provides additional supervision for the temporal encoder, and it imposes a geometric prior that ties magnitudes to hyperbolic distances while modeling polarity in the tangent space. In contrast, Euclidean regressors or generic MLP decoders (Goyal et al., 2018) ignore curvature and risk losing this interpretive structure.

**Magnitude as a Power Law of Hyperbolic Distance.** We begin by predicting node-dependent scale and decay factors from the tangent features $X_i^{(t)} \in T_0\mathbb{B}_c^d \simeq \mathbb{R}^d$, $s_i = f_\sigma(X_i^{(t)})$, $k_i = f_\kappa(X_i^{(t)})$, where $f_\sigma$ and $f_\kappa$ are MLPs. $s_i$ & $\kappa_i$ adjust the local steepness and scale of the magnitude–distance relation. For each edge $(u, v)$ these corrections are used symmetrically as $(s_u, s_v)$ and $(k_u, k_v)$. The slope of the power law is then allowed to vary slightly with local context:

$$\alpha_{uv} = \alpha_0 + \delta\alpha \cdot \tanh\big(\phi_{\mathrm{MLP}}([X_u^{(t)}, X_v^{(t)}, e_{uv}])\big), \tag{10}$$

where $\alpha_0$ is a global baseline, $\delta\alpha$ a small deviation, and $\phi_{\mathrm{MLP}}$ an MLP (Allard et al., 2017). This design captures mild structural asymmetries between edge types while keeping the decay law interpretable and stable. Given these components, the predicted magnitude follows a heavy-tailed power law (Allard et al., 2017):

$$\widehat{|w_{uv}^{(t)}|} = \exp(\log\nu + s_u + s_v)\exp\big(-(1 - \tfrac{\alpha_{uv}}{d})(k_u + k_v)\big)\big(d_{uv}^{(t)}\big)^{-\alpha_{uv}}, \tag{11}$$

where $\nu > 0$ is a global scale and $d_{uv}^{(t)} = d_c(z_u^{(t)}, z_v^{(t)})$ is the Poincaré distance. Large magnitudes, therefore, correspond to smaller hyperbolic distances, while small magnitudes produce larger metric separation (Simas & Rocha, 2014). This establishes a direct geometric interpretation of weight strength.

**Polarity in the Tangent Space.** Distances on the manifold are positive definite (Simas & Rocha, 2014). Polarity is therefore modeled as a directional effect in the tangent space, where inner products are well defined. We map $z_v^{(t)}$ into the Tangent Space at $z_u^{(t)}$ using the logarithmic map (equation 12),

$$\delta_{u\to v}^{(t)} = \log_{z_u^{(t)}}(z_v^{(t)}), \quad \eta_u = W_\eta X_u^{(t)}, \quad \xi_u = \frac{\eta_u}{\lambda_{z_u^{(t)}}\|\eta_u\|}, \quad \lambda_{z_u^{(t)}} = \frac{2}{1 - c\|z_u^{(t)}\|^2}, \tag{12}$$

and normalize the source feature $\eta_u$ by the conformal factor so that directionality is consistent with the Riemannian metric. The sign logit is then given by a hyperbolically consistent inner product,

$$s_{uv}^{(t)} = \beta\langle\delta_{u\to v}^{(t)}, \xi_u\rangle, \qquad \widehat{w}_{uv}^{(t)} = \widehat{|w_{uv}^{(t)}|}\tanh(s_{uv}^{(t)}). \tag{13}$$

This construction separates magnitude and polarity in a geometrically faithful manner (equation 13). A distance-based heavy-tailed law controls magnitude (equation 11), while polarity arises from the angle between transported displacements and conformally normalized source features (equation 12). As a result, short hyperbolic distances naturally represent strong connections, whereas longer distances encode weaker or negligible ones, and the directional alignment in the tangent space determines the sign. The module therefore provides an interpretable decomposition of each weight into strength and direction, consistent with the geometry of the Poincaré ball.

## 3.4 LINK DECODER, SUPERVISION, AND OBJECTIVE

To translate the hyperbolic embeddings into edge predictions, we use the Fermi–Dirac decoder ($\psi^{(t)}$) (equation 14) (Nickel & Kiela, 2017; 2018; Wen et al., 2024), which links *connectivity* directly to the geodesics on the Poincaré ball, for a given time step $t$:

$$\psi^{(t)}(u, v) = \Big(1 + \exp\big(\tfrac{d_c(z_u^{(t)}, z_v^{(t)}) - R}{T}\big)\Big)^{-1}. \tag{14}$$

Here, the radius $R$ controls the effective neighborhood boundary and the temperature $T$ controls the sharpness of the transition, both learned during training. Under this decoder, nodes that lie close in hyperbolic distance are classified as likely neighbors, and distant nodes as unlikely. Training uses a binary cross-entropy loss together with dynamic negative sampling from both same-layer and uniform distributions. Moreover, connectivity alone does not describe the strength or direction of influence between two parameters. For this reason, we include a signed weight regressor (Section 3.3) that predicts the full value of each positive edge. The regression loss encourages the predicted weight to match the true magnitude, while a separate classification loss predicts its polarity. The magnitude loss measures the difference between predicted and true weights, while the polarity loss compares the predicted sign logit with the binary label indicating whether the weight is positive or negative. Formal expressions for these losses and their components are provided in Appendix G. This design ties geometric separation to functional strength: large-magnitude edges correspond to smaller hyperbolic distances, and the sign is encoded through directions in the tangent space. As a result, the embeddings acquire semantic meaning: hubs drift outward, stronger edges cluster radially, and excitatory versus inhibitory effects appear as angular variation. To maintain coherent geometry across time, we add a small set of regularizers. A slope prior keeps the decay exponents near a global baseline, a temporal smoothness term limits abrupt movement between snapshots, and a ranking loss enforces that stronger interactions correspond to shorter hyperbolic distances. The overall objective is a weighted sum of the Fermi–Dirac cross-entropy, the magnitude and sign supervision terms, and these geometric regularizers. Training follows a gentle curriculum that anneals the ranking margin and progressively hardens negative sampling to stabilize learning before introducing difficult examples. Ablations about specific terms can be found in Appendix J.

Optimization proceeds in two distinct phases. This separation is necessary because the attention stack and regressors operate in Euclidean space, whereas the node embeddings evolve on a curved manifold; using a single update rule would either ignore curvature or force the Euclidean modules to inherit manifold constraints they were not designed for. In the first phase, a Euclidean pass updates all kernel weights, regressors, and the link decoder using standard backpropagation. In the second phase, a Riemannian pass refines the node positions $z^{(t)}$ directly on the Poincaré ball. This intrinsic step begins by rescaling Euclidean gradients with the conformal factor $\lambda_z^2 = 2/(1 - c\|z\|^2)$ so that updates follow the manifold geometry rather than the ambient space (Bridson & Haefliger, 1999; Bonnabel, 2013; Nickel & Kiela, 2017). To stabilize learning, first and second moments of these gradients are accumulated in a RAMSGrad–style procedure (Bécigneul & Ganea, 2019; Sakai & Iiduka, 2020), which extends adaptive optimization to curved spaces. Each accepted update is applied via the exponential map, projecting the tangent step back onto the manifold. Finally, moment vectors are parallel transported to the new tangent space, keeping the optimizer's memory coherent across iterations. This two–phase design decouples the stability of the Euclidean attention stack from the dynamics of hyperbolic embeddings, ensuring that geometry is respected while long training traces remain stable. We describe the algorithm in Appendix G. Overall, this scheme does more than minimize loss: it shapes an embedding space where magnitudes follow radial distance, polarity is encoded in tangent orientation, and temporal smoothness emerges from controlled geodesic motion. Geometry thus becomes a built-in inductive bias for compact and interpretable representations.

## 4 EXPERIMENTAL SETUP AND RESULTS

**Classification of Images via INR Traces.** This experiment evaluates whether the temporal evolution of the parameter graph of the intrinsic neural representation (INR) of an image contains enough structure for a meta-network to infer the class of the underlying image. Crucially, the meta-network never sees the image itself; it only receives the evolution of the INR weights over optimization. We fit each image from MNIST and Fashion-MNIST using a shallow INR, recording its parameter values over optimization, yielding a temporal trace of parameter graphs. Each trace contains $T \in [80 - 100]$ checkpoints, i.e., a sequence of signed parameter-graphs. Dataset splits are made at the *image* level to avoid leakage; we use $45,000$ images for training, $5,000$ for validation, and $10,000$ for testing. A spatio-temporal meta-network is pretrained with intrinsic (hyperbolic) optimization on a joint link-prediction and signed weight-regression objective over the full kept trace per image, along with other losses in our curriculum; within each checkpoint, edge embeddings are pooled to a snapshot vector, and snapshot vectors are then pooled over time to produce a fixed $n$-dimensional representation. A small feed-forward classifier is trained on these representations to predict the digit or fashion class. The same encoding procedure is applied at validation and test time, and we report classification accuracy on the held-out test images. Results are shown in Table 1.

Table 1: **INR classification in static vs. temporal settings.** Average test accuracies ($\pm$ standard error) on MNIST and Fashion-MNIST INRs. Baselines are reproduced from DWSNets (Navon et al., 2024) and NFNs (Zhou et al., 2023a). Our spatio-temporal variant uses pooled representations across $K$ traces.

| Model | MNIST INR | Fashion-MNIST INR |
|---|---|---|
| MLP | $17.55 \pm 0.01$ | $19.91 \pm 0.47$ |
| MLP + Perm. aug | $29.26 \pm 0.18$ | $22.76 \pm 0.13$ |
| MLP + Alignment | $58.98 \pm 0.52$ | $47.79 \pm 1.03$ |
| INR2Vec (Arch.) | $23.69 \pm 0.10$ | $22.33 \pm 0.41$ |
| Transformer | $26.57 \pm 0.18$ | $26.97 \pm 0.33$ |
| DWSNets | $85.71 \pm 0.57$ | $67.06 \pm 0.29$ |
| NFN$_{\text{HNP}}$ | $92.5 \pm 0.07$ | $72.7 \pm 1.53$ |
| NFN$_{\text{NP}}$ | $92.9 \pm 0.22$ | $75.6 \pm 1.07$ |
| **Ours (Temporal, $K$-trace pooling)** | $\mathbf{95.6 \pm 0.18}$ | $\mathbf{80.72 \pm 0.29}$ |

**Predicting DNN Generalization from Weights.** This experiment tests whether the *temporal evolution* of the parameters of a model contains a predictive signal about its test accuracy. More specifically, can we recover accuracy-related information solely from a deep neural network's weight structure? Here, each trial $i$, corresponds to an MLP trained on a subset of the CIFAR-10 dataset. For each trial $i$ we gather checkpoints $\{\theta_{i,t}\}_{t=1}^{T_i}$ and the corresponding test accuracy (per-epoch) $y_{i,t}$, and convert every checkpoint to a parameter graph $G_{i,t}$ with corresponding node/edge features. Traces are encoded over fixed-length (stride-rolled) windows or complete sequences, with *last-step* supervision, i.e., the intermediate steps evolve while the final step is supervised. We classify the divisions by trial, and the fidelity of the classification is measured using Kendall's $\tau$ (Zhou et al., 2023a; Navon et al., 2024; Lim et al., 2023). See Results in (Table 2).

Table 2: Kendall's $\tau$ correlation on the CIFAR-10 benchmark. This table compares the rank correlation between predicted and ground-truth accuracies of neural architectures (INRs) as captured by different meta-network representations. Higher values indicate stronger consistency between the learned representation space and the empirical performance ranking of architectures (Unterthiner et al., 2021; Zhou et al., 2023a).

| Metanetwork | CIFAR-10 |
|---|---|
| **NFN$_{\text{HNP}}$** | $\mathbf{0.934 \pm 0.001}$ |
| NFN$_{\text{NP}}$ | $0.922 \pm 0.001$ |
| StatNN | $0.915 \pm 0.002$ |
| GTH-GMN (ours) | $0.846 \pm 0.004$ |

Table 3: Results of meta-net representations on sinusoid-MLPs. Entries report **test MSE** of a downstream MLP regressor trained on the learned representations. We compare with other baselines by Lim et al. (2023); Navon et al. (2024); Zhou et al. (2023a).

| Metanetwork | Test MSE |
|---|---|
| MLP | $7.39 \pm 0.19$ |
| MLP + Perm. aug | $5.65 \pm 0.01$ |
| MLP + Alignment | $4.47 \pm 0.15$ |
| INR2Vec (Arch.) | $3.86 \pm 0.32$ |
| Transformer | $5.11 \pm 0.12$ |
| DWSNets | $1.39 \pm 0.06$ |
| GMN | $1.13 \pm 0.08$ |
| GTH-GMN (ours) | $1.06 \pm 0.24$ |

**Predicting Sine Wave Frequency.** Additionally, we also evaluate our temporal model on the benchmark introduced by (Navon et al., 2024), in which each input network is an MLP trained to fit a one-dimensional sinusoidal function of the form $x \mapsto a\sin(bx)$, with coefficients $a, b \in \mathbb{R}$ varying across samples. The goal is to learn a meta-network encoder that maps these trained MLP temporal parameter graphs to meaningful representations, while remaining invariant to neuron permutations, where representation quality is assessed in a contrastive-learning framework inspired by SimCLR (Lim et al., 2023; Navon et al., 2024; Zhou et al., 2023a; Chen et al., 2020). We introduce a *temporal* variant: link $K$ successive checkpoints per INR, build an edge–centric hyperbolic representation, mean–pool, and predict $(a, b)$ with a small MLP. See results in Table 3.

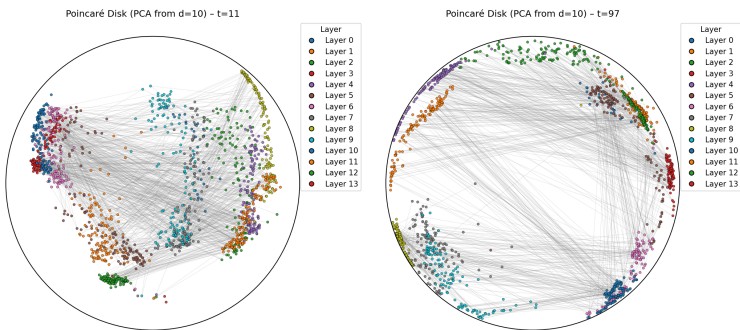

Figure 2: Hyperbolic embeddings of parameter graphs from a 13-layer MLP trained as an INR on CIFAR-10, shown at checkpoints $t = 11$ (left) and $t = 97$ (right). Embeddings are learned in $d = 10$ dimensions with 4 HGAT layers and 4 EvolveGCN–H kernels (tangent dim. 128) and projected to the Poincaré disk via PCA. Nodes are colored by layer index, with edges in gray, illustrating the progressive reorganization and separation of layers over training.

## 5 DISCUSSION

**Interpreting the Results.** Across the three experiments in Section 4, we aimed to test whether a single temporal hyperbolic meta network can generalize across very different prediction tasks. The results suggest that the evolution of the parameter graph carries a stable and predictive structure. In the INR classification, the encoder achieves average test accuracies of $95.6 \pm 0.18\%$ on MNIST and $80.72 \pm 0.29\%$ on Fashion MNIST, revealing that the way INRs self-organize during training encodes class-specific geometric structures that the model is able to consistently separate. In the CI-FAR 10 generalization prediction task, we achieve a relatively modest Kendall's $\tau = 0.846 \pm 0.004$ compared to the neural functional baselines. Our factorization emphasizes global geometric structure and temporal coherence over preserving all fine-grained tensor details. It has been shown that representation learning of neural networks using Autoencoders often exhibits a smoothening bias as MSE reconstruction loss tends to coarse-averaged representations (Meynent et al., 2025). In contrast, permutation-equivariant neural functionals operate directly on raw weight tensors (Zhou et al., 2023a) and retain more accuracy-correlated micro-structure at the cost of reduced geometric interpretability. This likely contributes to the relatively lower Kendall's $\tau$ achieved over the CIFAR-10 dataset (Fu et al., 2021). We see this as a trade-off between raw performance over the generalization prediction task and generating geometric and interpretable embeddings of the underlying DNN. Moreover, over the sinusoid benchmark, our temporal hyperbolic model reaches the $1.06 \pm 0.24$ test MSE, comparable in mean to GMN's $1.13 \pm 0.08$, though with higher variance. This arises from the interaction of curvature (Fu et al., 2021) and temporal coupling: in the Poincaré ball, the effective step size increases with radius, so small early differences can produce divergent trajectories, and the recurrent kernel further amplifies such variation (Nickel & Kiela, 2017; Bonnabel, 2013; Ganea et al., 2018). Variability can be reduced with more conservative Riemannian step control, stronger temporal regularization, or light ensembling, which are standard techniques for stabilizing hyperbolic and recurrent models (Ganea et al., 2018; Chen et al., 2024; Sun et al., 2021b).

**Implications for Explainability.** The embeddings reveal coherent patterns of self-organization: points drift toward the boundary, layers separate in both radial and angular directions, and trajectories evolve smoothly across training (Figure 2). These behaviors suggest several potential signals for intrinsic explainability. Radial position and drift may reflect how influence strengthens or weakens over time; angular separation may indicate the emergence of functional specialization; geometric proximity and repeated strong connections may reveal central processing hubs; and consistent magnitude–distance alignment may highlight connections that are both geometrically and functionally salient. Conversely, nodes whose trajectories collapse together or drift outward uniformly may signal redundancy and pruning opportunities (Nickel & Kiela, 2017; Ángeles Serrano et al., 2008). Additionally, Appendix Section K explains how our approach can be extended beyond multilayered perceptrons (MLPs).

ACKNOWLEDGEMENTS

I thank Eindhoven University of Technology (TU/e) for providing access to the Umbrella cluster, which supported part of the experimentation. Moreover, I would like to extend my thanks to Dr. Indika Kumara and Dr. Damian Tamburri, from Tilburg University and Eindhoven University respectively for their supervisory contribution during the course of this project. I also acknowledge SURF (www.surf.nl) for enabling use of the National Supercomputer Snellius. The use of large language models (LLMs) was restricted exclusively to polishing the text for brevity under space constraints. The author(s) take full responsibility for the research ideation and the experimentation conducted and reported in this work.

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

## APPENDIX

## A    NOTATIONS

| **Parameter Graphs & Features** | |
|---|---|
| $G_t$ | Parameter-graph snapshot at epoch $t$ |
| $V,\ E_t$ | Node set; edge set at epoch $t$ |
| $w_{uv}^{\star}$ | Signed edge label on $\{u, v\}$ |
| $e_{uv}$ | Edge context (bias/kernel, interlayer flag) |
| $L,\ d,\ c$ | #layers; embedding dim; curvature parameter |
| $\ell(u)$ | Layer label of node $u$ |
| $\tau(u)$ | Type label (input / neuron / bias) |
| $s_{\text{abs}}(u)$ | Incident absolute-weight sum at $u$ |
| $s_{\text{sgn}}(u)$ | Incident signed-weight sum at $u$ |
| $\tilde{s}_{\text{abs}},\ \tilde{s}_{\text{sgn}}$ | Standardized node-weight signals |
| $\boldsymbol{x_u}$ | Final node feature vector |
| $V_\ell,\ N_\ell$ | Nodes in layer $\ell$; their count |
| **Temporal States and Recurrence** | |
| $t$ | Time/epoch index |
| $X_i^{(t)}$ | Tangent-backbone feature of node $i$ at $t$ |
| $p_\ell^{(t-1)}$ | Mean-pooled tangent for layer $\ell$ |
| $u_{\ell,r}^{(t)}$ | Recurrent state (layer $\ell$, head $r$) |
| $\theta_{\ell,r}^{(t)}$ | Packed attention parameters (head $r$) |
| **Hyperbolic Geometry (Poincaré Ball)** | |
| $\mathbb{B}_c^d$ | $d$-dimensional Poincaré ball |
| $d_{\mathbb{B}}(z_i, z_j)$ | Geodesic distance on $\mathbb{B}_c^d$ |
| $\exp_x(\cdot),\ \log_x(\cdot)$ | Exponential / logarithmic maps at $x$ |
| $\lambda_z$ | Conformal factor at point $z$ |
| $\text{PT}_{x \to y}(\cdot)$ | Parallel transport from $x$ to $y$ |
| $K(\cdot),\ K^{-1}(\cdot)$ | Poincaré$\leftrightarrow$Klein coordinate maps |
| **Hyperbolic Attention Block** | |
| $r$ | Attention head index |
| $q_u^{(r,t)},\ k_v^{(r,t)},\ v_u^{(r,t)}$ | Query, key, value points on $\mathbb{B}_c^d$ |
| $\tau^{(r,t)}$ | Softmax temperature (head $r$) |
| $b_{uv}$ | Edge bias from context |
| $g_{uv}$ | Edge gate from context |
| $\alpha_{u \to v}^{(r,t)}$ | Attention coefficient (head $r$) |
| $m_v^{(r,t)}$ | Head message in tangent space |
| $M_v^{(t)}$ | Multi-head combined message |
| $z_v^{(t)}$ | Node position update on $\mathbb{B}_c^d$ |
| $\mathcal{N}(v)$ | Neighborhood of node $v$ |
| $y$ | Klein coordinate ($\|y\| < 1$) |
| $\gamma(y)$ | Lorentz factor in Klein model |

**Decoders and Regressors**

| | |
|---|---|
| $\psi^{(t)}(u,v)$ | Fermi–Dirac link probability |
| $R,\ T$ | Link radius; temperature (FD decoder) |
| $\hat{w}_{uv}^{(t)},\ |\hat{w}_{uv}^{(t)}|$ | Predicted signed weight; magnitude |
| $\alpha_{uv}$ | Distance–magnitude exponent (edge-wise) |
| $\nu,\ \beta$ | Magnitude scale; sign-logit scale |
| $\delta_{u \to v}^{(t)}$ | Hyperbolic displacement (source $u$ to $v$) |
| $\xi_u$ | Conformal direction for sign prediction |
| $s_{uv}^{(t)}$ | Sign logit |
| $\varepsilon$ | Small positive constant (stability) |

**Losses and Optimization**

| | |
|---|---|
| $\mathcal{L}_{\mathrm{FD}}$ | BCE loss for Fermi–Dirac link decoder |
| $\mathcal{L}_{\mathrm{wrec}}$ | Magnitude regression loss |
| $\mathcal{L}_{\mathrm{sign}}$ | Sign prediction loss (BCE) |
| $\mathcal{L}_{\alpha}$ | Prior/regularizer on $\alpha_{uv}$ |
| $\mathcal{L}_{\mathrm{smooth}}$ | Temporal smoothness penalty |
| $\mathcal{L}_{\mathrm{rank}}$ | Pairwise ranking loss |
| $E^{+},\ E^{-}$ | Positive edges; sampled negatives |
| $\gamma_{\mathrm{rank}}$ | Ranking margin parameter |
| $\nabla_z^{\mathrm{Euc}}\mathcal{L},\ \nabla_z^{\mathrm{Riem}}\mathcal{L}$ | Euclidean / Riemannian gradients |
| $\Delta_i$ | Intrinsic update step at node $i$ |
| $z_i^{(t+1)}$ | Updated node position |
| $\eta$ | Learning rate |

## B  THEORETICAL PRELIMINARIES

### B.1  RIEMANNIAN MANIFOLD

A Riemannian manifold $(\mathcal{M}, g^{\mathcal{M}})$ is a smooth manifold $\mathcal{M}$ endowed with a Riemannian metric $g^{\mathcal{M}}$, which assigns to each tangent space $T_x\mathcal{M}$ at a point $x \in \mathcal{M}$ a positive definite inner product. The tangent space $T_x\mathcal{M}$ consists of all tangent vectors at $x$ and may be regarded as the best linear, first-order approximation to the manifold in a neighborhood of that point. The metric equips this linear space with a bilinear form $\langle \cdot, \cdot \rangle : T_x\mathcal{M} \times T_x\mathcal{M} \to \mathbb{R}$, from which a norm is induced by $\|v\|_g = \sqrt{g(v,v)}$ for any $v \in T_x\mathcal{M}$. With this structure, the manifold acquires the ability to measure lengths, angles, and volumes, thereby extending the familiar tools of Euclidean geometry to curved spaces. If $\gamma : [a,b] \to \mathcal{M}$ is a smooth curve, its Riemannian length is defined as $L(\gamma) = \int_a^b \|\gamma'(t)\|_g \, dt$, a quantity that depends on the metric through the induced norm on tangent vectors. Among all curves connecting two points $x, y \in \mathcal{M}$, geodesics play a central role: they are the locally length-minimizing paths, serving as the natural analogue of straight lines in Euclidean space. The Riemannian distance is accordingly given by $d_{\mathcal{M}}(x,y) = \inf_{\gamma} L(\gamma)$, where the infimum is taken over all smooth curves $\gamma$ with $\gamma(a) = x$ and $\gamma(b) = y$. At this point, fundamental questions of differential geometry naturally arise. Given a point $x \in \mathcal{M}$ and a tangent vector $v \in T_x\mathcal{M}$, how can one move from the linearized tangent space back to the curved manifold? Conversely, given two points $x, y \in \mathcal{M}$, how can their displacement be expressed in the linear coordinates of $T_x\mathcal{M}$? These questions motivate the introduction of the **exponential** and **logarithmic** maps. The exponential map at $x$, denoted $\exp_x : T_x\mathcal{M} \to \mathcal{M}$, takes a tangent vector $v \in T_x\mathcal{M}$ and returns the point on the manifold obtained by following the geodesic that begins at $x$ in the direction of $v$, parameterized so that unit time corresponds to unit length. In this way, the exponential map provides a principled mechanism for "lifting" linear displacements in $T_x\mathcal{M}$ into the curved geometry of $\mathcal{M}$. The logarithmic map, denoted $\log_x : \mathcal{M} \to T_x\mathcal{M}$, is defined locally as the inverse of the

exponential map. For a point $y \in \mathcal{M}$ sufficiently close to $x$, it yields the unique vector $v \in T_x\mathcal{M}$ such that $\exp_x(v) = y$. Thus, the log map addresses the complementary problem: it represents the displacement between manifold points in the linear framework of the tangent space. Taken together, these constructions endow the Riemannian manifold with a powerful interpretative scheme. Tangent vectors become infinitesimal displacements that may be integrated into finite motions along the manifold via $\exp_x$, while finite displacements can be linearized into tangent vectors through $\log_x$. This duality shows why exponential and logarithmic maps are often viewed as rigorous analogues of addition and subtraction on curved spaces, providing the formal apparatus that underpins the heuristic extension of Euclidean intuition to Riemannian geometry.

### B.2 POINCARÉ BALL MODEL

A hyperbolic manifold can be described as a Riemannian manifold of constant negative curvature $c$ with $c < 0$. Among the isomorphic realizations of hyperbolic space, three models are commonly employed: the Poincaré ball model, the Lorentz model, and the Klein model. In what follows, we focus on the Poincaré ball model, which offers a particularly convenient formulation for learning problems in hyperbolic geometry.

The Poincaré ball model is denoted as $(\mathbb{B}_c^n, g_x^B)$, where the domain $\mathbb{B}_c^n = \{x \in \mathbb{R}^n : \|x\|^2 < -1/c\}$ is the open $n$-dimensional ball of radius $1/\sqrt{|c|}$. Its Riemannian metric tensor is defined as $g_x^B = (\lambda_x^c)^2 g_x^E$, where $g_x^E$ is the Euclidean metric, i.e. the identity matrix $I_n$, and $\lambda_x^c = \frac{2}{1+c\|x\|^2}$ is the conformal factor that encodes the curvature dependence.

Given two points $x, y \in \mathbb{B}_c^n$, the induced geodesic distance is given by

$$d_B^c(x, y) = \frac{1}{\sqrt{|c|}} \cosh^{-1}\left(1 - \frac{2c\|x - y\|^2}{(1 + c\|x\|^2)(1 + c\|y\|^2)}\right). \tag{15}$$

Closed-form expressions for the exponential and logarithmic maps in the Poincaré ball model were derived in Ganea et al. (2018). The exponential map at $x \in \mathbb{B}_c^n$ applied to a tangent vector $v \in T_x\mathbb{B}_c^n$ takes the form

$$\exp_x^c(v) = x \oplus_c \left(\tanh\left(\sqrt{|c|}\frac{\lambda_x^c\|v\|}{2}\right)\frac{v}{\sqrt{|c|}\|v\|}\right), \tag{16}$$

where $\oplus_c$ denotes Möbius addition. This expression formalizes the process of mapping linear displacements in the tangent space to points on the curved manifold along the geodesic through $x$.

The logarithmic map, which serves as the inverse of the exponential map, is given by

$$\log_x^c(y) = \frac{2}{\sqrt{|c|}\lambda_x^c}\tanh^{-1}\left(\sqrt{|c|}\| - x \oplus_c y\|\right)\frac{-x \oplus_c y}{\| - x \oplus_c y\|}. \tag{17}$$

Here, $\log_x^c(y)$ produces the unique tangent vector at $x$ that corresponds to the geodesic displacement from $x$ to $y$.

Another key operation in hyperbolic geometry is *parallel transport*, which allows one to move a tangent vector from $T_x\mathbb{B}_c^n$ to $T_y\mathbb{B}_c^n$ along the geodesic connecting $x$ and $y$, without altering its length or intrinsic orientation. This operation is formally defined as

$$PT_{x \to y}^c(v) = \frac{\lambda_x^c}{\lambda_y^c}\text{gyr}[y, -x]\, v, \tag{18}$$

where $\text{gyr}[y, -x]$ is the gyration operator associated with Möbius addition, capturing the non-associative structure of hyperbolic translations Ungar (2007; 2022). Taken together, the distance function, exponential and logarithmic maps, and parallel transport furnish the Poincaré ball with the full differential-geometric apparatus needed for optimization and representation learning in spaces of constant negative curvature.

## C ADDITIONAL RELATED WORK

**Network science and geometric models** The study of complex networks has long emphasized the interplay between efficiency, clustering, and degree heterogeneity. Small-world properties, for instance, can be captured through measures of global and local efficiency (Latora & Marchiori, 2001).

Geometry itself may emerge from simple generative rules: simplicial-complex models yield clustered and modular graphs whose spectral dimension is finite, with curvature distributions obeying Gauss–Bonnet relations (Wu et al., 2015). Empirical evidence further suggests that hidden metric spaces shape both connectivity and weights, as shown by the strong coupling between triangle formation and edge strengths (Allard et al., 2017).

A recurrent observation is the coexistence of heavy-tailed degree distributions and high clustering (Kim et al., 2007a; Serrano et al., 2008; Amaral et al., 2000), a structural combination that classical random-graph or lattice-based models cannot reproduce (Newman, 2010; Cohen & Havlin, 2010). Renormalization studies reveal that such networks are often self-similar: under box covering, they remain scale-free, exhibiting fractal organization (Song et al., 2005). This fractality can be traced to critical branching "skeletons," whereas supercritical skeletons drive small-world behavior (Kim et al., 2007b). Beyond standard box dimension, information-dimension measures have been developed to capture self-similar structure in real and weighted networks (Wei et al., 2013). A recent synthesis unifies these microscopic and macroscopic exponents, demonstrating robust scaling laws across web, brain, and citation networks (Fronczak et al., 2024).

Such findings motivate latent-space models in which link probabilities decay with hidden distance (Papadopoulos et al., 2008; 2012). The hyperbolic framework of Krioukov et al. (2010) provides a particularly powerful resolution, reconciling hierarchical expansion with geometric growth. In this view, the popularity–similarity model represents degree through radial coordinates and similarity through angular ones, thereby unifying hub formation with community structure across diverse real-world systems (Krioukov et al., 2010; Allard et al., 2017).

**Neuroscience and Graph Learning** Neurobiological work has used graph theory to study how anatomical and functional connectomes are organized, translating empirical association matrices into graphs and then asking what principles explain their structure. Early cortical mapping of the cat showed that laminar hierarchy and nonmetric multidimensional scaling reveal coherent system-level topologies that are only partly explained by simple neighborhood wiring, suggesting additional long-range organizing factors (Scannell et al., 1995). This line of inquiry converged on the view that brain networks combine high clustering with short paths, a small world organization that can be quantified by clustering, path length, small worldness, and efficiency, and related to plausible cost-benefit trade-offs (Bassett & Bullmore, 2006). Reviews then systematized how to construct graphs from imaging data and how measures of segregation, integration, centrality, motifs, assortativity, and robustness jointly characterize architecture, while emphasizing that node definitions, edge weighting, thresholding, and null models strongly shape inference (Bullmore & Sporns, 2009; Rubinov & Sporns, 2010).

Building on this foundation, learning based approaches seek predictive models that preserve neurobiological interpretability. BrainGNN encodes region identity with region-of-interest-aware convolutions and performs region selection pooling to yield subject-level and cohort-level biomarkers that align with prior findings (Li et al., 2021). BrainNNExplainer complements this with a group-level explainer that learns a sparse global edge mask, improving diagnosis while revealing disorder-specific subgraphs (Cui et al., 2021). Dynamic representations extend beyond static functional connectivity: STAGIN learns spatio-temporal graph embeddings with attention across windows of connectivity (Kim et al., 2021); IBGNN proposes an interpretable backbone that couples edge weight aware message passing with a shared mask for disorder specific biomarkers (Cui et al., 2022); TBDS learns sparse directed acyclic graphs from BOLD time series and couples them to signed edge graph neural networks (Yu et al., 2022); and DynDepNet replaces fixed connectivity with time varying dependencies that are optimized jointly with a recurrent classifier (Campbell et al., 2023). Parallel developments adjust data geometry and architectural bias: R Mixup interpolates correlation and adjacency matrices along log Euclidean geodesics on the symmetric positive definite manifold to stabilize training and improve generalization (Kan et al., 2023); BrainRGIN adapts graph isomorphism style aggregation, region aware pooling, and attention based readouts to predict individual differences with interpretable subnetworks (Thapaliya et al., 2025); and BioBGT encodes hub roles and modular structure within a transformer by combining diffusion based node importance with module aware self attention (Peng et al., 2025).

Recent benchmarking advises caution when applying message aggregation to functional connectomes. Broad comparisons indicate that aggregation can underperform strong non-graph baselines as graph density increases, while hybrids that separate global vectorized patterns from localized

graph structure provide competitive accuracy and clearer interpretation (Han et al., 2025). In parallel, work on hyperbolic embeddings of multilayer networks, including formulations in the Lorentz model, points toward latent geometric explanations, although current treatments remain focused on static graphs and often rely on stochastic block models to encode topology (Guillemaud et al., 2025).

## D  THEORETICAL PROPOSITIONS & PROOFS

**Setup.** Fix an integer $L \geq 2$. Let the common vertex set be

$$V \;=\; I \mathbin{\dot\cup} H_1 \mathbin{\dot\cup} \cdots \mathbin{\dot\cup} H_{L-1} \mathbin{\dot\cup} O \mathbin{\dot\cup} B,$$

a disjoint union of inputs $I$, hidden layers $H_\ell$ ($\ell = 1, \ldots, L-1$), outputs $O$, and bias nodes $B = \{b^{(1)}, \ldots, b^{(L)}\}$. Let $T$ be a nonempty index set of time steps (epochs). For each $t \in T$, the parameter snapshot is a directed (or undirected as well), edge-labeled graph

$$G_t \;=\; (V, E_t, \theta_t),$$

where $E_t \subseteq V \times V$ is the edge set and $\theta_t : E_t \to \mathbb{R}$ assigns a real weight to each edge present at time $t$. The family $(G_t)_{t \in T}$ is called a *trace*.

**Definition 1.** *A family $(\varphi_t)_{t \in T}$ with $\varphi_t : V \to V$ bijective for each $t$ is a snapshot automorphism family for the trace $(G_t)_{t \in T}$ if, for every $t \in T$,*

1. *Edge preservation at time $t$:*

$$(u, v) \in E_t \quad \Longleftrightarrow \quad (\varphi_t(u), \varphi_t(v)) \in E_t \quad \text{for all } u, v \in V.$$

2. *Node label preservation at time $t$: input nodes, output nodes, and bias nodes are fixed points; that is,*

$$\varphi_t(i) = i \ \forall\, i \in I, \qquad \varphi_t(o) = o \ \forall\, o \in O, \qquad \varphi_t(b) = b \ \forall\, b \in B.$$

*(Lim et al., 2023)*

**Definition 2.** *Let $\mathcal{T}$ denote the set of all traces on the fixed vertex set $V$. An encoder is a map*

$$F : \mathcal{T} \to Z$$

*into a representation space $Z$ equipped with a (left) group action $\rho$ of the product group $\prod_{t \in T} \mathrm{Aut}(G_t)$ on $Z$. The encoder $F$ is called* snapshot-wise permutation equivariant *if, for every trace $(G_t)_{t \in T}$ and every snapshot automorphism family $(\varphi_t)_{t \in T}$ with $\varphi_t \in \mathrm{Aut}(G_t)$,*

$$F\big((\varphi_t \cdot G_t)_{t \in T}\big) \;=\; \rho\big((\varphi_t)_{t \in T}\big)\, F\big((G_t)_{t \in T}\big).$$

*When $Z$ carries an index structure inherited from $V$ at each $t$ (e.g., per-node or per-edge features over time), $\rho\big((\varphi_t)_t\big)$ is the natural permutation of those indices, acting componentwise in $t$. If, in addition, a readout $P : Z \to Y$ is permutation-invariant with respect to $\rho$, then $P \circ F$ is invariant under all snapshot automorphism families (Bronstein et al., 2021).*

**Proposition 1.** *For any epoch $t \in T$, a per-snapshot permutation $\varphi_t : V \to V$ belongs to the snapshot automorphism family at time $t$ if and only if it fixes all input, output, and bias nodes and permutes only hidden neurons within their own layer; equivalently, By membership, this is exactly the class of permutations that preserve the time-$t$ edges:*

$$(u, v) \in E_t \iff (\varphi_t(u), \varphi_t(v)) \in E_t \ \text{for all } u, v \in V.$$

*Proof.* Fix $t \in T$.

($\Rightarrow$) If $\varphi_t$ belongs to the snapshot automorphism family at time $t$, then by node–label preservation, it fixes $I$, $O$, and $B$. Moreover, as a graph isomorphism of $G_t$ it preserves lengths of directed paths. In an MLP DAG with edges only between adjacent layers, the layer index $\ell(h)$ of a hidden node $h$ equals the maximum length of a directed path from an input to $h$. Since permutations render this maximal path length unchanged under $\varphi_t$ i.e., $\ell(\varphi_t(h)) = \ell(h)$ which implies that $\varphi_t$ maps each hidden layer to itself.

($\Leftarrow$) Conversely, if $\varphi_t$ fixes $I$, $O$, and $B$, maps each hidden layer to itself, and preserves $E_t$, then it satisfies the snapshot automorphism conditions at time $t$ by definition. $\qquad\square$

**Proposition 2.** *For any trace $(G_t)_{t \in T}$ and any family of per-snapshot permutations $(\psi_t)_{t \in T}$ with $\psi_t \in \mathrm{Aut}(G_t)$ at each $t$, the GTH-GMN encoder satisfies:*

1. *__Snapshot equivariance:__ at every time $t$, relabeling the snapshot by $\psi_t$ relabels the time-$t$ output in the same way*

2. *__Previous-step invariance:__ the temporal transition from step $t - 1$ to $t$ is invariant to any per-snapshot permutation applied at $t - 1$;*

*Consequently, the full encoder is equivariant to $\phi_t$ at the current step $t$, while the update from $t - 1$ to $t$ is unaffected by any admissible relabeling at $t - 1$.*

*Proof.* **1.** At snapshot $t$, the parameter graph is $G_t = (V, E_t, \theta_t)$. Each node $u \in V$ carries raw node features

$$x_t(u) = \big[\, \ell(u),\ \tau(u),\ \tilde{s}_{\mathrm{abs}}(u),\ \tilde{s}_{\mathrm{sgn}}(u) \,\big],$$

where $\ell(u)$ is the layer index, $\tau(u) \in \{\mathrm{in}, \mathrm{hid}, \mathrm{out}, \mathrm{bias}\}$ the node type, and $\tilde{s}_{\mathrm{abs}}, \tilde{s}_{\mathrm{sgn}}$ are standardized incident-weight aggregations (as defined in Sec. 3.2) computed *at node $u$* from the coalesced signed weights at time $t$. Each edge $(u, v) \in E_t$ carries raw edge features

$$e_t(u, v) = \big[\, \mathtt{is\_bias}(u, v),\ \Delta \ell(u, v) \,\big],$$

encoding whether the edge originates at a bias node and the layer difference $\Delta \ell(u, v) = \ell(v) - \ell(u)$. By construction, these descriptors are *structural*: they do not depend on arbitrary node indices and are invariant under within-layer permutations that fix inputs/outputs/biases.

*Group action on features.* Let $\psi_t \in \mathrm{Aut}(G_t)$ be any per-snapshot permutation (automorphism) at time $t$. Its action on the node and edge features is the natural reindexing:

$$\big[\psi_t(x_t)\big]_{\psi_t(u)} = x_t(u), \qquad \big[\psi_t(e_t)\big]_{\psi_t(u),\, \psi_t(v)} = e_t(u, v).$$

Equivalently, if $P_{\psi_t} \in \{0, 1\}^{|V| \times |V|}$ is the permutation matrix induced by $\psi_t$ on nodes and $\mathcal{P}_{\psi_t} := P_{\psi_t} \otimes P_{\psi_t}$ the induced action on ordered node pairs, then in matrix/tensor form

$$\psi_t(x_t) = P_{\psi_t} x_t, \qquad \psi_t(e_t) = \mathcal{P}_{\psi_t} \cdot e_t,$$

(where the second equality is understood as reindexing the first two modes of the edge-feature tensor). The values of $\ell, \tau, \tilde{s}_{\mathrm{abs}}, \tilde{s}_{\mathrm{sgn}}$ and $[\mathtt{is\_bias}, \Delta \ell]$ are preserved because $\psi_t$ fixes node labels for $I \cup O \cup B$ and only permutes hidden nodes within layers; hence $\ell$ and $\tau$ are unchanged, and the incident-weight aggregations at $u$ move to $\psi_t(u)$ without alteration of their numeric value.

*Neighborhoods and reindexing.* Write $\mathcal{N}_t(u) := \{\, v \in V : (u, v) \in E_t \,\}$ for the (directed or undirected) out-neighborhood at time $t$. Automorphism implies $\mathcal{N}_t(\psi_t(u)) = \psi_t(\mathcal{N}_t(u))$, so any neighbor-wise aggregation satisfies

$$\sum_{v \in \mathcal{N}_t(\psi_t(u))} f\big(\psi_t(v)\big) = \sum_{v \in \psi_t(\mathcal{N}_t(u))} f(v) = \sum_{v \in \mathcal{N}_t(u)} f\big(\psi_t(v)\big),$$

and likewise for means and other symmetric reductions.

*Equivariance of the hyperbolic graph attention block.* Let $\mathrm{HGAT}_t$ denote the per-snapshot hyperbolic attention block at time $t$, with shared parameters $W_t$ supplied by the temporal kernel. Its forward map can be abstracted as

$$(X_t, z_t) = \mathrm{HGAT}_t\big(x_t, e_t;\, W_t\big),$$

where $X_t \in \mathbb{R}^{|V| \times d_h}$ are tangent embeddings and $z_t \in (\mathbb{B}_c^{d_z})^{|V|}$ are hyperbolic embeddings. Internally, the block computes nodewise (tangent) projections, hyperbolic queries/keys/values via shared Möbius-linear maps, attention logits that depend *only* on index-symmetric quantities (e.g., $d_c(z_t(u), z_t(v))$ and $e_t(u, v)$), neighbor-wise softmax normalizations, and Möbius-weighted sums over $\mathcal{N}_t(u)$. All these operations commute with the reindexing induced by $\psi_t$. Concretely, letting

$$\ell_{uv} = g\big(d_c(\phi_q(z_t(u)), \phi_k(z_t(v))),\ e_t(u, v),\ X_t(u),\ X_t(v)\big), \qquad \alpha_{uv} = \mathrm{softmax}_{v \in \mathcal{N}_t(u)} \ell_{uv},$$

one has $\ell_{\psi_t(u)\psi_t(v)} = \ell_{uv}$ and hence $\alpha_{\psi_t(u)\psi_t(v)} = \alpha_{uv}$; messages and updates therefore reindex accordingly. It follows that there exists the induced permutation $P_{\psi_t}$ such that

$$\mathrm{HGAT}_t\big(\psi_t(x_t),\, \psi_t(e_t)\,;\, W_t\big) \;=\; P_{\psi_t}\,\mathrm{HGAT}_t\big(\,x_t,\, e_t\,;\, W_t\big),$$

i.e., the per-snapshot block is permutation equivariant in our setting on *both* the tangent and hyperbolic branches.

**2.** Fix $t \in T$ and let $\psi \in \mathrm{Aut}(G_{t-1})$ act on node indices with permutation matrix $P_\psi$. Let $X_{t-1} \in \mathbb{R}^{|V| \times d_x}$ and $Z_{t-1} \in (\mathbb{B}_c^{d_z})^{|V|}$ denote the *tangent* and *hyperbolic* node embeddings produced at time $t-1$. The action of $\psi$ on embeddings is

$$(P_\psi X_{t-1})(u) \;=\; X_{t-1}\big(\psi^{-1}(u)\big), \qquad (P_\psi z_{t-1})(u) \;=\; z_{t-1}\big(\psi^{-1}(u)\big).$$

*Euclidean (tangent) pooling is permutation-invariant.* Define the pooled tangent vector(s)

$$p^{(t-1)} \;=\; \frac{1}{|V|} \sum_{u \in V} X_{t-1}(u) \quad \text{or layerwise} \quad p_\ell^{(t-1)} \;=\; \frac{1}{|H_\ell|} \sum_{u \in H_\ell} X_{t-1}(u).$$

Because $\psi$ fixes $I \cup O \cup B$ and only permutes hidden nodes within layers, each $H_\ell$ is preserved. Summation is order-agnostic; hence

$$\frac{1}{|V|} \sum_{u \in V} (P_\psi X_{t-1})(u) \;=\; p^{(t-1)}, \qquad \frac{1}{|H_\ell|} \sum_{u \in H_\ell} (P_\psi X_{t-1})(u) \;=\; p_\ell^{(t-1)} \quad \forall \ell.$$

*Einstein gyromidpoint aggregation (Klein computation) and permutation invariance.* Since linear averages are not geometrically compatible on the Poincaré ball, we aggregate values using the Einstein (gyro) midpoint aggregation, which we restate here. The computation is performed in the Klein model, where weighted barycenters admit a closed form. Let $\kappa : \mathbb{B}_c^d \to \mathbb{K}^d$ be the Poincaré-to-Klein map,

$$\kappa\big(z^{(t)}\big) \;=\; \frac{2\sqrt{c}\, z^{(t)}}{1 + c\|z^{(t)}\|^2}, \qquad z^{(t)} \in \mathbb{B}_c^d, \tag{19}$$

and $\kappa^{-1} : \mathbb{K}^d \to \mathbb{B}_c^d$ its inverse,

$$\kappa^{-1}\big(y^{(t)}\big) \;=\; \frac{1}{\sqrt{c}} \frac{y^{(t)}}{1 + \sqrt{1 - \|y^{(t)}\|^2}}, \qquad y^{(t)} \in \mathbb{K}^d. \tag{20}$$

We also denote by $\gamma(y^{(t)}) = 1/\sqrt{1 - \|y^{(t)}\|^2}$ the Lorentz factor. For node $v$ and head $r$ at time $t$, the Klein-space barycenter of the incoming neighborhood is

$$\kappa\big(m_v^{(r,t)}\big) \;=\; \frac{\displaystyle\sum_{u:\,(u \to v) \in E_t} \widetilde{\alpha}_{u \to v}^{(r,t)}\, \gamma\big(\kappa(v_u^{(r,t)})\big)\, \kappa\big(v_u^{(r,t)}\big)}{\displaystyle\sum_{u:\,(u \to v) \in E_t} \widetilde{\alpha}_{u \to v}^{(r,t)}\, \gamma\big(\kappa(v_u^{(r,t)})\big)}. \tag{21}$$

We then map back to the Poincaré ball and linearize at the origin for the residual path:

$$m_v^{(r,t)} \;=\; \kappa^{-1}\Big(\kappa\big(m_v^{(r,t)}\big)\Big) \in \mathbb{B}_c^d, \qquad \widehat{m}_v^{(r,t)} \;=\; \log_0\Big(m_v^{(r,t)}\Big) \in T_0\mathbb{B}_c^d, \tag{22}$$

and average across heads,

$$M_v^{(t)} \;=\; \frac{1}{H} \sum_{r=1}^{H} \widehat{m}_v^{(r,t)}. \tag{23}$$

*Per-snapshot permutation equivariance of the Einstein aggregator.* Let $\psi_t \in \mathrm{Aut}(G_t)$ be any per-snapshot permutation. Automorphism implies $\{u : (u \to v) \in E_t\}$ is sent to $\{u' : (u' \to \psi_t(v)) \in E_t\}$ with $u' = \psi_t(u)$, and snapshot equivariance of attention yields $\widetilde{\alpha}_{\psi_t(u) \to \psi_t(v)}^{(r,t)} = \widetilde{\alpha}_{u \to v}^{(r,t)}$. Applying equation 21 at node $\psi_t(v)$ and reindexing the sums by $u' = \psi_t(u)$ gives

$$\kappa\big(m_{\psi_t(v)}^{(r,t)}\big) = \frac{\sum_u \widetilde{\alpha}_{u \to v}^{(r,t)}\, \gamma\big(\kappa(v_{\psi_t(u)}^{(r,t)})\big)\, \kappa\big(v_{\psi_t(u)}^{(r,t)}\big)}{\sum_u \widetilde{\alpha}_{u \to v}^{(r,t)}\, \gamma\big(\kappa(v_{\psi_t(u)}^{(r,t)})\big)}.$$

Thus the Einstein weighted midpoint is *symmetric* in the multiset of weighted Klein vectors and therefore commutes with the reindexing induced by $\psi_t$; mapping back by $\kappa^{-1}$ and $\log_0$ preserves this equivariance. Consequently, $M^{(t)}_{\psi_t(v)}$ is exactly the $P_{\psi_t}$-reindexing of $M^{(t)}_v$.

At time $t-1$, let the hyperbolic node embeddings be $Z_{t-1} = \{z^{(t-1)}_u\}_{u \in V} \subset \mathbb{B}^d_c$ and the tangent embeddings be $X_{t-1} = \{X_{t-1}(u)\}_{u \in V} \subset \mathbb{R}^{d_x}$. We summarize these for the temporal kernel using (i) Euclidean means in the tangent space and (ii) Einstein gyromidpoints in Klein space, consistent with Eqs. equation 19–equation 23.

Define the global (or layerwise) tangent summaries

$$p^{(t-1)} = \frac{1}{|V|} \sum_{u \in V} X_{t-1}(u), \qquad p^{(t-1)}_\ell = \frac{1}{|H_\ell|} \sum_{u \in H_\ell} X_{t-1}(u).$$

These are permutation-invariant because they are simple averages over sets preserved by automorphisms.

Let $\kappa$ and $\kappa^{-1}$ be the Poincaré–Klein maps in equation 19–equation 20, and $\gamma(y) = 1/\sqrt{1 - \|y\|^2}$. With uniform weights, the *global* Klein barycenter is

$$\kappa(\bar{z}_{t-1}) = \frac{\sum\limits_{u \in V} \gamma\big(\kappa(z^{(t-1)}_u)\big)\, \kappa\big(z^{(t-1)}_u\big)}{\sum\limits_{u \in V} \gamma\big(\kappa(z^{(t-1)}_u)\big)}, \qquad \bar{z}_{t-1} = \kappa^{-1}\big(\kappa(\bar{z}_{t-1})\big).$$

For layerwise pooling, restrict the sums to $u \in H_\ell$ to obtain $\bar{z}^{(\ell)}_{t-1}$. For compatibility with the temporal GRU/MLP operating in the tangent space, we finally use

$$\widehat{\bar{z}}_{t-1} = \log_0\big(\bar{z}_{t-1}\big), \qquad \widehat{\bar{z}}^{(\ell)}_{t-1} = \log_0\big(\bar{z}^{(\ell)}_{t-1}\big).$$

Let $\psi \in \mathrm{Aut}(G_{t-1})$ with induced reindexing $(P_\psi z^{(t-1)})_u = z^{(t-1)}_{\psi^{-1}(u)}$. In the global Einstein midpoint, the numerator and denominator are sums over the multiset $\big\{\gamma(\kappa(z^{(t-1)}_u))\, \kappa(z^{(t-1)}_u)\big\}_{u \in V}$, which are merely *reordered* by $\psi$. Hence

$$\kappa\big(\bar{z}_{t-1}(P_\psi z^{(t-1)})\big) = \kappa\big(\bar{z}_{t-1}(z^{(t-1)})\big), \quad \implies \quad \bar{z}_{t-1}(P_\psi z^{(t-1)}) = \bar{z}_{t-1}(z^{(t-1)}),$$

and applying $\log_0$ preserves equality: $\widehat{\bar{z}}_{t-1}(P_\psi z^{(t-1)}) = \widehat{\bar{z}}_{t-1}(z^{(t-1)})$. The same argument holds layerwise because $\psi$ maps each $H_\ell$ to itself. Therefore, the hyperbolic pooled summaries $\widehat{\bar{z}}_{t-1}$ and $\{\widehat{\bar{z}}^{(\ell)}_{t-1}\}_\ell$ are invariant to any admissible permutation at time $t-1$.

Let $X_{t-1}$ be the *tangent* embeddings at $t-1$ ($\log_0(z^{t-1})$), and let $p^{(t-1)}$ (or $\{p^{(t-1)}_\ell\}$) denote their mean pool(s). Since $p^{(t-1)}$ is an ordinary Euclidean average and layer sets are preserved by automorphisms, $p^{(t-1)}$ is permutation invariant. The Evolve-GCN-H style kernel then consumes only node-agnostic summaries,

$$(W_t, u_t) = \phi_{\mathrm{MLP}}\big(\mathrm{GRU}\big(p^{(t-1)}, u_{t-1}\big)\big) \quad \text{(or with layerwise pools)},$$

where $u_{t-1}$ is learned hidden temporal vector, which proves invariance of $(W_t, u_t)$ to any per-snapshot permutation at time $t-1$. This completes the previous-step invariance argument, i.e, the temporal kernel renders the module invariant to permutations in the previous time step (Bronstein et al., 2021). □

# E    EXPERIMENT DETAILS

## E.1    PROBLEM FORMULATION

We study dynamic parameter graphs extracted from the training of a base multilayer perceptron (MLP). A dynamic parameter graph is a sequence $\mathcal{G} = \{G_t = (V, E_t, W_t)\}_{t=1}^T$ of $T$ snapshots,

where $V$ is the fixed node set (inputs, hidden units, and biases), $E_t$ is the set of undirected edges at time $t$, and $W_t = \{w_{uv}^{\star,(t)} : (u,v) \in E_t\}$ the associated signed weights. The goal is to learn node embeddings $\{z_i^{(t)} \in \mathbb{B}_c^d : i \in V,\ t = 1, \ldots, T\}$ in the $d$–dimensional Poincaré ball such that:

(i) a Fermi–Dirac decoder $\psi^{(t)}(u,v)$ predicts adjacency labels, (ii) a signed regressor $\widehat{w}_{uv}^{(t)}$ estimates $w_{uv}^{\star,(t)}$, and (iii) trajectories $\{z_i^{(t)}\}_t$ evolve smoothly across time.

Supervision is provided on positive edges, with negative samples generated adaptively, placing the task in the semi–supervised, temporal graph embedding setting.

### E.2 Additional Details: Classification of Images via INR Traces

**Base INR architecture and training.** For each image $x \in \{$MNIST, Fashion-MNIST$\}$ we train a shallow implicit neural representation (INR) to fit pixel intensities as a continuous function of coordinates. The INR has three fully connected layers of width 32 with $\tanh$ activations, mapping $u \in [0,1]^2$ (pixel coordinates) to grayscale value $x(u)$. Weights are initialized with Xavier initialization and optimized using Adam (learning rate $5 \times 10^{-4}$) for up to 100 epochs, with early stopping if the PSNR of the reconstruction exceeds 40. This produces a sequence of checkpoints $\{\theta_{i,t}\}_{t=1}^{T_i}$ for image $i$, where $T_i \in [80, 100]$ depending on early stopping.

**Parameter-graph construction.** At each checkpoint $t$, the INR parameters $\theta_{i,t}$ are transformed into a signed parameter graph $G_{i,t}$:

- nodes correspond to input, hidden, and bias units;
- edges connect units across successive layers and from biases to their layer, labeled by the signed weight;
- node features include layer index, node type, and $z$–scored sums of incident weights (absolute and signed);
- edge features indicate whether the edge originates from a bias node and the layer jump $\Delta\ell$.

This yields a temporal trace $\{G_{i,t}\}_{t=1}^{T_i}$ per image.

**Trace dataset and splits.** Each image contributes one temporal trace. Dataset splits are made strictly at the *image* level to prevent leakage across time. We use 45,000 images for training, 5,000 for validation, and 10,000 for testing. All checkpoints from an image remain in the same split. Each trace contains between 80 and 100 checkpoints, ensuring sufficient temporal depth.

**Meta-network pretraining.** We pretrain our Geometric Temporal Hyperbolic Graph Meta-Network (GTH-GMN) on the training traces. The backbone consists of $L = 4$ *Hyperbolic Graph Attention (HGAT) layers*, each paired with one *EvolveGCN–H temporal kernel*, so that the number of HGAT layers and temporal kernels is always equal. Embeddings live in $\mathbb{B}_c^d$ with dimension $d = 125$ and curvature parameter $c = -1$. The optimization alternates between:

1. Euclidean updates of the attention kernels, regressors, and decoder;
2. Riemannian updates of node embeddings with trust-region control.

We use RAMSGrad with learning rate $10^{-3}$ and trust radius $\tau = 0.1$.

**Losses and curriculum.** The training objective combines:

- Fermi–Dirac link prediction loss (binary cross-entropy);
- signed weight regression (L2 loss on log–magnitude + sign classification);
- quadratic prior on the learned distance exponent;
- temporal smoothness penalty on successive embeddings;
- pairwise logistic ranking loss on semi–hard negative pairs.

A gentle curriculum is applied: the ranking margin increases over epochs, and mining hardness is annealed to prevent early saturation.

**Representation extraction.** During pretraining, edge embeddings within each checkpoint are pooled into a snapshot vector by mean aggregation. For downstream classification, snapshot vectors are pooled temporally (mean across checkpoints in the trace) to yield a fixed $n$–dimensional representation per image.

**Classifier and evaluation.** An MLP classifier ($n \rightarrow 128 \rightarrow 10$, with ReLU) is trained on these representations using cross-entropy loss. The classifier is trained on the training set, tuned on the validation set, and evaluated on the test set. Accuracy on the test set is reported in Table 1.

### E.3 ADDITIONAL DETAILS: PREDICTING DNN GENERALIZATION FROM WEIGHTS

We evaluate on *checkpoint traces* from implicit neural representation (INR) trials trained for CIFAR-10 classification. For each trial $i$ we collect the ordered checkpoints $\{\theta_{i,t}\}_{t=1}^{T_i}$ together with the corresponding per-epoch test accuracy $y_{i,t}$. Using the architecture metadata stored with each trial, we reconstruct a shape-compatible network skeleton and convert every checkpoint to a parameter-graph $G_{i,t}$, where nodes and edges encode their respective features. No image content is used; supervision is solely the scalar accuracy aligned to checkpoints. Each trace is processed by the

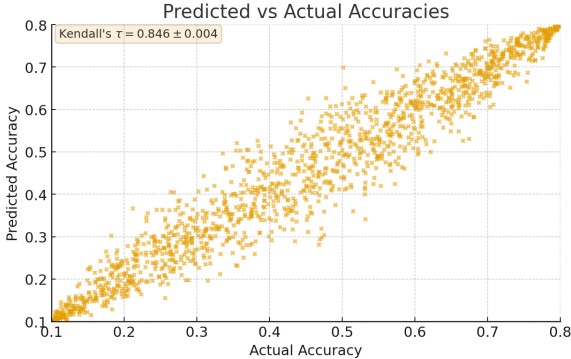

Figure 3: Predicted versus actual generalization performance on CIFAR-10 checkpoint traces. Each point corresponds to a model trial at a given epoch, where predictions are obtained from the learned GTH-GMN embeddings. The strong monotonic trend (Kendall's $\tau = 0.846 \pm 0.004$) indicates that hyperbolic embeddings capture meaningful structure in the parameter dynamics, even if absolute correlation lags *slightly* behind NFN baselines.

encoder introduced in the previous section over fixed-length, stride-rolled windows or full traces (depending on computational constraints). We adopt the *last-step* supervision regime: intermediate steps are used only to evolve the state, and supervision is applied to the final step target. Dataset splits are made strictly by trial (not by checkpoint) to prevent leakage across traces. For all experiments, we embed parameter-graphs into a hyperbolic ball of dimension $d \in [80, 125]$, with four HGAT layers and four EvolveGCN–H temporal kernels of hidden size 128. Hyperbolic embeddings are optimized with the two-phase scheme described in Section 3.4, combining Euclidean updates for kernel parameters with Riemannian gradient steps for node positions. Training employs binary cross-entropy on Fermi–Dirac link logits, signed weight regression, a prior on the learned exponent, temporal smoothness penalties, and a ranking curriculum.

We report Kendall's $\tau$ for ranking fidelity, following Zhou et al. (2023a); Navon et al. (2024); Lim et al. (2023). Results are shown in Table 2 and illustrated qualitatively in Figure 3. Our method GTH-GMN achieves $\tau = 0.846 \pm 0.004$, lower than NFN baselines that learn end–to–end from raw weight tensors. This gap is expected, since NFNs exploit stronger task-aligned inductive biases while our approach compresses parameters into permutation-invariant graphs and trains with geometric proxy losses under last-step supervision, inevitably discarding certain accuracy-correlated cues. Nonetheless, GTH-GMN offers a complementary benefit by jointly modeling link structure,

signed weights, and temporal evolution in hyperbolic space, producing embeddings that are *robust, interpretable, and faithful to training dynamics*.

### E.4 ADDITIONAL DETAILS: PREDICTING SINE WAVE FREQUENCY

In addition to supervised objectives, we employ a SimCLR-style (Chen et al., 2020) contrastive regularizer adapted to temporal checkpoint traces. This setup treats the optimization trajectory of a neural network analogously to an image in a classification task, operating in "parameter-time space" rather than pixel space.

For each trial, two correlated *views* of the same checkpoint subsequence are generated by combining:

(i) **Gaussian perturbations** of node and edge features, and

(ii) **Random temporal masking**, where a subset of checkpoints in the subsequence is dropped with probability $p = 0.5$.

The encoder processes both views to produce pooled representations, which are learned using a contrastive loss (Chen et al., 2020). Positive pairs correspond to augmented views of the same checkpoint subsequence (representing the same underlying function dynamics), while negative pairs are drawn from other trials. This auxiliary objective ensures that the learned geometric representation captures the intrinsic signature of the optimization trajectory.

### E.5 ADDITIONAL DETAILS: WEIGHTS VS DISTANCE RELATIONSHIPS

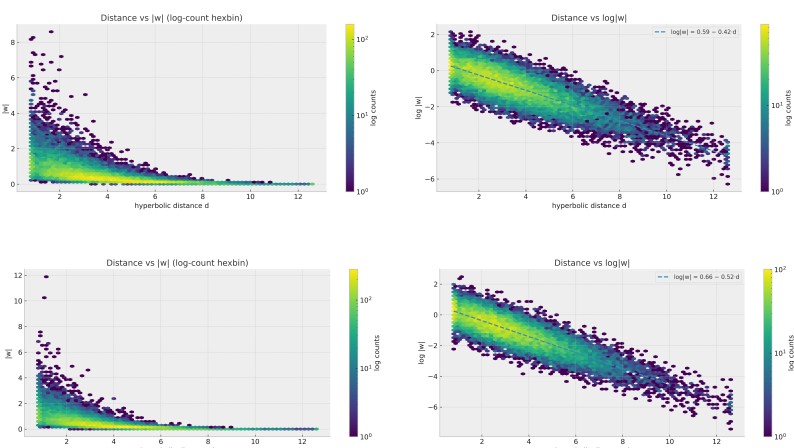

Figure 4: Relationship between hyperbolic distance $d$ and edge weights $w$ on CIFAR-10 INR traces, shown as log-count hexbins. (Top Left, Bottom Left) Distance versus absolute weight $|w|$ (at $t \in T$), highlighting the concentration of large weights at small distances and the rapid decay at larger distances. (Top Right, Bottom Right) Distance versus log-magnitude $\log|w|$ (at $t' \in T$), which reveals an approximately linear inverse trend, consistent with a power–law dependence of weight magnitudes on hyperbolic distance. Together, these plots illustrate that the hyperbolic geometry learned by GTH-GMN aligns distances with functional weight strengths, capturing both edge existence and graded magnitude in a geometrically coherent manner.

To probe the strength of the geometric relationship captured by GTH-GMN, we visualize the alignment between learned hyperbolic distances and underlying edge weights on the CIFAR-10 INR traces. Figure 4 reports hexbinned joint distributions of hyperbolic distance $d$ against both $|w|$ and $\log|w|$. *The results reveal a clear inverse correlation: large weights tend to concentrate at smaller hyperbolic distances, while edges at larger distances exhibit systematically lower magnitudes*. On the logarithmic scale, the relationship is approximately linear, consistent with the power–law parameterization employed in our signed edge–weight regressor. These findings support the claim that temporal hyperbolic embeddings faithfully encode functional structure: the geometry reflects not only the presence of connections but also their graded strengths across training, thereby grounding the link between parameter values and hyperbolic distance.

## F    HYPERPARAMETER INVESTIGATION

Table 5: Architecture and data settings for our Experiments.

| Component | Hyperparameter | Range / Value |
|---|---|---|
| Manifold | Curvature $|c|$ | fixed to 1.0 |
| | Embedding dimension $d$ | $80 - 125$ |
| HGAT (per layer) | # layers $L$ | 4 |
| | # heads $H$ | $4 - 8$ (tuned) |
| | Head temperature | learned, init $0.5 - 1.0$ |
| | Residual gate | learned scalar $\in (0,1)$ |
| | Geodesic trust radius $\tau_\ell$ | $0.05 - 0.15$ |
| EvolveGCN–H | # temporal kernels | 4 (equal to HGAT layers) |
| | GRU hidden size | $64 - 128$ |
| | Input to GRU | mean pool in $T_0$ (prev. snapshot) |
| Graph construction | Node features | $(\ell, t, s^{\text{abs}}, s^{\text{sgn}})$ |
| | Edge features | (is_bias, $\Delta\ell$) |
| Temporal window | Sequence length $K$ | $64 - T$ (full trace, $T \in [80, 100]$) |
| | Stride | $4 - 8$ |
| Contrastive views | Feature noise (Gaussian) | $\sigma = 0.1 - 0.3$ |
| (SimCLR-style) | Temporal masking | drop prob. $p = 0.3 - 0.5$ |
| | Temperature (NT-Xent) | $0.1 - 0.5$ |

**Hyperparameter Investigation.**    Tables 5–6 summarize the settings used in our studies. Architecture choices fix the manifold curvature at $|c|$=1.0 and sweep embedding dimension $d \in [80, 125]$; HGAT uses $L$=4 layers with $H \in [4, 8]$, learned head temperatures, a learned residual gate, and per–layer geodesic trust radii $\tau_\ell \in [0.05, 0.15]$. EvolveGCN–H mirrors the HGAT depth with GRU size 64–128, while the parameter graph encodes $(\ell, t, s^{\text{abs}}, s^{\text{sgn}})$ at nodes and (is_bias, $\Delta\ell$) at edges. Temporal windows use $K \in [64, T]$ with stride 4–8, and SimCLR-style augmentations vary feature noise, temporal masking, and NT-Xent temperature. Optimization (Table 6) employs Adam for Euclidean parameters with lr $\in [1 \times 10^{-3}, 3 \times 10^{-3}]$, weight decay $10^{-6}$–$10^{-4}$, StepLR, and 30–50 epochs; node positions use Riemannian AMSGrad (RAMSGrad) with matched learning rates, trust radius $\tau \in [0.05, 0.15]$, $(\beta_1, \beta_2) = (0.9, 0.99$–$0.999)$, and conformal preconditioning. The FD decoder learns radius $R$ and temperature $T$ from initialized ranges, negatives mix same-layer and uniform samples, and losses combine FD BCE, signed weight reconstruction, sign BCE, exponent prior, temporal smoothness, and pairwise ranking with reported weights; the same losses are reused in a $z$–only pass with $X^{(t)}$ detached. For throughput, we cap positive edges and chunk FD/distance computations as listed.

## G    ALGORITHMS

**Overview of Algorithms.**    Algorithm 2 converts a single MLP checkpoint into a permutation–invariant parameter graph by instantiating input, neuron, and bias nodes; coalescing directed parameter incidences into undirected edges with context and signed labels; and standardizing node-wise strength features to form $G_t = (X_t, E, W_t)$. Algorithm 1 then updates node positions intrinsically on the Poincaré ball: Euclidean gradients are rescaled to Riemannian ones, Adam–style moments are accumulated, a geodesic trust region clips the tangent step, the exponential map applies the update with a safety projection, and first moments are parallel transported. The two procedures are independent and can be composed sequentially within each training step.

Table 6: Optimization and loss settings for INR-trace image classification. Ranges correspond to grid or random search over hyperparameters.

| Component | Hyperparameter | Range / Value |
|---|---|---|
| Euclidean params | Optimizer / LR | Adam, $\text{lr} = 1 \times 10^{-3} - 3 \times 10^{-3}$ |
| (kernels, heads) | Weight decay | $10^{-6} - 10^{-4}$ |
| | LR schedule | StepLR (step $= 10 - 20$, $\gamma = 0.6 - 0.8$) |
| | Epochs | $30 - 50$ |
| Manifold params | Optimizer | RAMSGrad (Riemannian) |
| (node positions $z$) | LR / trust radius | $\text{lr} = 1 \times 10^{-3} - 3 \times 10^{-3}$, $\tau = 0.05 - 0.15$ |
| | $(\beta_1, \beta_2)$ | $(0.9, 0.99 - 0.999)$ |
| | Conformal precond. | divide by $\lambda_x^2$ |
| Link decoder | Fermi–Dirac radius $R$ | init $1.5 - 2.5$ (learned) |
| | Temperature $T$ | init $0.3 - 0.7$ (learned) |
| Negatives | Neg. multiplier | $3 - 10\times$ positives |
| | Same-layer ratio | $0.3 - 0.7$ |
| Losses (Euclid pass) | FD BCE (links) | weight ramp to 1.0 |
| | Signed weight recon | $\lambda_{\text{wrec}} = 0.2 - 0.6$ |
| | Sign BCE | $\lambda_{\text{sign}} = 0.2 - 0.4$ |
| | Exponent prior | $\lambda_\alpha = 10^{-3} - 10^{-2}$ |
| | Temporal smoothness | $\beta_{\text{smooth}} = 0.2 - 0.6$ |
| | Ranking (pairs) | $\lambda_{\text{rank}} = 0.5 - 1.0$, margin $\gamma = 0.2 - 0.5$ |
| Losses ($z$-only pass) | Same terms | as Euclid pass; $X^{(t)}$ detached |
| Practical caps | Pos. edges per step | $10{,}000 - 20{,}000$ |
| | FD / distance chunks | $20{,}000 - 40{,}000$ edges / chunk |

**Optimization Curriculum** We map hyperbolic geometry to link probabilities with a Fermi–Dirac (FD) decoder acting on Poincaré geodesics (Nickel & Kiela, 2017; 2018; Wen et al., 2024):

$$\psi^{(t)}(u, v) = \left(1 + \exp\left(\frac{d_c(z_u^{(t)}, z_v^{(t)}) - R}{T}\right)\right)^{-1}, \tag{24}$$

with learned radius $R$ and temperature $T$. Training uses positives $E_t$ and dynamic negatives from a mixture of same–layer and uniform samplers (mixture capped at 50%). The FD loss is binary cross–entropy,

$$\mathcal{L}_{\text{FD}} = -\mathbb{E}_{(u,v) \in E^+}\left[\log \psi^{(t)}(u, v)\right] - \mathbb{E}_{(u,v) \in E^-}\left[\log(1 - \psi^{(t)}(u, v))\right]. \tag{25}$$

To capture edge magnitudes and signs, we add the signed regressor of Section 3.3. For $(u, v) \in E^+$ with weight $w_{uv}^\star$ and prediction $\widehat{w}_{uv}^{(t)}$ from equation 13 (Goyal et al., 2018; Pareja et al., 2019),

$$\mathcal{L}_{\text{wrec}} = \mathbb{E}_{(u,v) \in E^+}\left[\left\|\widehat{w}_{uv}^{(t)} - w_{uv}^\star\right\|_p\right], \tag{26a}$$

$$\mathcal{L}_{\text{sign}} = -\mathbb{E}_{(u,v) \in E^+}\left[y_{uv} \log \sigma\big(s_{uv}^{(t)}\big) + (1 - y_{uv}) \log\big(1 - \sigma(s_{uv}^{(t)})\big)\right]. \tag{26b}$$

with $p \in \{1, 2\}$ and $y_{uv} = \mathbb{1}[w_{uv}^\star > 0]$. We regularize the exponent via

$$\mathcal{L}_\alpha = \mathbb{E}_{(u,v) \in E^+}\left[(\alpha_{uv} - \alpha_0)^2\right], \tag{27}$$

encourage smooth trajectories (Goyal et al., 2018; Yang et al., 2023),

$$\mathcal{L}_{\text{smooth}} = \frac{1}{|V|} \sum_{i \in V} d_c\big(z_i^{(t)}, z_i^{(t-1)}\big)^2, \tag{28}$$

and enforce distance–magnitude monotonicity with a logistic pairwise rank loss (Nickel & Kiela, 2017),

$$\mathcal{L}_{\text{rank}} = \mathbb{E}_{u,(v^+,v^-)} \log\big(1 + \exp(d_c(z_u^{(t)}, z_{v^+}^{(t)}) - d_c(z_u^{(t)}, z_{v^-}^{(t)}) + \gamma)\big). \tag{29}$$

The total objective is

$$\mathcal{L}^{(t)} = \lambda_{\text{FD}}\mathcal{L}_{\text{FD}} + \lambda_{\text{wrec}}\mathcal{L}_{\text{wrec}} + \lambda_{\text{sign}}\mathcal{L}_{\text{sign}} + \lambda_\alpha\mathcal{L}_\alpha + \lambda_{\text{smooth}}\mathcal{L}_{\text{smooth}} + \lambda_{\text{rank}}\mathcal{L}_{\text{rank}}, \tag{30}$$

with a gentle curriculum: anneal $\gamma$ and harden mining over epochs.

Training runs in two phases: a Euclidean pass updates kernels, regressors, and decoder; a Riemannian pass updates node positions $z^{(t)}$ on the Poincaré ball. With conformal factor $\lambda_z^2 = \frac{2}{1-c\|z\|^2}$, the Riemannian gradient rescales the Euclidean one (Bridson & Haefliger, 1999; Bonnabel, 2013; Nickel & Kiela, 2017; 2018):

$$\nabla_z^{\text{Riem}}\mathcal{L} = \frac{1}{\lambda_z^2}\nabla_z^{\text{Euc}}\mathcal{L}. \tag{31}$$

We maintain first and second moments and take an Adam–style tangent step (Bécigneul & Ganea, 2019; Sakai & Iiduka, 2020):

$$\Delta_i^{(t)} = -\eta\,\frac{m_i^{(t)}}{\sqrt{v_i^{(t)} + \varepsilon}}. \tag{32}$$

A geodesic trust region clips the move, not its Euclidean norm. With

$$z_i^{\text{try},(t)} = \exp_{z_i^{(t)}}(\Delta_i^{(t)}), \qquad d_i^{(t)} = d_c\big(z_i^{(t)}, z_i^{\text{try},(t)}\big), \tag{33}$$

we scale $\Delta_i^{(t)}$ by $\min\{1, \tau/d_i^{(t)}\}$ and accept the intrinsic update

$$z_i^{(t+1)} = \exp_{z_i^{(t)}}(\Delta_i^{(t)}), \tag{34}$$

followed by projection into the ball. First–moment vectors are then parallel transported:

$$m_i^{(t+1)} = \text{PT}_{z_i^{(t)} \to z_i^{(t+1)}}\big(m_i^{(t)}\big). \tag{35}$$

The manifold step optimizes a $z$–only objective mirroring equation 25–equation 29, with $X^{(t)}$ detached. This decouples the hyperbolic graph attention head's stability from embedding motion and enforces controlled geodesic updates near the boundary.

## H   HYPERBOLIC EMBEDDINGS OF INRS

We visualize hyperbolic embeddings of parameter graphs from a 25-layer INR–MLP at several checkpoints 5. Embeddings are learned with 4 HGAT layers and 4 EvolveGCN–H kernels (tangent dim. 128) in $d = 10$ and $d = 25$, then projected to the Poincaré disk via PCA for display. Points are colored by layer, and edges are gray. Early snapshots show mixed layers and long inter-layer edges. As training proceeds, layers separate along angular directions, intra-layer bands tighten, and high-strength nodes drift radially inward, indicating hub formation and growing specialization. The smooth drift of clusters across checkpoints reflects temporal kernel meta-evolution rather than abrupt reconfiguration. Because these plots are PCA projections from a higher-dimensional ball, exact angles are not preserved; nevertheless, the progressive stratification and the shortening of within-layer geodesics are consistent across panels and align with the learned distance–magnitude coupling.

## I   COMPUTATIONAL COST ANALYSIS

We analyze the time and space complexity of GTH–GMN in terms of the size of the parameter graphs and the temporal window length. Let a parameter graph snapshot $G_t$ have $N$ nodes and $E$ edges. Embeddings live in a $d$ dimensional Poincaré ball, the hyperbolic encoder uses $H$ attention heads and $L$ HGAT layers, the temporal kernel has window length $K$ and GRU hidden size $h$, and each training batch contains $E_{\text{batch}}^+$ positive edges.

---

**Algorithm 1** Intrinsic Riemannian Optimization of Node Positions on the Poincaré Ball

---

**Input:** Snapshots $\{G_t\}_{t=1}^T$; initial positions $\{z_i^{(t)}\}$; moments $m_i^{(t)} = 0, v_i^{(t)} = 0$; learning rate $\eta$; trust radius $\tau$; tolerance $\varepsilon$; curvature $c$

**Output:** Updated positions $\{z_i^{(t)}\}_{t=1}^T$

1: **for** $t = 1$ to $T$ **do**                                              $\triangleright$ after Euclidean parameter update

2:     Detach $X^{(t)}$; compute Euclidean gradient $\nabla_{z_i^{(t)}}^{\mathrm{Euc}}\mathcal{L}$ for losses equation 25–equation 29

3:     Convert to Riemannian gradient with conformal factor $\lambda_z^2 = \frac{2}{1-c\|z\|^2}$:

$$\nabla_{z_i^{(t)}}^{\mathrm{Riem}}\mathcal{L} = \lambda_{z_i^{(t)}}^{-2}\,\nabla_{z_i^{(t)}}^{\mathrm{Euc}}\mathcal{L}$$

4:     Update moments (Adam/AMSGrad) (Bécigneul & Ganea, 2019; Sakai & Iiduka, 2020):

$$m_i^{(t)} \leftarrow \beta_1 m_i^{(t)} + (1 - \beta_1)\nabla_{z_i^{(t)}}^{\mathrm{Riem}}\mathcal{L}$$

$$v_i^{(t)} \leftarrow \beta_2 v_i^{(t)} + (1 - \beta_2)\left(\nabla_{z_i^{(t)}}^{\mathrm{Riem}}\mathcal{L}\right)^{\odot 2}$$

5:     Compute tangent step:

$$\Delta_i = -\eta\,\frac{m_i^{(t)}}{\sqrt{v_i^{(t)}} + \varepsilon}$$

6:     Trial update and geodesic distance:

$$z_i^{\mathrm{try}} = \exp_{z_i^{(t)}}(\Delta_i), \qquad d_i = d_c(z_i^{(t)}, z_i^{\mathrm{try}})$$

7:     Trust scaling: $s_i = \min\{1, \tau/(d_i + \varepsilon)\}; \quad \Delta_i \leftarrow s_i\Delta_i$

8:     Accept update and project inside ball:

$$z_i^{(t+1)} = \exp_{z_i^{(t)}}(\Delta_i), \qquad z_i^{(t+1)} \leftarrow \Pi_{\mathbb{B}_c^d}(z_i^{(t+1)}; \varepsilon)$$

9:     Parallel transport first moment:

$$m_i^{(t+1)} = \mathrm{PT}_{z_i^{(t)} \rightarrow z_i^{(t+1)}}(m_i^{(t)})$$

10: **end for**

---

**Time Complexity of the Hyperbolic Encoder.** For a single HGAT layer on $G_t$, hyperbolic queries, keys and values are obtained by applying the affine maps $\Phi_{W,b}$ at each node and head. Each such map performs one logarithmic map $\log_0$, one matrix–vector product $W\log_0(x)$, one exponential map $\exp_0$, and a bias transport plus exponential at $\Phi_W(x)$, all on $d$ dimensional vectors. The cost per head and node is therefore $O(d^2 + d)$, and aggregated over all nodes and heads this yields

$$\mathrm{Cost}_{\mathrm{QKV}} = O(HNd^2). \tag{36}$$

Attention logits and softmax normalization are computed per edge and head. For each directed edge $(u, v)$, the Poincaré distance $d_c(q_u, k_v)$ requires a constant number of inner products and norms in $\mathbb{R}^d$, so $O(d)$ operations, followed by scalar exponentiation and normalization. Einstein gyromidpoint aggregation in the Klein model combines value vectors from the incoming edges of each node and maps the result back via $\kappa^{-1}$ and $\log_0$, which again contributes $O(d)$ per edge and head plus $O(d)$ per node and head. Summed over all edges, nodes and heads, the attention and aggregation stage costs

$$\mathrm{Cost}_{\mathrm{att}} = O(HEd + HNd). \tag{37}$$

The total cost of a single HGAT layer is therefore

$$\mathrm{Cost}_{\mathrm{HGAT, layer}} = O(HNd^2 + HEd), \tag{38}$$

and a stack of $L$ layers has

$$\mathrm{Cost}_{\mathrm{HGAT}} = O(LHNd^2 + LHEd). \tag{39}$$

---

**Algorithm 2** MLP $\rightarrow$ Parameter Graph (single snapshot at epoch $t$)

---

**Input:** Layer widths $(n_0, \ldots, n_L)$; weights $\{W_\ell\}_{\ell=1}^L$; biases $\{b_\ell\}_{\ell=1}^L$
**Output:** Snapshot $G_t = (X_t, E, W_t)$

1: $V \leftarrow \{ i_1^{(0)}, \ldots, i_{n_0}^{(0)} \}$          $\triangleright$ input nodes
2: **for** $\ell \leftarrow 1$ to $L$ **do**
3:      add bias node $b^{(\ell)}$ to $V$
4:      add neuron nodes $i_1^{(\ell)}, \ldots, i_{n_\ell}^{(\ell)}$ to $V$
5:      set $(\ell(u), \tau(u))$ for all newly added $u$        $\triangleright$ layer index, type
6: **end for**
7: $\mathcal{A}_\rightarrow \leftarrow \emptyset$          $\triangleright$ directed edge bag with context + weight
8: **for** $\ell \leftarrow 1$ to $L$ **do**
9:      **for** $p \leftarrow 1$ to $n_{\ell-1}$ **do**
10:          **for** $q \leftarrow 1$ to $n_\ell$ **do**
11:             append $\big( i_p^{(\ell-1)} \rightarrow i_q^{(\ell)}, \ (\mathrm{is\_bias} = 0, \Delta\ell = 1), \ w \leftarrow W_\ell[q, p] \big)$ to $\mathcal{A}_\rightarrow$
12:          **end for**
13:      **end for**
14:      **for** $q \leftarrow 1$ to $n_\ell$ **do**
15:          append $\big( b^{(\ell)} \rightarrow i_q^{(\ell)}, \ (\mathrm{is\_bias} = 1, \Delta\ell = 0), \ w \leftarrow b_\ell[q] \big)$ to $\mathcal{A}_\rightarrow$
16:      **end for**
17: **end for**
18: $M \leftarrow$ empty map from unordered pairs $\{u, v\}$ to running sums
19: **for all** $(u \rightarrow v, \ e, \ w) \in \mathcal{A}_\rightarrow$ **do**
20:      $k \leftarrow \{u, v\}$
21:      update $M[k].\mathrm{sum\_ctx} \mathrel{+}= e;$     $M[k].\mathrm{sum\_w} \mathrel{+}= w;$     $M[k].\mathrm{cnt} \mathrel{+}= 1$
22: **end for**
23: $E \leftarrow \emptyset, \ \ W_t \leftarrow \emptyset$
24: **for all** $k = \{u, v\}$ in $M$ **do**
25:      $e_{uv} \leftarrow M[k].\mathrm{sum\_ctx}/M[k].\mathrm{cnt}$        $\triangleright$ coalesced context (mean)
26:      $w_{uv}^\star \leftarrow M[k].\mathrm{sum\_w}/M[k].\mathrm{cnt}$        $\triangleright$ coalesced label (mean)
27:      insert $\{u, v\}$ into $E$ with context $e_{uv};$     set $W_t[\{u, v\}] \leftarrow w_{uv}^\star$
28: **end for**
29: **for all** $u \in V:$    $s^{\mathrm{abs}}(u) \leftarrow \sum_{v:\{u,v\} \in E} |W_t[\{u, v\}]|;$    $s^{\mathrm{sgn}}(u) \leftarrow \sum_{v:\{u,v\} \in E} W_t[\{u, v\}]$
30: $\mu_{\mathrm{abs}}, \sigma_{\mathrm{abs}} \leftarrow$ mean/std of $\{s^{\mathrm{abs}}(u) : u \in V\}$
31: $\mu_{\mathrm{sgn}}, \sigma_{\mathrm{sgn}} \leftarrow$ mean/std of $\{s^{\mathrm{sgn}}(u) : u \in V\}$
32: **for all** $u \in V$ **do**
33:      $\widetilde{s}^{\mathrm{abs}}(u) \leftarrow \begin{cases} \dfrac{s^{\mathrm{abs}}(u) - \mu_{\mathrm{abs}}}{\sigma_{\mathrm{abs}}}, & \sigma_{\mathrm{abs}} > 0 \\ 0, & \text{otherwise} \end{cases}$
34:      $\widetilde{s}^{\mathrm{sgn}}(u) \leftarrow \begin{cases} \dfrac{s^{\mathrm{sgn}}(u) - \mu_{\mathrm{sgn}}}{\sigma_{\mathrm{sgn}}}, & \sigma_{\mathrm{sgn}} > 0 \\ 0, & \text{otherwise} \end{cases}$
35: **end for**
36: $X_t \leftarrow \big\{ x_u = [\, \ell(u), \tau(u), \widetilde{s}^{\mathrm{abs}}(u), \widetilde{s}^{\mathrm{sgn}}(u) \,] \ : \ u \in V \big\}$
37: **return** $G_t = (X_t, E, W_t)$

---

In the regimes we study, $E$ is much larger than $N$ and $d$ is moderate, so the edge term $O(LHEd)$ dominates and the cost is effectively linear in the number of edges. The hyperbolic operations (exp, log, distance and parallel transport) are all $O(d)$ vector calculations and do not change the asymptotic order relative to a Euclidean graph attention network with the same $(L, H, d)$.

**Time Complexity of Temporal Kernel Evolution.** Temporal evolution follows the EvolveGCN–H scheme (Pareja et al., 2019), where each HGAT layer maintains a GRU with hidden size $h$ over a window of length $K$. A single GRU step processes a pooled $d$ dimensional summary and costs $O(h^2 + hd)$. For $K$ steps and $L$ layers, the total temporal cost is

$$\mathrm{Cost}_{\mathrm{temp}} = O\big(LK(h^2 + hd)\big). \tag{40}$$

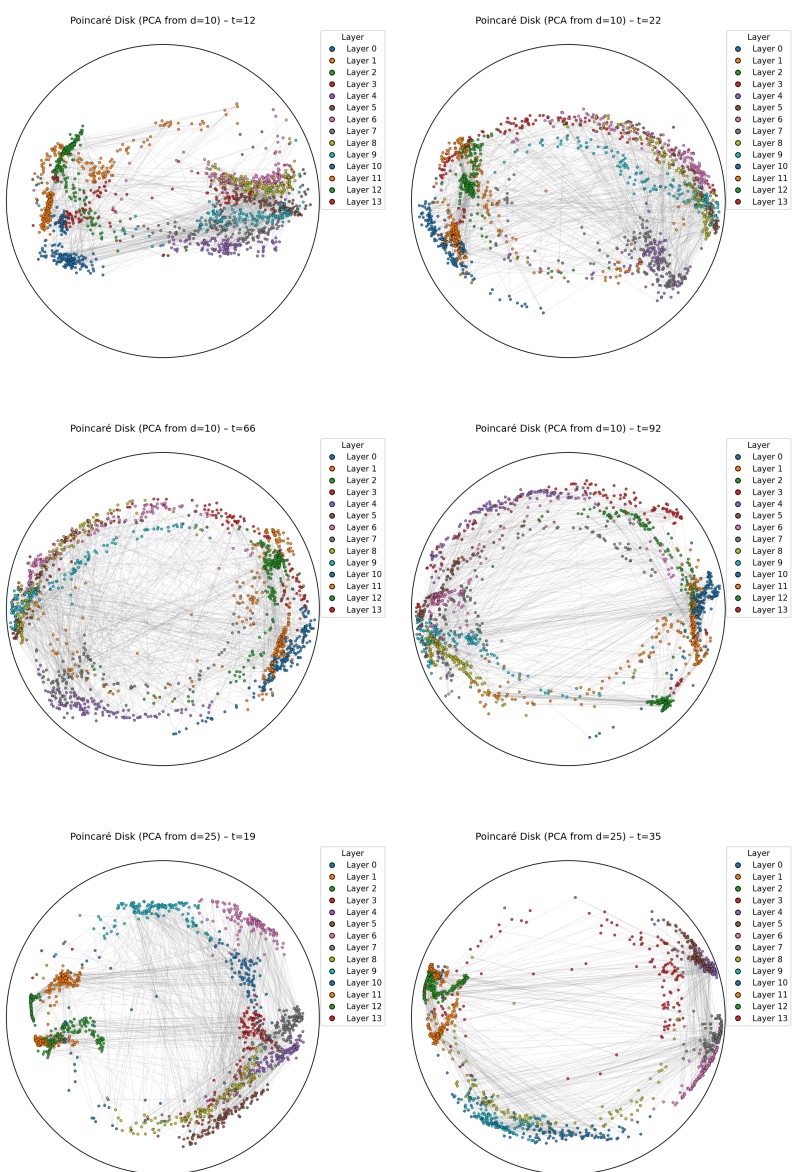

Figure 5: Hyperbolic embeddings of parameter graphs from a 25-layer MLP trained as an INR on CIFAR-10, shown at checkpoints. Embeddings are learned in $d = 10, 25$ dimensions with 4 HGAT layers and 4 EvolveGCN–H kernels (tangent dim. 128) and projected to the Poincaré disk via PCA. Nodes are colored by layer index, with edges in gray, illustrating the progressive reorganization and separation of layers over training.

This term is linear in the window length and independent of $E$. For the configurations used in our experiments, $HEd$ dominates $K(h^2 + hd)$, so temporal coupling is lower order than the HGAT edge processing.

**Time Complexity of the Link Decoder and Signed Regressor.** The Fermi–Dirac decoder and signed weight regressor operate on positive edges and their sampled negatives. For each edge we compute a Poincaré distance $d_c(z_u, z_v)$, a Fermi–Dirac probability, a power law magnitude, and a sign logit from a tangent inner product. Each of these involves a constant number of $d$ dimensional vector operations, so the cost per edge is $O(d)$. If $E_{\text{batch}}^+$ positive edges are present in the batch and

$\kappa$ negatives are sampled per positive, then

$$\text{Cost}_{\text{dec}} = O\big(d(1 + \kappa)E^+_{\text{batch}}\big). \tag{41}$$

In practice $E^+_{\text{batch}}$ is capped (for example at $10^4$ to $2 \times 10^4$), and distance computations are performed in chunks of at most $40\,000$ edges to keep buffers bounded. This term is therefore linear in a controlled edge count and does not affect the overall scaling in $E$ and $K$.

**Two Phase Optimization Overhead.** Each training step consists of a Euclidean pass and a Riemannian refinement pass. In the Euclidean pass we backpropagate through the HGAT stack, temporal kernels and decoders, updating all Euclidean parameters. In the Riemannian pass we update only the node embeddings $z_i$ on the Poincaré ball. This involves rescaling gradients by the conformal factor $\lambda_z^2$, applying RAMSGrad updates in the tangent space, mapping back via the exponential map, and parallel–transporting optimizer moments, all on $d$ dimensional vectors for each node. The cost is therefore

$$\text{Cost}_{\text{Riem}} = O(Nd). \tag{42}$$

We reuse the same sampled edges and losses and do not recompute the attention stack or temporal kernels in this phase, so the two phase scheme introduces a constant factor overhead over a single Euclidean training pass with the same architecture while preserving the linear dependence on $E$ and $K$.

**Space Complexity.** Memory consumption is dominated by current node embeddings, attention buffers and temporal states. Storing node embeddings requires $O(Nd)$ memory. Attention coefficients and intermediate messages are kept for the current batch of edges, which costs $O(EH + Ed)$. The temporal kernels maintain GRU hidden states and pooled summaries of size $O(Lh + Kd)$. In addition, the decoder holds temporary buffers for at most $40\,000$ edges at a time for distance and loss computation, which is bounded by the edge caps discussed above. Crucially, we never store embeddings for all checkpoints in a trace: only the current temporal window and the recurrent state are retained. The overall space complexity is therefore

$$\text{Space} = O\big(Nd + EH + Kh\big), \tag{43}$$

rather than $O(TNd)$ in the length $T$ of the full optimization trace.

**Scaling Behaviour.** Collecting all components, the full time complexity of one training step is

$$\underbrace{O\big(LHNd^2 + LHEd\big)}_{\text{HGAT encoder}} + \underbrace{O\big(LK(h^2 + hd)\big)}_{\text{temporal kernels}} + \underbrace{O\big(d(1+\kappa)E^+_{\text{batch}}\big)}_{\text{decoder}} + \underbrace{O(Nd)}_{\text{Riemannian refinement}}. \tag{44}$$

In our setting, the parameter graphs satisfy $E \gg N$ and the embedding dimension $d$ is moderate. Under these conditions, the edge term $LHEd$ dominates the node–dependent terms $LHNd^2$ and $Nd$. Moreover, $E^+_{\text{batch}}$ is capped in practice, so the decoder contributes only a controlled linear factor in $E$.

Under these assumptions, the effective leading dependence simplifies to

$$O(LHEd) + O\big(LK(h^2 + hd)\big) + O\big(d(1+\kappa)E^+_{\text{batch}}\big), \tag{45}$$

which is linear in the number of edges $E$ and in the temporal window length $K$. Hyperbolic operations and the Riemannian refinement introduce only $O(d)$ costs per node or edge, adding a modest constant factor relative to a Euclidean graph meta network with the same $(L, H, d)$ but not altering the asymptotic scaling in model size.

## J Ablations

**Design of the ablations.** The full objective can be viewed as a core prediction loss equipped with a set of geometric and temporal regularizers. The core part consists of the Fermi–Dirac link loss $\mathcal{L}_{\text{FD}}$, which teaches the model which edges should exist, and the signed regression head $\mathcal{L}_{\text{wrec}} + \mathcal{L}_{\text{sign}}$, which aligns the predicted weights with the ground truth values. On top of this, the slope prior $\mathcal{L}_\alpha$ and the ranking loss $\mathcal{L}_{\text{rank}}$ encourage a monotone relation between hyperbolic distance and weight

Table 7: Ablation scenarios and their relation to the three guiding questions. Each row indicates which loss terms are active in the corresponding variant.

| Variant | Question | $L_{\text{FD}}$ | $L_{\text{wrec}}$ | $L_{\text{sign}}$ | $L_{\alpha}$ | $L_{\text{rank}}$ | $L_{\text{smooth}}$ |
|---|---|---|---|---|---|---|---|
| Full objective | – | ✓ | ✓ | ✓ | ✓ | ✓ | ✓ |
| No ranking loss | (1) | ✓ | ✓ | ✓ | ✓ | ✗ | ✓ |
| No ranking & no smoothness | (2) | ✓ | ✓ | ✓ | ✓ | ✗ | ✗ |
| No signed regression head | (3) | ✓ | ✗ | ✗ | ✗ | ✓ | ✓ |

magnitude, while the temporal smoothness term $\mathcal{L}_{\text{smooth}}$ prevents node embeddings from moving abruptly between consecutive checkpoints (equation 30).

In the ablation study we retain $\mathcal{L}_{\text{FD}}$ so that the model continues to learn meaningful link structure, and we selectively remove geometric terms to examine their individual contribution. Removing $\mathcal{L}_{\text{rank}}$ tests whether enforcing distance–magnitude ordering matters primarily for interpretability or also for INR accuracy. Removing both $\mathcal{L}_{\text{rank}}$ and $\mathcal{L}_{\text{smooth}}$ probes the importance of temporal regularity for maintaining a coherent geometry across the optimization trace. Finally, removing the signed regression head ($\mathcal{L}_{\text{wrec}}$, $\mathcal{L}_{\text{sign}}$, and $\mathcal{L}_{\alpha}$) asks whether explicit supervision on magnitudes and signs is required to bind weights to distance, or whether link prediction alone can sustain the geometric coupling. The following ablations on MNIST and Fashion–MNIST evaluate these questions in detail. We conduct ablations on the INR–trace image classification task because it is the setting in which all components of the composite objective are jointly active: link prediction, signed weight regression, distance–magnitude priors, ranking, and temporal smoothness all contribute to shaping the learned hyperbolic geometry. This task provides long, dense temporal traces (80–100 checkpoints per image), rich variation in weight magnitudes and signs, and a strong supervisory signal, making it the most sensitive and diagnostic environment for assessing how each loss term influences the geometry. Moreover, this is the setting in which our temporal and geometric meta–network attains high classification accuracy, ensuring that ablations probe a regime where the model operates at full capacity rather than in a degraded or low-signal setting.

**Ablation questions.** We design our ablations around three fundamental questions (Table 7):

1. Does the model still produce a meaningful geometry if it is never explicitly told that stronger weights should be closer? (No Ranking Loss)

2. If neither local geometric ordering nor temporal continuity is enforced, does the temporal hyperbolic geometry collapse? (No Ranking and Temporal Smoothness)

3. Can hyperbolic distances encode magnitude and polarity purely from link prediction? (No Sign Magnitude)

Each ablation scenario disables a specific subset of loss terms in order to answer exactly one of these questions while keeping the remaining components of the objective intact.

**Ablations on geometric objectives (MNIST / Fashion-MNIST).** We examine four variants of GTH–GMN on the INR classification task in order to isolate the role of each geometric loss. The variants are:

- The full objective.
- A model without the ranking loss ($\lambda_{\text{rank}} = 0$).
- A model without both ranking and temporal smoothness ($\lambda_{\text{rank}} = \lambda_{\text{smooth}} = 0$).
- A model without the signed regression head (dropping $\mathcal{L}_{\text{wrec}}$, $\mathcal{L}_{\text{sign}}$, and $\mathcal{L}_{\alpha}$ while keeping the Fermi–Dirac decoder).

For each condition we report Kendall's $\tau$ between predicted and actual accuracies, and the fitted slope of the distance–$\log |w|$ relation (Table 8).

Table 8: Ablation Study Table

| Condition | Metric | MNIST Value | Fashion-MNIST Value |
|---|---|---|---|
| Full Objective | $\tau$ | 0.946 | 0.935 |
| | Slope ($b$) | $-0.617$ | $-0.322$ |
| No Ranking Loss | $\tau$ | 0.811 | 0.797 |
| | Slope ($b$) | $-0.473$ | $-0.189$ |
| No Rank/Smooth | $\tau$ | 0.765 | 0.764 |
| | Slope ($b$) | $-0.365$ | $-0.061$ |
| No Signed Head | $\tau$ | 0.738 | 0.726 |
| | Slope ($b$) | $-0.211$ | $+0.125$ |

**Full objective.** With all losses active, the model exhibits the strongest alignment between its hyperbolic representations and the underlying predictive signal: Kendall's $\tau$ reaches 0.946 on MNIST and 0.935 on Fashion–MNIST. Geometrically, the distance–weight relation is sharply expressed. The inverse slope is steepest on MNIST ($-0.617$) and moderately steep on Fashion–MNIST ($-0.322$), reflecting the dataset-specific geometric requirements. The data shows a narrow concentration of points around this trend. Large weights consistently lie close to the origin, weaker weights move outward, and the trajectory across checkpoints remains stable. This is the regime in which the geometry, temporal evolution, and predictive signal remain coherent, forming the reference structure against which the ablations can be interpreted (Figures 6 ,7 ).

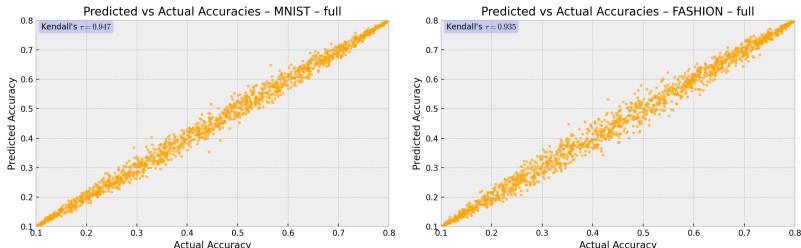

Figure 6: Kendall $\tau$ over the full objective for the INR classification task on MNIST and Fashion-MNIST datasets. GTH-GMN with full objective achieves a Kendall's $\tau$ of 0.946 on MNIST and 0.935 on Fashion–MNIST datasets.

**No ranking loss.** Removing $\mathcal{L}_{\mathrm{rank}}$ has a negative effect on both accuracy and the learned geometry. Kendall's $\tau$ drops to 0.811 in MNIST and 0.797 in Fashion–MNIST, reflecting that the model retains a broad performance gradient, but loses the finer monotonic structure present in the full objective (Figure 8). The fitted slope of the distance–$\log|w|$ relation (Figure 9) flattens significantly to $-0.473$ on MNIST and $-0.189$ on Fashion–MNIST, and the hexbins widen, indicating more frequent local inversions where strong and weak edges intermingle. The geometry remains meaningful, but it no longer enforces a crisp ranking between weight magnitudes.

**No ranking and no temporal smoothness.** When both $\mathcal{L}_{\mathrm{rank}}$ and $\mathcal{L}_{\mathrm{smooth}}$ are removed, the geometry loses coherence across time as well as within each snapshot. Kendall's $\tau$ declines further to 0.765 on MNIST and 0.764 on Fashion–MNIST (Figure 10), and the distance–$\log|w|$ slope becomes much shallower, falling to $-0.365$ on MNIST and $-0.061$ on Fashion–MNIST. The scatter rises sharply, and the expected ordering "stronger weights lie closer" breaks down in many regions. Without the temporal smoothness term, node embeddings can jump between checkpoints, making the temporal signal harder to follow and reducing the stability of the mapping between weights and their geometric placement. This ablation shows that ranking and smoothness act jointly: one enforces local ordering, the other maintains temporal continuity, and removing both produces a visibly unstable geometric structure (Figure 12).

**No signed regression head.** Removing the signed regression head produces the strongest degradation in the geometry. INR accuracy remains comparatively close to the full model, yet Kendall's

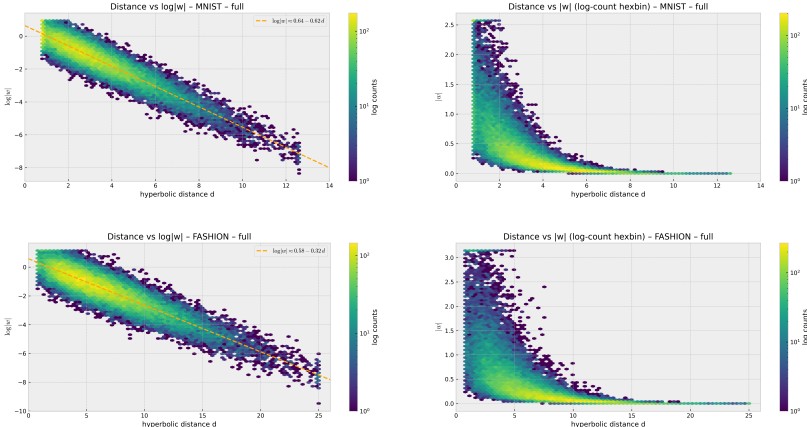

Figure 7: Analysis of the relationship between hyperbolic distance ($d$) and edge weights for the "full objective" condition. The top row displays results for MNIST, while the bottom row shows Fashion-MNIST. For each dataset, the left image shows the hexbin plot of distance vs. $\log|w|$ with a fitted linear regression line (dashed orange). The right image shows the hexbin plot of distance vs. the raw weight magnitude $|w|$. The fitted slope for MNIST in this condition ($-0.617d$) indicates a strong decay of edge weights, significantly steeper than for Fashion-MNIST ($-0.322d$), reflecting distinct geometric influences on model training for each dataset.

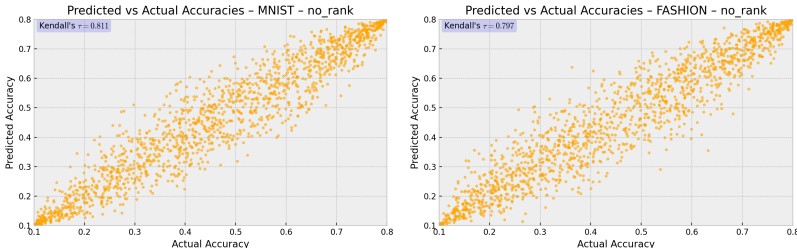

Figure 8: Kendall $\tau$ over the "No ranking loss" condition for the INR classification task on MNIST and Fashion-MNIST datasets. GTH-GMN without $\mathcal{L}_{\text{rank}}$ achieves a Kendall's $\tau$ of $0.811$ on MNIST and $0.797$ on Fashion–MNIST datasets, indicating a noticeable degradation in the monotonicity of the predictive signal compared to the full objective.

$\tau$ drops to $0.738$ on MNIST and $0.726$ on Fashion–MNIST (Figure 11). The distance–magnitude slopes collapse significantly on MNIST ($-0.211$) and even reverse on Fashion–MNIST ($+0.125$) (Figure 13), and the hexbins lose almost all discernible structure. Without supervision on magnitudes and signs, the model only learns which edges exist; it no longer learns how strong or influential they are, nor how polarity should be represented in tangent directions. Hyperbolic distances therefore reflect connectivity but not graded strength, and the embedding becomes insensitive to functional variation. This confirms that the signed head is the main mechanism that binds the geometry to weight values rather than merely assisting link prediction.

Taken together, these ablations show that the ranking and smoothness losses preserve the temporal and geometric coherence of the embeddings, while the signed head is essential for encoding magnitude–distance coupling. The INR classification task remains robust in several ablations, yet the auxiliary metrics and geometric plots reveal a steady erosion of the hyperbolic structure that supports our explanatory interpretations when each constraint is removed.

## K  FUTURE WORKS

Our approach of geometrically representing the temporal dynamics of deep neural networks naturally extends theoretically beyond MLPs because any neural architecture can be represented as a parameter graph (Lim et al., 2023). For CNNs, convolutional kernels give structured edge sets between input and output channel nodes forming a *multi-graph*, with kernel position and related

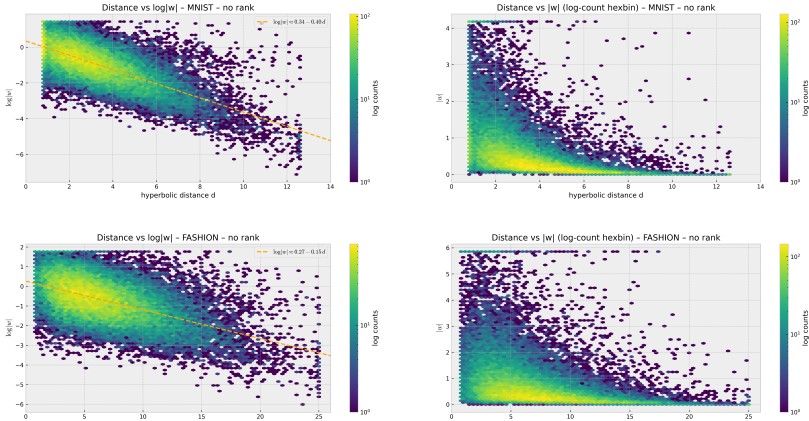

Figure 9: Analysis of the relationship between hyperbolic distance ($d$) and edge weights for the "No ranking loss" condition. The top row displays results for MNIST, while the bottom row shows Fashion-MNIST. For each dataset, the left image shows the hexbin plot of distance vs. $\log|w|$ with a fitted linear regression line (dashed orange). The right image shows the hexbin plot of distance vs. the raw weight magnitude $|w|$. The fitted slope for MNIST in this condition ($-0.473d$) indicates a moderate decay of edge weights, which is substantially shallower than the full objective model, while the decay for Fashion-MNIST ($-0.189d$) is notably weaker. This indicates that removing the ranking loss leads to a loss of the crisp, monotone geometric ordering.

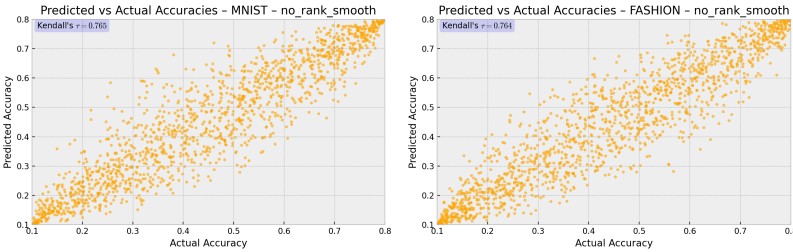

Figure 10: Kendall $\tau$ over the "No ranking and no temporal smoothness" condition for the INR classification task on MNIST and Fashion-MNIST datasets. GTH-GMN without both $\mathcal{L}_{\text{rank}}$ and $\mathcal{L}_{\text{smooth}}$ achieves a Kendall's $\tau$ of $0.765$ on MNIST and $0.764$ on Fashion–MNIST datasets, indicating a further decline in the ordering coherence of the predictive signal.

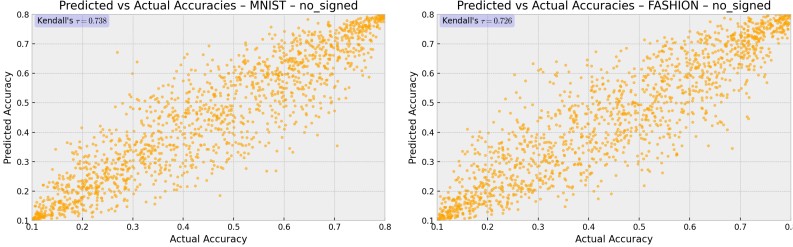

Figure 11: Kendall $\tau$ over the "No signed regression head" condition for the INR classification task on MNIST and Fashion-MNIST datasets. GTH-GMN without the signed head achieves a Kendall's $\tau$ of $0.738$ on MNIST and $0.726$ on Fashion–MNIST datasets, representing the strongest degradation in the geometric ordering of the predictive signal.

structure stored as edge features. For Transformers, attention layers induce subgraphs that connect query, key, value, and output channels, with head and layer indices and normalization parameters encoded as node and edge features (Lim et al., 2023). The deeper layers form the MLP parameter graphs and can be modeled as stated in this work for both CNNs and Transformers. Since the hyperbolic encoder operates only on node and edge structure, these architectures require no modification

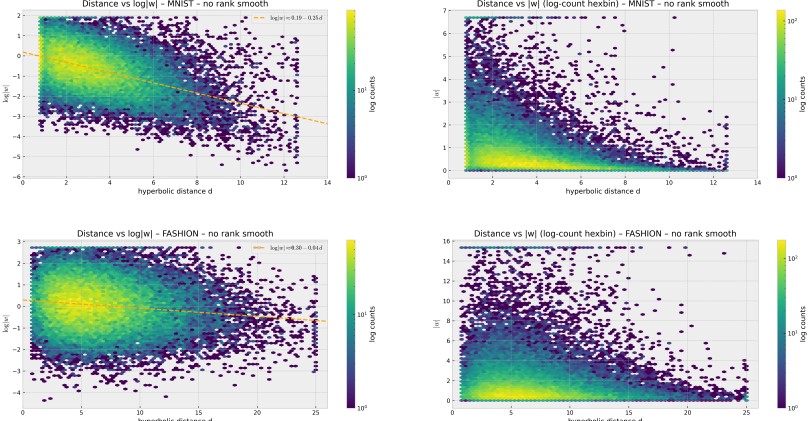

Figure 12: Analysis of the relationship between hyperbolic distance ($d$) and edge weights for the "No ranking and no temporal smoothness loss" condition. The top row displays results for MNIST, while the bottom row shows Fashion-MNIST. For each dataset, the left image shows the hexbin plot of distance vs. $\log|w|$ with a fitted linear regression line (dashed orange). The right image shows the hexbin plot of distance vs. the raw weight magnitude $|w|$. The fitted slope for MNIST in this condition ($-0.365d$) indicates a substantially shallower decay of edge weights compared to the full model, while the decay for Fashion-MNIST is minimal ($-0.061d$), reflecting a loss of coherent geometric structure when both regularization terms are removed.

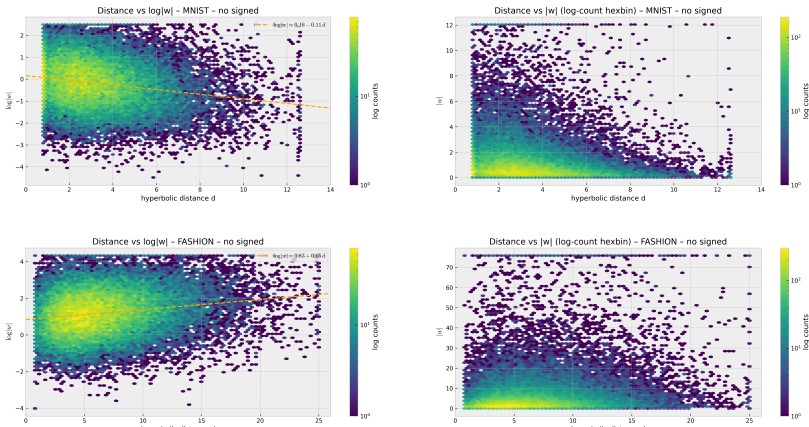

Figure 13: Analysis of the relationship between hyperbolic distance ($d$) and edge weights for the "No signed regression head" condition. The top row displays results for MNIST, while the bottom row shows Fashion-MNIST. For each dataset, the left image shows the hexbin plot of distance vs. $\log|w|$ with a fitted linear regression line (dashed orange). The right image shows the hexbin plot of distance vs. the raw weight magnitude $|w|$. The fitted slope for MNIST in this condition ($-0.211d$) is near-zero, and the slope for Fashion-MNIST ($+0.125d$) is **positive**, indicating that the expected decay of weights with distance has collapsed or reversed. This confirms that the signed head is the primary mechanism binding the hyperbolic geometry to the functional magnitude of the weights.

to the core method: their parameters simply instantiate richer graph topologies that the same temporal geometric machinery can process. Apart from that, recent work has shown that optimization trajectories admit coarse equivalence classes through topological conjugacy and Koopman spectral analysis (Redman et al., 2024). These dynamical characterizations complement our hyperbolic parameter graph approach. A promising future direction is to incorporate such global dynamical information into the temporal update module or the final task-specific readout, allowing these signals to enrich the geometric structure captured by GTH-GMN through a broader global context.

