}_{\mathrm{FD}} = -\,\mathbb{E}_{(u,v)\in E^+}\big[\log \psi^{(t)}(u, v)\big] - \mathbb{E}_{(u,v)\in E^-}\big[\log(1 - \psi^{(t)}(u, v))\big]. \tag{25}$$

To capture edge magnitudes and signs, we add the signed regressor of Section 3.3. For $(u, v) \in E^+$ with weight $w_{uv}^\star$ and prediction $\widehat{w}_{uv}^{(t)}$ from equation 13 (Goyal et al., 2018; Pareja et al., 2019),

$$\mathcal{L}_{\mathrm{wrec}} = \mathbb{E}_{(u,v)\in E^+}\left[\big\|\widehat{w}_{uv}^{(t)} - w_{uv}^\star\big\|_p\right], \tag{26a}$$

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

22: **end for**
23: $E \leftarrow \emptyset, \quad W_t \leftarrow \emptyset$
24: **for all** $k = \{u, v\}$ in $M$ **do**
25:      $e_{uv} \leftarrow M[k].\text{sum\_ctx}/M[k].\text{cnt}$           $\triangleright$ coalesced context (mean)
26:      $w_{uv}^\star \leftarrow M[k].\text{sum\_w}/M[k].\text{cnt}$           $\triangleright$ coalesced label (mean)
27:      insert $\{u, v\}$ into $E$ with context $e_{uv}$;      set $W_t[\{u, v\}] \leftarrow w_{uv}^\star$
28: **end for**
29: **for all** $u \in V$:    $s^{\text{abs}}(u) \leftarrow \sum_{v:\{u,v\}\in E} |W_t[\{u, v\}]|; \quad s^{\text{sgn}}(u) \leftarrow \sum_{v:\{u,v\}\in E} W_t[\{u, v\}]$
30: $\mu_{\text{abs}}, \sigma_{\text{abs}} \leftarrow$ mean/std of $\{s^{\text{abs}}(u) : u \in V\}$
31: $\mu_{\text{sgn}}, \sigma_{\text{sgn}} \leftarrow$ mean/std of $\{s^{\text{sgn}}(u) : u \in V\}$
32: **for all** $u \in V$ **do**
33:      $\widetilde{s}^{\text{abs}}(u) \leftarrow \begin{cases} \dfrac{s^{\text{abs}}(u) - \mu_{\text{abs}}}{\sigma_{\text{abs}}}, & \sigma_{\text{abs}} > 0 \\ 0, & \text{otherwise} \end{cases}$
34:      $\widetilde{s}^{\text{sgn}}(u) \leftarrow \begin{cases} \dfrac{s^{\text{sgn}}(u) - \mu_{\text{sgn}}}{\sigma_{\text{sgn}}}, & \sigma_{\text{