# OpenReview forum: "Temporal Geometry of Deep Networks: Hyperbolic Representations of Training Dynamics for Intrinsic Explainability"
_ICLR.cc/2026/Conference — ICLR 2026 Poster_

### Official Review · Reviewer_3BLp · 2025-10-19

**Soundness:** 2
**Presentation:** 1
**Contribution:** 2
**Rating:** 4
**Confidence:** 2

**Summary:**

This paper proposes to incorporate temporal and geometric considerations into study neural networks as data points (e.g. in INRs). The authors propose a methodology the process INR trajectories using the Poincare ball model of hyperbolic geometry to use for metanetwork purposes (process NNs as datapoints e.g. for classification of INRs or predicting generalistion from the trained weights themselves)

I have given a 4 but would give a 3 if there was the option.

**Strengths:**

1. The paper seems correct and seems to have succeeded in adding temporal and geometric considerations into a framework for meta-networks.
2. There is a lot of background material involved in the paper. this is both a strength, as there is a lot of technical detail to get right, but also  a weakness (see below).

**Weaknesses:**

I'd say overall my criticisms of the paper stem from the fact that I found it hard to read. This is by no means my area, but even still I think the authors could have made several improvements to the presentation:

1. The motivation isn't very strong. The paper focuses on the technical problem (of adding temporal and geometric aspects to meta-networks), but not enough of the paper addresses why this is a problem to focus on.
2. Likewise, the empirical results for the methodology (the proposed GTH-GMN) of this paper are quite bare. There are 6 terms in the loss function, each of which are motivated in section 3: how is the empirical performance of GTH-GMN affected without each of the 6 terms? This is essential for motivating the method. Why do we need two optimisation steps? The paper is missing quite a lot of ablations imo. Relatedly, is it essential to have temporal *and* geometric aspects to the framework: what happens if you remove one (how does performance vary as the number of checkpoints changes)?
3. The contributions section (last paragraph of section 1) isn't very clear imo. The paragraph reads as if the contributions are methodological, are there any theoretical challenges that the authors needed to overcome in order to create the methodology (or was it mainly a case of applying and combining existing theory into a method). Likewise, I think a background section is missing between the related work and the research method sections in order to provide context for the reader of the most relevant background info (e.g. what is the most relevant methodological baseline and how does it work? Is this GMN, based on the name GTH-GMN?)
4. How much more expensive is it to have to track the full trajectory that say the final checkpoint?

**Questions:**

See above

---

> ### Author Response · Authors · 2025-11-20
> **Response to Reviewer 3BLp:**
>
> Weaknesses:
>
> 1. The motivation isn't very strong. The paper focuses on the technical problem (of adding temporal and geometric aspects to meta-networks), but not enough of the paper addresses why this is a problem to focus on.
>
>
> Response: We have acknowledged the feedback of the reviewer and have made attempts to bolster our motivation. In Lines 43-48 we now specifically address why this is a problem to focus on by specifying:
>
>  “Our focus in this work is not only to investigate the performance of hyperbolic and temporal meta-networks, but also to expose how a neural network’s structure forms and shifts during training. A temporal geometric view makes it possible to track the emergence of hierarchy, the consolidation of functional modules, and the redistribution of influence across layers by studying the evolution of its parameter graph. This perspective offers intrinsic explanations grounded in the model’s own dynamics and self-organization, rather than relying on post-hoc probes for explanation.”
>
>
> Continued below..

---

> > ### Author Response · Authors · 2025-11-20
> >
> > 2. Likewise, the empirical results for the methodology (the proposed GTH-GMN) of this paper are quite bare. There are 6 terms in the loss function, each of which are motivated in section 3: how is the empirical performance of GTH-GMN affected without each of the 6 terms? This is essential for motivating the method. Why do we need two optimisation steps? The paper is missing quite a lot of ablations imo. Relatedly, is it essential to have temporal and geometric aspects to the framework: what happens if you remove one (how does performance vary as the number of checkpoints changes)?
> >
> > Response: We thank the reviewer for identifying the lack of empirical validation for our loss components, and we fully agree that determining the contribution of each term is essential for motivating the method. To address this, we have added a comprehensive Ablation Study in Appendix J of the revised manuscript, where we quantitatively and geometrically evaluate the impact of removing specific constraints on the INR classification task. We tested the model under four distinct conditions: the full objective, without the ranking loss, without both ranking and temporal smoothness, and without the signed regression head. We retained the Fermi-Dirac link loss in all scenarios to ensure the model could always learn a baseline topological structure .
> > The results confirm that each term plays a distinct and critical role. Removing the Ranking Loss caused a significant drop in predictive fidelity, indicating that strict geometric ordering is necessary for high accuracy rather than just interpretability. Furthermore, removing the Temporal Smoothness regularizer led to additional degradation and unstable embeddings, confirming that temporal continuity is required to maintain a coherent trajectory across optimization steps. Most critically, removing the Signed Regression Head resulted in a complete collapse of the geometric structure; the characteristic distance-magnitude slope flattened to near-zero, essentially decoupling the embedding geometry from the functional weight values. These findings empirically demonstrate that the full objective is not merely a collection of auxiliary terms, but a cohesive system where the signed head binds magnitude to geometry, the ranking loss refines this ordering for accuracy, and the smoothness term ensures the trajectory represents a valid evolutionary path.
> >
> > We hope these changes satisfy the concerns raised in the feedback.
> >
> > Continued below..

---

> ### Author Response · Authors · 2025-11-20
>
> Now we would like to respond to another query of the reviewer “Why do we need two optimisation steps?”. We have made the point regarding the two-phase optimisation explicit in Lines 389-392, acknowledging the reviewers' feedback. To specifically answer the question:
>
> We employ a two-phase optimization strategy because our model architecture bridges two fundamentally different geometric spaces. The attention stack and regressors operate in Euclidean space, whereas the node embeddings evolve on a curved manifold (the Poincaré ball); using a single update rule would inevitably either ignore the curvature required for the embeddings or force the Euclidean modules to inherit manifold constraints they were not designed for. By separating the process, Phase 1 allows for standard backpropagation on the kernels and regressors, while Phase 2 performs a dedicated Riemannian pass that refines node positions $z^{(t)}$ directly on the manifold. This intrinsic second step rescales Euclidean gradients with the conformal factor to ensure updates follow the manifold geometry, applies updates via the exponential map, and parallel transports moment vectors to maintain optimizer coherence. Ultimately, this design decouples the stability of the Euclidean components from the dynamics of the hyperbolic embeddings, ensuring that the geometry is strictly respected while maintaining stability over long training traces.
> Regarding the concern “Relatedly, is it essential to have temporal and geometric aspects to the framework: what happens if you remove one (how does performance vary as the number of checkpoints changes)?”, we would like to state the following:
> We thank the reviewer for this question, as it allows us to clarify the fundamental premise of our framework. While we provide empirical ablations in Appendix J to demonstrate that removing certain loss components indeed degrades predictive performance, we emphasize that the inclusion of temporal and geometric modules is not merely a strategy for maximizing raw accuracy on certain meta-predictive tasks. Rather, these components are ontologically necessary to achieve the paper's primary goal, which is capturing and interpreting the training dynamics of neural networks. To study the evolution of a network, one must fundamentally model the temporal dimension; removing the temporal component collapses the trajectory into a single static point, rendering the analysis of "dynamics" (e.g., drift rate, stabilization) impossible. Similarly, to study the structure of the parameter graph as it evolves, one must assume an underlying geometry that can represent its topological properties. We employ hyperbolic geometry specifically because it naturally accommodates the scale-free, hierarchical organization of neural weights. Therefore, removing either the geometric or temporal aspect would not just vary performance; it would fundamentally dissolve the object of study, stripping the framework of its ability to provide the intrinsic explainability and trajectory visualizations (as shown in Figure 2) that constitute our main contribution.
> We hope that the above explanation clarifies the reviewers concern regarding our methodology and ideology.
>
> Continued Below..

---

> ### Author Response · Authors · 2025-11-20
>
> 3. The contributions section (last paragraph of section 1) isn't very clear imo. The paragraph reads as if the contributions are methodological, are there any theoretical challenges that the authors needed to overcome in order to create the methodology (or was it mainly a case of applying and combining existing theory into a method). Likewise, I think a background section is missing between the related work and the research method sections in order to provide context for the reader of the most relevant background info (e.g. what is the most relevant methodological baseline and how does it work? Is this GMN, based on the name GTH-GMN?)
>
> Response: We thank the reviewer for highlighting the concern regarding the contributions paragraph. We acknowledge that the initial phrasing may have appeared to present the work as a straightforward combination of existing methods. To address this, we have revised the Introduction (Lines 81-101) to explicitly articulate the core theoretical challenge: establishing a unified framework that simultaneously respects the permutation symmetries of weight space, the hierarchical topology of scale-free networks, and the temporal continuity of optimization trajectories.
> This work represents a non-trivial assimilation of concepts from disparate fields, specifically temporal graph learning, hyperbolic geometry, and network science physics. The primary theoretical hurdle was defining a rigorous mapping from Euclidean parameter tensors to the Poincaré ball that preserves functional equivalence (invariance to node permutations) while enforcing a consistent metric interpretation (distance-magnitude coupling) across time.
> The revised text in the manuscript now reads as follows:
> "We close this gap by developing a geometric temporal hyperbolic framework for parameter graphs that treats training itself as an object of study. Conceptually, we view the optimization of a neural network as a trajectory across weighted parameter graphs evolving in a negatively curved space, and we argue that intrinsic explanations should be grounded in this evolution rather than in isolated checkpoints. Methodologically, we (a) construct a temporal hyperbolic meta network (GTH–GMN) that (b) models MLP parameter traces as permutation equivariant parameter graphs, (c) embedding them in the Poincaré ball, and (d) coupling distance-biased hyperbolic attention with meta evolution of the attention kernels over time. The architecture is designed to respect the symmetries of weight space: it is equivariant to the neuron permutations within each snapshot and invariant to permutations of past snapshots, so that functionally equivalent networks share the same representation, while their latent geometry can still be recovered. A signed weight regressor links geodesic distance to edge magnitude through a power law prior, decoupling magnitude from polarity and importing scale-free inductive biases from network science. Empirically, we show on INR classification, CIFAR–10 generalisation prediction, and sinusoid regression, that this temporal hyperbolic encoder matches or heavily approaches strong Euclidean and tensor-based baselines, while producing compact embeddings whose radial and angular structure tracks the self-organization of the underlying networks. In this way, the method offers a concrete route toward intrinsic explainability through the geometry of parameter graph trajectories"
> We believe this formulation better captures the theoretical synthesis required to make these distinct geometric and temporal constraints coexist within a single differentiable framework.
>
> Additionally, regarding the inclusion of a background section in the manuscript, we thank the reviewer for this suggestion and agree that establishing the theoretical context is vital for a complex geometric framework. However, due to strict page limitations and the necessity of detailing the mathematical machinery of our specific encoder, it was not possible to include a general background section within the main text. Instead, we have ensured the manuscript remains rigorously self-contained by providing comprehensive Theoretical Preliminaries in Appendix B. This section meticulously defines the Riemannian Manifold and Poincaré ball model, including the specific exponential and logarithmic maps used in our architecture. Furthermore, relevant methodological baselines, such as Graph Meta Networks (GMN), are discussed directly within the Related Work section to immediately contextualize our contribution. To further aid the reader, we have also revised Sections 3.2 to 3.4 to include "Intuition" paragraphs, which serve to bridge the gap between these theoretical foundations and our specific methodological choices without requiring a separate section.
>
> We hope that our response clarifies the reviewer’s concern.
>
> Continued Below..

---

> ### Author Response · Authors · 2025-11-20
>
> 4. How much more expensive is it to have to track the full trajectory that say the final checkpoint?
>
> Response: We address the specific cost of tracking the full trajectory versus a single checkpoint in Appendix I (Computational Cost Analysis). We explicitly derive that the computational expense scales linearly with the number of temporal snapshots $K$ included in the trace window. Specifically, the temporal kernel evolution introduces a cost term of $O(LK(h^2+hd))$, meaning that processing a trajectory of $K$ steps requires $K$ updates to the recurrent state, compared to a single update for a final checkpoint. However, we emphasise that this linear increase in compute does not translate to a linear explosion in memory. By utilising recurrent kernel meta-evolution, we update the weights of the hyperbolic encoder over time without storing the high-dimensional node embeddings for the entire history. Consequently, the space complexity remains $O(Nd + EH + Kh)$ rather than scaling with the full trace length $O(TNd)$, rendering the tracking of full training dynamics computationally feasible even for long trajectories.
>
> We hope that this addresses the reviewer’s concerns regarding processing temporal traces.

---

### Official Review · Reviewer_dcw3 · 2025-10-29

**Soundness:** 3
**Presentation:** 2
**Contribution:** 2
**Rating:** 4
**Confidence:** 2

**Summary:**

This paper models the training process of neural networks as a sequence of evolving graphs. Each of these graphs is embedded in hyperbolic space. The authors introduce a Hyperbolic Graph Meta-Network (GTH-GMN) that learns to encode the temporal evolution of these graphs using hyperbolic attention and recurrent updates. This geometric representation is intended to provide intrinsic explainability by revealing structure in training dynamics. Experiments on small-scale tasks show consistent improvements over prior Euclidean and static graph baselines, along with interpretable visualizations of how networks evolve during training.

**Strengths:**

The paper's main strength is in its main idea: to think about the training of a neural network not just as a sequence of parameter updates, but as a geometric process that unfolds in hyperbolic space. This view allows the authors to tie together many ideas from disparate fields. The resulting metrics for tracking training seem to capture meaningful structure in how networks evolve, and the visualizations provide some insights into the learning process.

**Weaknesses:**

I should preface my comments by noting that I am not an expert in the specific technical areas this paper draws on, which is reflected in my relatively low confidence score. I found the paper quite challenging to follow at times, largely due to its dense mathematical notation and the level of background knowledge it assumes from the reader. Despite the impressive mathematical machinery, I am not entirely convinced that the insights gained from this approach justify the conceptual and computational complexity it introduces.

**Questions:**

1. How sensitive are the results to the choice of curvature or manifold model? For example, if the same temporal graph encoder were trained in Euclidean or spherical space, would the geometric patterns and performance differences still hold?

2. Can the authors clarify what specific types of interpretability their method enables? Beyond visualizing trajectories, are there measurable insights (such as neuron redundancy, layer specialization, or early-stopping indicators) that can be extracted from the hyperbolic embeddings?

3. The experiments focus on relatively small MLPs and simple datasets. How might this framework scale to larger architectures, such as Transformers or CNNs, where parameter spaces and symmetries are more complex? Are there computational or conceptual challenges expected in extending the approach?

---

> ### Author Response · Authors · 2025-11-20
> **Response to Reviewer dcw3**
>
> Weaknesses:
>
> 1. I should preface my comments by noting that I am not an expert in the specific technical areas this paper draws on, which is reflected in my relatively low confidence score. I found the paper quite challenging to follow at times, largely due to its dense mathematical notation and the level of background knowledge it assumes from the reader. Despite the impressive mathematical machinery, I am not entirely convinced that the insights gained from this approach justify the conceptual and computational complexity it introduces.
>
> Response: We appreciate the candid feedback from the reviewer and acknowledge that the previous draft was mathematically dense, affecting readability to quite an extent. Considering this specific feedback, we have performed a significant restructuring of Section 3, specifically Sections 3.2 to 3.4. To address the readability concerns, we have comprehensively revised the manuscript structure by prioritising intuitive explanations over dense formalism in Sections 3.2, 3.3, and 3.4. Specifically, we moved the detailed optimisation derivation to Appendix G to allow for a clearer and conceptual exposition of the two-phase schema in the main text.  Please look into the general comment for more specific details on this. Moreover, we have also provided clear goals of our experiments along with a detailed discussion on their interpretation in Section 5, along with the implications for explainability and outlining a path for adapting the framework for more complex neural models such as CNNs and Transformers.
> Regarding the concern on whether the complexity justifies the insights, we have clarified that this machinery is necessary to achieve intrinsic explainability, which is a capability standard Euclidean methods lack. In this light, we have provided a comprehensive ablation study in Appendix J demonstrating that removing these geometric constraints causes the interpretable structure to collapse, confirming they are essential for the method's function rather than just theoretical flourishes. Additionally, we added a cost analysis in Appendix I showing that the method scales linearly, ensuring that the theoretical complexity remains computationally tractable. Please check the response to reviewer fCBW for more detailed comments on ablations and their implications.

---

> > ### Author Response · Authors · 2025-11-20
> >
> > Questions:
> > 1. How sensitive are the results to the choice of curvature or manifold model? For example, if the same temporal graph encoder were trained in Euclidean or spherical space, would the geometric patterns and performance differences still hold?
> >
> > Response: We thank the reviewer for this fundamental question regarding the role of curvature. The choice of hyperbolic geometry is not merely an implementation detail but a structural prior motivated by the hypothesis that neural parameter graphs share key topological properties with complex scale-free networks, such as heavy-tailed degree distributions and hierarchical modularity. Our empirical findings, particularly the emergent distance-magnitude coupling shown in Figure 4 and the layer-wise separation in Figure 2, strongly validate this assumption. Hyperbolic space ($c < 0$) is uniquely suited for this data structure because its volume expands exponentially, allowing it to embed these emerging hierarchies with arbitrarily low distortion. In our embeddings, this specific geometry is what enables the distinct radial separation of layers and the outward drift of hubs; the negative curvature provides the necessary "room" for the hierarchy to organize itself naturally.
> > In contrast, Euclidean space ($c=0$) expands only polynomially. To embed a deepening hierarchy such as a training trajectory refining its features into Euclidean space, nodes must either be crowded together or the embedding dimensionality must be drastically increased to compensate for the lack of volume. If we trained the same encoder in Euclidean space, the distinct "bands" and radial stratification observed in our results would likely collapse into a dense, uninterpretable cloud, effectively destroying the geometric explanatory power of the method. Similarly, Spherical geometry ($c>0$) has finite volume and is cyclical. While useful for periodic data, neural network training is a trajectory of increasing complexity and specialization, which may not always be a cycle. A spherical embedding would force the diverging trajectories of weights to wrap around themselves, creating false proximities and failing to capture the "expansion" of the network's functional space.
> > Therefore, while a Euclidean encoder might achieve comparable predictive performance on some metrics (often by utilizing much higher dimensions), the resulting geometric patterns would fundamentally change. Specifically, the monotonic coupling between distance and weight magnitude, as well as the visualizable radial organization of layer-wise specialization, are intrinsic properties of the negative curvature and would not hold in flat or positively curved spaces.
> >
> > Continued below..

---

> > > ### Author Response · Authors · 2025-11-20
> > >
> > > 2. Can the authors clarify what specific types of interpretability their method enables? Beyond visualizing trajectories, are there measurable insights (such as neuron redundancy, layer specialization, or early-stopping indicators) that can be extracted from the hyperbolic embeddings?
> > >
> > > Response: We thank the reviewer for pushing us to clarify the specific, measurable forms of interpretability our method enables beyond qualitative visualization. While our primary contribution is establishing the theoretical framework for modeling temporal parameter graphs in negatively curved spaces (specifically, ensuring functional invariance to past snapshots and equivariance to within-snapshot permutations) , we agree that this geometric structure naturally yields quantifiable metrics for intrinsic explainability.
> > >
> > > In the revised Section 5 (Discussion), we have explicitly connected the geometric properties of the Poincaré embeddings to specific, measurable insights regarding network self-organization.
> > >
> > > The hyperbolic embeddings show a measurable "drift" of layers. We can quantify Radial Separation to measure how distinct layers become in terms of their hierarchical importance (distance from the origin) and Angular Separation to measure functional divergence. A high rate of angular separation between layers over time indicates the emergence of specialized features, while low separation suggests redundancy. Apart from that, in the Poincaré ball, nodes close to the origin act as hierarchies or "hubs," while leaf nodes reside near the boundary. We can track the Radial Drift Rate of individual neurons. Neurons that remain stable near the origin are central processing hubs, while those that drift rapidly toward the boundary with high angular variance likely represent specialized, fine-grained features. Conversely, nodes whose trajectories collapse together or drift uniformly to the boundary without distinct angular separation are candidates for redundancy. Additionally, the speed at which embeddings move through the hyperbolic space (the geodesic velocity) serves as a proxy for the rate of self-organization. The smooth evolution of trajectories allows us to track the stabilization of the network's topological structure over time.
> > >
> > > Furthermore, we would like to also outline how these geometric metrics translate into concrete actions for model analysis and optimization. Firstly, standard early stopping relies on validation loss plateaus, which can be noisy. Our framework allows for tracking the aggregate geodesic drift rate of the parameter graph. A plateau in the drift rate may signals that the network’s topological structure has stabilised, offering an intrinsic, geometry-based stopping criterion. Secondly, by identifying Angular Clustering near the boundary, we can detect neurons that provide redundant functional information. Clusters of neurons that drift to the boundary while maintaining high angular proximity represent functional duplicates that can be pruned without impacting the core topology. Third, when a model is transferred to a new domain, the geometric volatility (high geodesic velocity) of specific layers may highlight where the domain shift impacts the internal representation the most. This allows for a principled adaptation by targeting fine-tuning efforts at the layers undergoing the most rapid geometric reorganisation.
> > >
> > > Continued below..

---

> > > > ### Author Response · Authors · 2025-11-20
> > > >
> > > > 3. The experiments focus on relatively small MLPs and simple datasets. How might this framework scale to larger architectures, such as Transformers or CNNs, where parameter spaces and symmetries are more complex? Are there computational or conceptual challenges expected in extending the approach?
> > > >
> > > > Response: We thank the reviewer for raising this crucial point. We agree that validating the framework on MLPs is a starting point, and addressing the scalability to complex architectures like CNNs and Transformers is essential for assessing the method's broader utility. To address this, we have expanded Section 5 (Discussion and Future Work) and added Appendix I (Computational Cost Analysis) to explicitly outline the conceptual mapping and computational feasibility of these extensions.
> > > > 1. The core insight of our framework is that the hyperbolic encoder operates on graph topology, not tensor dimensions. Therefore, extending to CNNs or Transformers requires only a modification of the graph construction step, while the geometric machinery remains unchanged.
> > > > CNNs: These are modeled as multi-graphs where nodes represent input and output channels. The convolutional kernel weights become directed edges connecting these channel nodes. To preserve the specific inductive biases of CNNs, the spatial position of a weight within the kernel (e.g., center vs. corner) is encoded as a distinct edge feature. This construction naturally respects channel permutation symmetries, just as our current model respects neuron permutation symmetries.
> > > > Transformers: Transformer blocks are decomposed into functional subgraphs. The Query, Key, Value, and Output projection matrices act as bipartite graphs connecting embedding channels. Attention head indices are encoded as node features, allowing the graph to capture the complex symmetries of the multi-head attention mechanism without altering the encoder's core logic.
> > > > 2. Regarding the computational challenge, we provide a detailed complexity analysis in Appendix I. We show that the time complexity of our method is dominated by the edge processing term $O(LHEd)$.
> > > > Linear Scaling: This means the method scales linearly with the number of edges $E$. While CNNs and Transformers have denser parameter graphs than MLPs, the cost remains asymptotically comparable to standard Euclidean GNNs operating on the same graph data.
> > > > Memory Efficiency: By using recurrent kernel meta-evolution, we avoid storing the full history of embeddings, keeping space complexity $O(Nd + EH + Kh)$ decoupled from the trace length.
> > > > Therefore, while "heavier" architectures increase the constant factors involved (due to larger $E$), the approach is computationally tractable and mathematically consistent with the complex parameter spaces of modern deep learning models.

---

### Official Review · Reviewer_fCBW · 2025-10-30

**Soundness:** 3
**Presentation:** 3
**Contribution:** 3
**Rating:** 6
**Confidence:** 2

**Summary:**

This paper proposes a novel framework for studying MLP training dynamics through hyperbolic geometry. The authors construct temporal parameter graphs from checkpoints during training and embed them in the Poincaré ball using hyperbolic graph attention networks (HGAT) with kernel meta-evolution. The approach maintains permutation equivariance and ties edge weights to hyperbolic distances via power-law relationships, providing interpretable geometric representations of network self-organization during training.

**Strengths:**

- Novel geometric perspective: The application of hyperbolic geometry to temporal parameter graphs is original and well-motivated. The connection to network science findings (scale-free, hierarchical organization) provides strong theoretical justification.
- Temporal modeling: Unlike most prior meta-learning work that analyzes single checkpoints, this work explicitly models training trajectories.
- Comprehensive methodology: The signed weight regressor with magnitude-distance power law (Eq. 13) and conformal sign prediction (Eq. 15) shows careful design. The two-phase optimization (Euclidean + Riemannian) is well-executed.

**Weaknesses:**

- Limited performance gains: On CIFAR-10 generalization prediction, the method achieves τ=0.846±0.004, notably below NFN baselines (0.922-0.934). While the authors acknowledge this is "expected," it raises questions about practical utility. The gap suggests the geometric compression may discard accuracy-correlated information that end-to-end methods capture.
- Architectural limitations: The approach is restricted to MLPs. The authors mention this limitation but don't provide a clear path to extending to CNNs, Transformers, or other architectures that dominate modern deep learning. This severely limits practical applicability and impact.
- Computational cost not addressed: The paper doesn't discuss training time, memory requirements, or scalability. Hyperbolic operations, Riemannian optimization, and temporal processing likely add significant overhead.
Tables 5-6 mention "practical caps" but no runtime comparisons are provided.
- High variance in some experiments: Sinusoid task: 1.06±0.24 MSE vs GMN's 1.13±0.08 - the 3x higher variance is concerning and poorly explained. The authors attribute this to "known sensitivities in Riemannian optimization" but don't investigate mitigation strategies.

**Questions:**

- Can you provide runtime/memory comparisons with baselines? How does the approach scale with model size?
- What specific architecture modifications would be needed to extend beyond MLPs? Is this fundamentally intractable?
- Could you provide ablations showing the contribution of each loss term (Eq. 32)?

---

> ### Author Response · Authors · 2025-11-20
> **Response to reviewer fCBW:**
>
> Weaknesses:
>
> 1. Limited performance gains: On CIFAR-10 generalization prediction, the method achieves τ=0.846±0.004, notably below NFN baselines (0.922-0.934). While the authors acknowledge this is "expected," it raises questions about practical utility. The gap suggests the geometric compression may discard accuracy-correlated information that end-to-end methods capture.
>
> Response: We acknowledge the reviewer for the thoughtful comments. To an extent, we agree that we achieve a relatively modest Kendall’s tau over the CIFAR-10 dataset as compared to the neural functional (NFN) baselines. We would also like to present a supporting insight on this matter. Recent work by Meynent et al. (https://arxiv.org/abs/2503.17138) corroborates our findings regarding the performance gap on CIFAR-10. They demonstrate that methods relying on structural reconstruction of neural weights exhibit a 'smoothing bias' that captures coarse, global features while averaging out fine-grained functional details. Our GTH-GMN is explicitly designed to capture geometric structure (parameter graphs in hyperbolic space). Consequently, it prioritises the 'coarse' macroscopic dynamics required for intrinsic explainability (such as the layer separation seen in Figure 2) over the microscopic weight fluctuations that raw-tensor methods like NFNs leverage for higher predictive precision. Thus, we see this as a natural trade-off for achieving geometric interpretability from a practical utility standpoint. In general, we agree with the reviewer's assessment that our geometric compression likely discards fine-grained information correlated with accuracy. However, we view this as a necessary trade-off to achieve intrinsic explainability (via representing the underlying structure within hyperbolic spaces), a view strongly corroborated by recent findings in weight-space learning. Empirically, the model is competitive and produces near SOTA results on 2 out of 3 tasks we used in the evaluation.
>
>
> 2. Architectural limitations: The approach is restricted to MLPs. The authors mention this limitation but don't provide a clear path to extending to CNNs, Transformers, or other architectures that dominate modern deep learning. This severely limits practical applicability and impact.
>
> Response: We thank the reviewer for this insightful comment. We fully agree that restricting the empirical evaluation to MLPs is a limitation of the current study. Our primary objective in this work was to establish the theoretical framework for modelling the geometric and temporal dynamics of neural parameter graphs and to demonstrate the specific interpretability patterns that emerge in negatively curved spaces.
>
> To address the concern regarding broader applicability, we have expanded Section 5 (Lines 526-534 in the revised manuscript) to provide a concrete theoretical roadmap for extending this framework to modern architectures. Specifically, we detail how CNNs can be modelled as multi-graphs where edges connect input/output channels, and how Transformers induce subgraphs connecting query, key, and value channels. Crucially, because our hyperbolic encoder operates strictly on the node and edge structure, the core methodology requires no modification; these complex architectures simply instantiate richer graph topologies for the same temporal-geometric machinery to process
>
> We hope this discussion clarifies the method's potential for broader application, as detailed in our General Response.

---

> > ### Author Response · Authors · 2025-11-20
> >
> > Continued from above..
> >
> > 3. Computational cost not addressed: The paper doesn't discuss training time, memory requirements, or scalability. Hyperbolic operations, Riemannian optimization, and temporal processing likely add significant overhead. Tables 5-6 mention "practical caps" but no runtime comparisons are provided.
> >
> > Response: We thank the reviewer for highlighting the need for a rigorous discussion on computational resources. We agree that the combination of temporal processing and hyperbolic geometry raises valid questions regarding scalability. To address this, we have added a detailed Computational Cost Analysis in Appendix I of the revised manuscript.
> >
> >
> > Below is a summary of our analysis demonstrating that the method scales linearly with graph size and remains memory-efficient.
> > Time Complexity & Scalability: We derive the full time complexity of a training step as the sum of the HGAT encoder, temporal kernels, decoder, and Riemannian refinement. In the regimes we study (where edges $E \gg$ nodes $N$), the cost is dominated by the edge processing term $O(LHEd)$. Consequently, the method is effectively linear in the number of edges $E$.
> > Hyperbolic Overhead: While hyperbolic operations (exponential/logarithmic maps, parallel transport) add computational steps, they are all $O(d)$ vector calculations. Asymptotically, this introduces only a constant factor overhead relative to a standard Euclidean Graph Attention Network with the same depth and width, preserving the linear scaling.
> > Temporal Processing & Memory: A key concern was the overhead of temporal processing. By employing kernel meta-evolution (evolving the weights of the GNN rather than storing embeddings), we decouple memory usage from the full trace length $T$. The space complexity is $O(Nd + EH + Kh)$, where $K$ is the window size. This avoids the prohibitive $O(TNd)$ cost of storing full trajectories, allowing the method to scale to long optimisation traces ($T=100+$) efficiently.
> > Riemannian Optimization: The intrinsic Riemannian refinement phase updates only node positions and costs $O(Nd)$. Since this reuse edge samples and losses from the Euclidean pass, it adds a minimal constant overhead without affecting the overall complexity class.
> >
> > We believe this analysis clarifies that GTH-GMN is computationally tractable and scales comparably to standard temporal GNNs.
> >
> > 4. High variance in some experiments: Sinusoid task: 1.06±0.24 MSE vs GMN's 1.13±0.08 - the 3x higher variance is concerning and poorly explained. The authors attribute this to "known sensitivities in Riemannian optimization" but don't investigate mitigation strategies.
> >
> > Response: We appreciate the reviewer's concern regarding the observed high variance on the Sinusoid task. We have addressed this point in Section 5 (Lines 510-516 ) of the updated manuscript. We have outlined the relatively high variance as an interplay between the effect of curvature and temporal coupling. Moreover, the study assumes a fixed curvature of -1 or |c| = 1, which is a simplification. Adapting the curvature of the model dynamically is one of the sensible and non-trivial approaches for reducing the observed variances, which we also outline as a pragmatic direction for future studies. Moreover, Lines 514-516 briefly address some mitigation strategies that could be applied to reduce the variability in performance.

---

> ### Author Response · Authors · 2025-11-20
>
> Questions:
> 1. Can you provide runtime/memory comparisons with baselines? How does the approach scale with model size?
>
> Response: We have provided a detailed computational cost analysis in Appendix I. Also, check the response to weakness 3 in the above comments.
>
>
> 2. What specific architecture modifications would be needed to extend beyond MLPs? Is this fundamentally intractable?
>
> Response: We thank the reviewer for this insightful question regarding the broader applicability of our framework. To answer directly, extending this approach to architectures beyond MLPs is not fundamentally intractable. The core insight of our framework is that the hyperbolic encoder and temporal kernel operate on the graph topology defined by nodes and edges rather than on specific tensor dimensions. Consequently, the underlying mathematical machinery spanning hyperbolic attention, signed regression, and Riemannian optimization remains unchanged regardless of the specific architecture being analyzed. The only necessary modification lies in the graph construction step, specifically regarding how parameters are mapped to nodes and edges.
>
> In the revised Section 5, we have explicitly outlined the theoretical roadmap for these extensions (Lines 526-534). Convolutional Neural Networks can be modeled as multi-graphs where the input and output channels of a layer serve as nodes, while the convolutional kernel weights act as directed edges connecting these channel nodes. Crucially, the spatial position of a weight within the kernel is encoded as a distinct edge feature to preserve spatial inductive biases. Similarly, Transformers can be decomposed into functional subgraphs where the Query, Key, Value, and Output projection matrices are treated as bipartite subgraphs connecting embedding channels, with head indices encoded as node features . Because our computational cost analysis in Appendix I demonstrates that the method is effectively linear in the number of edges, processing the denser parameter graphs associated with deep CNNs or Transformers remains computationally tractable.
>
> 3. Could you provide ablations showing the contribution of each loss term (Eq.~32)?
>
> Response. We thank the reviewer for requesting a breakdown of the loss components.
> To address this, we have added Appendix~J, which contains a comprehensive ablation study
> on the INR--trace classification task (MNIST / Fashion--MNIST). We evaluate the model
> under four configurations in order to isolate the contributions of the ranking term, the
> temporal smoothness term, and the signed regression head.
>
> 1. Impact of the ranking loss.
> Removing $\mathcal{L}_{\text{rank}}$ reduces Kendall's $\tau$ from $0.946$ to $0.811$ on
> MNIST. Geometrically, the slope of the distance--magnitude relation flattens from $-0.617$
> to $-0.473$. This shows that while the signed head induces a general trend, the ranking loss
> is essential for enforcing a crisp monotonic ordering in which larger weights are mapped to
> smaller hyperbolic radii. Without it, local inversions become more frequent, reducing
> predictive fidelity.
>
> 2. Removing both ranking and temporal smoothness further
> degrades performance, with $\tau$ dropping to $0.765$. The geometric slope becomes
> significantly shallower ($-0.365$), and the scatter of the distance--weight plots increases
> substantially (Figure~12). This indicates that $\mathcal{L}_{\text{smooth}}$ is not merely
> a stabiliser but a structural constraint: without it, node embeddings may jump
> between checkpoints, breaking the coherent temporal trajectory needed to track the
> evolution of weight importance.
>
> 3. Impact of the signed regression head.
> This is the most critical ablation. Removing the signed head (while keeping the link-prediction
> loss $\mathcal{L}_{\text{FD}}$) causes the largest geometric degradation. The distance--magnitude
> slope collapses to near zero ($-0.211$ on MNIST) and even reverses on Fashion--MNIST
> ($+0.125$). This shows that the geometry does not automatically capture weight magnitude
> from connectivity alone. The signed head is the primary mechanism that binds magnitude
> to hyperbolic radii; without it, distance reflects only edge existence rather than functional
> strength, making the embeddings unsuitable for interpretability.
>
> Table 7,8 and Figures 6--13 in the revised manuscript confirm that the full objective (Eq.~30)
> is necessary: the signed head couples magnitude to geometry, the ranking term sharpens
> ordering for accuracy, and the smoothness term ensures that the temporal trajectory forms
> a coherent evolutionary path.

---

> > ### Comment · Reviewer_fCBW · 2025-11-26
> >
> > Thank you for the thorough responses and the significant revisions made during the rebuttal period. The addition of the computational cost analysis (Appendix I) and the comprehensive ablation study (Appendix J) directly address my main concerns. I also appreciate the broader improvements to readability and the theoretical roadmap for extending to CNNs and Transformers. While some limitations remain (no empirical validation on modern architectures, unresolved variance on the Sinusoid task), I view these as reasonable future work directions for a paper focused on establishing a novel theoretical framework. I am increasing my score from 6 to 8.

---

### Official Review · Reviewer_oHUv · 2025-10-31

**Soundness:** 3
**Presentation:** 3
**Contribution:** 3
**Rating:** 4
**Confidence:** 2

**Summary:**

This submission builds on prior work that seeks to develop methods for identifying the behavior of a random deep neural network (DNN), as well possibly train the DNN. The authors use temporal information (i.e., the optimization history) to learn models with greater accuracy. In addition, the authors embed the snapshots of the DNNs (across training) in a hyperbolic geometry. This can aid in performance and interpretability.

**Strengths:**

1. The work was well motivated. The utility of metanetworks was clearly presented and the idea that optimization trajectories could be leveraged for further improvement was good.

2. The authors test on several different experimental set-ups, finding strong performance on in all cases. While they are not always best, their method is not so far off.

3. The visualization of the hyperbolic embedding (Figure 2) was very interesting and points to possibly new ways to understand DNN training.

**Weaknesses:**

1. I think the biggest weakness of this work is that it is so dense. I found the last paragraph of the Related Works Section and Sec. 3.2-3.4 just full of acronyms, method names, and details. I'm sure that this is partially due to the fact that I am not so familiar with this area, but it felt very difficult to follow.

2. The result of having so much detail crammed into the Methods section was that then there was little room for discussion on the experiments. It's not necessary to have all the model and experimental details, but it wasn't always clear to me what was even really being tested in the 3 experiments in Sec. 4. What exactly was tested in "Classification of Images via INR traces"? Which image is being shown to a MLP? if so, this seems to not be so aligned with the motivation in the Introduction. Similarly, I was confused as to what a "sinusoid–MLP" is and how the developed method was being used.

3. A minor point, but one way to study the optimization trajectory is to use tools from dynamical systems. These tools can be invariant to node ordering and can extract interpretable and comparable structure (e.g., https://proceedings.neurips.cc/paper_files/paper/2024/hash/2a07348a6a7b2c208ab5cb1ee0e78ab5-Abstract-Conference.html). How might the authors' work be extended/improved by including such dynamical characterization?

**Questions:**

1. What exactly was tested in "Classification of Images via INR traces"?

2. What is a "sinusoid–MLP" and what exactly was being tested in "Predicting Sine Wave Frequency"?

3. How might the authors' method be improved with - instead of passing in many optimization graphs - the dynamics of the optimization were first filtered with dynamical systems approaches?

---

> ### Author Response · Authors · 2025-11-20
> **Response to Reviewer oHUv:**
>
> Weaknesses:
> 1. I think the biggest weakness of this work is that it is so dense. I found the last paragraph of the Related Works Section and Sec. 3.2-3.4 just full of acronyms, method names, and details. I'm sure that this is partially due to the fact that I am not so familiar with this area, but it felt very difficult to follow.
>
> Response: We acknowledge the response of the reviewer regarding our work and have taken serious measures to address it. Taking into account the reviewers' feedback, we have reworked the last paragraph of the related works section by removing a lot of acronyms and focusing on the conceptual discourse of past contributions (Lines 137-161).
>
> Moreover, on the mathematical density of the work, we now provide an introductory paragraph before diving into the mathematical equations in Sections 3.2-3.4. Specifically, in Section 3.2, Lines 231-240 explain the overall goal and purpose of the module. For clarity and improving the readability, we break the previous mathematically dense works into multiple paragraphs: Hyperbolic Affine Maps (Q/K/V on the ball), Attention Logits and Structural Gating, Geometric Aggregation via Einstein Gyromidpoints, Temporal Evolution of Attention Kernels. Moreover, Lines 241-243 now discuss the nature of the attention maps and fundamental aspects, such as what roles the Q, K and V vectors play. Lines 251-254 discuss the roles of the log and exp operators, explaining their roles in these operations. Lines 259-263 now better explain the attention mechanism by fundamentally explaining the reasoning behind the formulation. Lines 276-280 now also address why we need a gyromidpoint aggregation, which might have been succinctly communicated in the previous draft. Similar to Section 3.2, we also add an introductory paragraph in Section 3.3, highlighting the overall purpose and motivation of the module. To reduce mathematical density and improve readability, we also restructured this section into two paragraphs: Magnitude as a Power Law of Hyperbolic Distance, Polarity in Tangent Space. We have restructured the text and have included clarifications for the mathematical equations that we felt lacked before (Lines 313-360). Additionally, in Section 3.4, we understood that the Section became too dense with all the formalisms of the loss terms in line with the words in the paragraphs, so we decided to explain the loss terms in linguistic terms along with their objective and basis for justification (Lines 363-387), moving all formalisms to Appendix G, improving the overall readability of the section. We hope this will ease the initial concerns regarding the readability of our work.

---

> > ### Author Response · Authors · 2025-11-20
> >
> > Continued from above..
> >
> > 2. :The result of having so much detail crammed into the Methods section was that then there was little room for discussion on the experiments. It's not necessary to have all the model and experimental details, but it wasn't always clear to me what was even really being tested in the 3 experiments in Sec. 4. What exactly was tested in "Classification of Images via INR traces"? Which image is being shown to a MLP? if so, this seems to not be so aligned with the motivation in the Introduction. Similarly, I was confused as to what a "sinusoid–MLP" is and how the developed method was being used.
> >
> > Response: We understand the reviewer's comment regarding the lack of clarity on the experiments and the terse nature of the discussions. Having leveraged an additional page in this phase of discussion, we have attempted to clarify the goal of each experiment (Lines 409-412, Lines 423-426, Lines 468-471). Moreover, we have significantly improved the discussions of the results and specifically their implications in Section 5 (Lines 478-516 (Interpreting the results)).
> >
> > For each experiment, we clarify the goals:
> >
> >
> > Classification of Images via INR Traces (Lines 409-412). This experiment evaluates whether the temporal evolution of the parameter graph of the intrinsic neural representation (INR) of an image contains enough structure for a meta-network to infer the class of the underlying image. Crucially, the meta-network never sees the image itself; it only receives the evolution of the INR weights over optimisation.
> > Predicting DNN Generalization from Weights (Lines 423-426). This experiment tests whether the temporal evolution of the parameters of a model contains a predictive signal about its test accuracy. More specifically, can we recover accuracy-related information solely from a deep neural network’s weight structure?
> > Predicting Sine Wave Frequency (Lines 468-472). In this experiment, each input network is an MLP trained to fit a one-dimensional sinusoidal function of the form x → a sin(bx), with coefficients a, b ∈ R varying across samples. This is what the implication of “sinusoid-MLP” was. We have clarified the notion in this draft of the manuscript. The goal was to learn a meta-network encoder that maps these trained MLPs to meaningful representations, while remaining invariant to neuron permutations.
> >
> >
> > We hope that these changes will placate the reviewers' concerns regarding the goal of
> > the experiments as well as their concerns regarding the discussions of the results.

---

> > > ### Author Response · Authors · 2025-11-20
> > >
> > > Continued from above..
> > >
> > > 3. A minor point, but one way to study the optimization trajectory is to use tools from dynamical systems. These tools can be invariant to node ordering and can extract interpretable and comparable structure (e.g., https://proceedings.neurips.cc/paper_files/paper/2024/hash/2a07348a6a7b2c208ab5cb1ee0e78ab5-Abstract-Conference.html). How might the authors' work be extended/improved by including such dynamical characterization?
> > >
> > > Response: We acknowledge the reviewer for providing a dynamical systems viewpoint for processing optimisation traces to better our work.  A natural way to relate our work to the paper cited above on topological conjugacy is to separate two levels of description: global dynamical invariants versus local structural geometry. The topological-conjugacy work models training as a discrete-time dynamical system on parameter space and then uses Koopman operators to extract global spectral invariants of this flow (eigenvalues, modes). These spectra are explicitly designed to be invariant under reparametrizations such as neuron permutations, and to collapse the whole trajectory into a compact signature of stability, timescales, and dynamical equivalence classes. In contrast, our temporal hyperbolic graph meta-network is built to preserve the full parameter-graph structure at each step and to learn how this structure reorganizes over training. We enforce permutation equivariance at the graph level, but we deliberately do not quotient out all differences that are dynamically “equivalent” in the sense of conjugacy: our downstream tasks (e.g., INR class, generalization properties, frequency) depend precisely on those architectural and structural details. In that sense, the two approaches are complementary: Koopman spectra provide a global summary of the optimization dynamics, while GTH-GMN focuses on the evolving internal graph geometry.
> > >
> > > We agree that combining these views is a promising direction (see Lines 535–539 in the revised paper). A natural extension, which is well beyond the scope of the present work, would be to incorporate coarse dynamical information computed directly from the optimization trajectory as gentle auxiliary regularizers for the temporal embeddings. The goal would not be to estimate or predict any specific dynamical quantities, but simply to allow the temporal representations to be informed by broader global signals about the behaviour of the optimizer–architecture pair. Such context could help the embeddings reflect both local structural organization and aspects of the overall training dynamics. We view this as an interesting avenue for future exploration, but our current paper focuses solely on showing that temporal hyperbolic parameter-graph embeddings are already predictive and interpretable on their own to quite an extent.

---

> ### Author Response · Authors · 2025-11-20
>
> continued from above..
>
> Questions:
>
> 1. What exactly was tested in "Classification of Images via INR traces"?
>
> Response: In this experiment, we tested whether the temporal evolution of the parameter graph of the intrinsic neural representation (INR) of an image contains enough structure for a meta-network to infer the class of the underlying image. Crucially, the meta-network never sees the image itself; it only receives the evolution of the INR weights over optimisation. This helps us evaluate the predictive power of the meta-network in recovering image classes from their parameter graph traces.
>
> 2. What is a "sinusoid–MLP" and what exactly was being tested in "Predicting Sine Wave Frequency"?
>
> Response: In this experiment, each input network is an MLP trained to fit a one-dimensional sinusoidal function of the form x → a sin(bx), with coefficients a, b ∈ R varying across samples. The goal, as stated above, was to map these MLPs to meaningful representations while remaining invariant to neuron permutations.
>
>
> 3. How might the authors' method be improved with - instead of passing in many optimization graphs - the dynamics of the optimization were first filtered with dynamical systems approaches?
>
> Response: We thank the reviewer for this suggestion. Pre-filtering the dynamics with dynamical systems tools such as Koopman or DMD would compress the trajectory into global invariants, but this would discard the node and edge level structure that GTH GMN is explicitly designed to exploit. We see these approaches as complementary rather than substitutes and view using such global dynamical quantities as additional inputs or auxiliary targets for our temporal encoder as an interesting direction for future work (see Lines 535-539 in the revised paper).

---

> > ### Comment · Reviewer_oHUv · 2025-11-20
> >
> > I thank the authors for their extensive rebuttal, for which they should be commended. I really appreciate that they took the reviews seriously and have put in quite a bit of work to revise and improve their submission.
> >
> > I have a few clarifying questions based on the rebuttal:
> >
> > 1. The additional details provided on the experiments was helpful. I think I now understand the MNIST/FashioNMNIST example. In this setting, the INRs were trained on individual images (basically, each INR was trained to represent one image) and the proposed approach enabled identification of the class, suggesting that the dynamics of the INR training were related to the class. Is that correct?
> >      I remain a little confused about the sinusoidal-MLP task. In this setting, I now understand that MLPs are trained to generate different a sin(bx) functions. But, what is meant by "The goal is to learn a meta-network encoder that maps these trained MLP temporal parameter graphs to meaningful representations, while remaining invariant to neuron permutations, where representation quality is assessed in a contrastive-learning framework inspired by SimCLR"? Is the meta-network evaluated on how well it can train an MLP for a new pair of a and b? And the performance being compared to the training trajectory?
> >
> > 2. I appreciate the authors' comments on the differences with the dynamical systems methods. The point about training geometry encoding information that is important (but quotiented out in the dynamical systems perspective) is interesting and is well taken. This may be too much to ask, but would it be possible to identify how much this information is providing? For instance, could it be that for one of the tasks (e.g., say the INR image classification), the training dynamics are different for each class, and that is really what the proposed framework is extracting? In which case, the extra machinery might not be necessary.
> >      I understand that knowing whether the geometry is or is not important is hard to identify a priori, and the fact that the authors method works well on a number of problems is evidence of its robustness, but it would helpful to have these "ablation" like results to assess when it is necessary.

---

> ### Author Response · Authors · 2025-11-21
> **Response to Reviewer oHUv**
>
> We thank the reviewer for their acknowledgement of our rebuttal and for a careful follow-up, allowing us the opportunity to clarify our work.
>
> 1a. The additional details provided on the experiments was helpful. I think I now understand the MNIST/FashioNMNIST example. In this setting, the INRs were trained on individual images (basically, each INR was trained to represent one image) and the proposed approach enabled identification of the class, suggesting that the dynamics of the INR training were related to the class. Is that correct?
>
> Response: Yes, that understanding is correct.
>
> 1b.   I remain a little confused about the sinusoidal-MLP task. In this setting, I now understand that MLPs are trained to generate different a sin(bx) functions. But, what is meant by "The goal is to learn a meta-network encoder that maps these trained MLP temporal parameter graphs to meaningful representations, while remaining invariant to neuron permutations, where representation quality is assessed in a contrastive-learning framework inspired by SimCLR"? Is the meta-network evaluated on how well it can train an MLP for a new pair of a and b? And the performance being compared to the training trajectory?
>
> Response: In the sinusoidal–MLP experiment, two distinct processes are occurring: first, the training of many small MLPs on simple regression tasks; and second, the training of a meta-network that learns to summarize the optimization dynamics of those MLPs.
>
> At the base level, for each experiment, we choose a pair of coefficients $(a, b)$. These define a target function $f_{a,b}(x) = a \sin(bx)$. From a training dataset of input–output pairs $(x_j, y_j)$, a standard MLP $g_\theta(x)$ is trained via gradient descent to minimize the mean squared error against $a \sin(bx_j)$. Conceptually, each trained MLP acts as an Implicit Neural Representation (INR) of that specific sine function: the continuous curve is encoded in the evolving weights $\theta$ of the network.
>
> Crucially, we do not look only at the final weights. At every optimization step $t$, we record the current parameter vector $\theta_t$ and construct a corresponding parameter graph where nodes represent neurons and edges represent signed weights. The full sequence of these graphs forms a temporal parameter graph for that specific MLP. Thus, for each sinusoid, we capture the trajectory of how the network’s topology reorganizes during training.
>
> The meta-network operates on top of this trajectory. It does not train the base MLPs, nor is it asked to produce weights for new ones. Instead, its role is to compress the entire training history into a single vector representation. We employ our hyperbolic graph encoder at each time step, coupled with a temporal module that aggregates across the sequence. This encoder is designed to be invariant to neuron permutations: if neurons within a hidden layer are permuted consistently along the trajectory, the resulting graph representation remains unchanged.
>
> Regarding the training of this encoder,  we primarily use the geometric objectives described in the paper (link prediction, signed weight regression, temporal smoothing, and ranking) to ensure the representation respects the underlying graph structure. Additionally, we incorporate a SimCLR-style contrastive regularizer applied to the trajectories. From a single training sequence, we generate two correlated "views" by applying Gaussian perturbations to node/edge features and random temporal masking (dropping subsets of checkpoints). We then encourage their representations to be close to each other while pushing apart representations from different trajectories. This contrastive term serves solely to regularize the embedding space; it is not used to generate new parameters.
>
> Finally, regarding evaluation, we do not ask the meta-network to initialize or train a new MLP. Instead, we treat the learned embedding  as a summary of the training process and attach a small regression head. This head is trained to predict the original generative coefficients $(a, b)$ directly from the trajectory representation. Performance is measured as the test Mean Squared Error (MSE) between the predicted and true $(a, b)$ on held-out sinusoidal functions. In essence, we are evaluating how much information about the underlying function parameters is recoverable purely from the geometric dynamics of the training process.
>
> Continued below..

---

> ### Author Response · Authors · 2025-11-21
>
> 2. I appreciate the authors' comments on the differences with the dynamical systems methods. The point about training geometry encoding information that is important (but quotiented out in the dynamical systems perspective) is interesting and is well taken. This may be too much to ask, but would it be possible to identify how much this information is providing? For instance, could it be that for one of the tasks (e.g., say the INR image classification), the training dynamics are different for each class, and that is really what the proposed framework is extracting? In which case, the extra machinery might not be necessary. I understand that knowing whether the geometry is or is not important is hard to identify a priori, and the fact that the authors method works well on a number of problems is evidence of its robustness, but it would helpful to have these "ablation" like results to assess when it is necessary.
>
>
> Response: We thank the reviewer for this important question, which touches on the core theoretical premise of our work. Asper our understanding of your question, you seem to ask whether the "extra machinery" of hyperbolic geometry is truly necessary, or if the model is simply picking up on raw training dynamics (e.g., trajectory speed or direction) that could be captured by a simpler dynamical system. While we cannot introduce entirely new dynamical-systems baselines at this stage, we believe the Ablation Study in Appendix J provides some quantitative and satiable evidence regarding the specific contribution of the geometric priors beyond raw dynamics.
>
> First, to test if the model relies solely on temporal evolution, we evaluated the "No Ranking Loss" condition. Here, the temporal kernel remains active, but the constraint enforcing strict hyperbolic ordering is removed. If coarse dynamics were sufficient, we believe that the performance should largely remain stable. However, we observed a clear degradation: Kendall’s $\tau$ dropped from 0.946 (Full Objective) MNIST to 0.811 (No Ranking) and 0.935 (full objective) to 0.797 (No Ranking) on FMNIST dataset. This gap suggests that the model does get augmented by a coordinate system that helps resolves how weights are ordered hierarchically in the Poincaré ball, not just that they are moving.
>
> Second, we investigated if hyperbolic distances can naturally encode magnitude from connectivity and dynamics alone. In the "No Signed Regression Head" ablation, we retained temporal modeling and link prediction but removed the explicit supervision tying distance to weight magnitude. The result was a degradation of Kendall’s $\tau$ from $0.946$ and $0.935$ to approx. $765$ on both MNIST and FMNIST datasets, along with a decline in the negative slopes from $-0.322$ to $-0.061$ on the FMNIST datasets.
> This demonstrates that the signed regression head is not a redundant component but a module that makes it an essential mechanism which forces the physical properties of the weights to align with the geometry.
>
> Together, these results indicate that the hyperbolic geometry contributes quantifiable information beyond raw trajectory behavior. While a comparison against pure dynamical invariants (e.g., Koopman spectra) is an exciting direction for future work, our current ablations to quite some extent confirm that the geometric machinery performs essential work in decoding the training signals.
>
> We again thank the reviewer for their inquiry and hope that our response satisfies their question and addresses their concerns.

---

> > ### Comment · Reviewer_oHUv · 2025-11-21
> >
> > I thank the authors for their continued discussion and clarifications. I now better understand the points above. The sine-MLP task is then very much like the MNIST/FashionMNIST task, only now in time as opposed to pixel space. The same conclusion about aspects of the features of the data being encoded in the training geometry is present and helps strengthen the overall claim. Adding a little more of this information to the paper (or adding it to an Appendix and referencing that appendix) would help clarify this.
> >
> > Thank you for pointing out the ablation studies in Appendix J. I apologize for having missed them. These results definitely start to get at what I was asking. It's interesting to see the degradation when using the no ranking loss. Maybe a slightly conservative conclusion from the results is that, the coarse dynamics are providing useful information (indeed, the Kendall correlation of > 0.75 is evidence of some information), but that using the extra machinery the authors developed provides a more nuanced and resolved ability to classify performance. I think emphasizing this point (and referencing the ablation studies) would help increase the impact of the paper.
> >
> > Again, I appreciate the authors willingness to engage constructively and I think the revisions made across the course of the rebuttal period significantly increase the quality of the work. While I still believe the paper is quite dense, I think it has become more readable and I imagine that others more in this field will be better able to access the framework than I. From a tools perspective, the results developed are interesting and I think point to new directions/ways in which to study DNN training. For these reasons, I have decided to increase my score to a 6.

---

> > > ### Author Response · Authors · 2025-11-22
> > >
> > > We are deeply grateful for your continued engagement and for raising your score. We sincerely appreciate your constructive feedback throughout this process, which has undeniably pushed us to improve the clarity, rigor, and readability of our work. We concur with your insights regarding the interpretation of our results and have incorporated your specific analogy comparing the Sinusoid task to image classification as operating in parameter-time space versus pixel space. We have explicitly augmented Appendix E.4 to include this framing, ensuring that future readers grasp the parallel immediately. Thank you again for your time and for helping us refine both the presentation and the scientific interpretation of our framework.

---

### Author Response · Authors · 2025-11-20
**A General Comment To All The Reviewers:**

First of all, we would like to thank all the reviewers for taking the time to read our paper and for providing their valuable and thoughtful feedback. Most importantly, we appreciate and feel encouraged by the strong recognition of our motivation as well as the overall consensus over the novelty of our work (oHUv, fCBW, dcw3). Additionally, we also appreciate that some of you (fCBW, oHUv) found the visualisations of our training dynamics interesting, sensing their implications towards intrinsic explainability. We also noted that the reviewers have not raised any major concerns regarding the technical validity of our approach in representing neural parameter temporal graphs within hyperbolic spaces, specifically the Poincaré ball, and have at times appreciated the mathematical machinery involved in the methodology.

However, we also hear your concerns clearly regarding the paper being too mathematically dense, in need of empirical ablations to justify the loss terms, and the requirement to address computational cost and scalability. Moreover, the most crucial weakness of our paper that was unanimously highlighted was the readability of the work. We have taken these concerns regarding the manuscript's density, lack of component validation, and scalability constraints, as well as readability, very seriously.

All revisions in the manuscript are highlighted in blue to assist the reviewing process. We note that while the changes may appear extensive at first glance, they are primarily presentational adjustments designed to mitigate mathematical density and improve readability, without altering the underlying semantics or the core essence of the work. Crucially, no new equations or formalisms have been introduced; instead, we have expanded the exposition of the existing formalism to facilitate comprehension.


Readability Issues. In response, we have undertaken a comprehensive revision of the paper. To address the consensus that the method section (Section 3) was "hard to read," we have:

1. Restructured Sections 3.2, 3.3, and 3.4 to prioritise intuitive explanations of the hyperbolic-attention graph and temporal mechanisms before introducing the mathematical formalisms.

2. Moreover, in Section 3.4, we now provide a less mathematically dense exposition of our optimisation terms, with the formalisms moved to Appendix G, along with a fundamental explanation of the necessity of using a two-phased optimisation schema.

3. Additionally, we have added more clarity in our motivation (Line 43- 48) to make the fundamental aspect of “why this problem needs to be focused” clear. Moreover, we address the concerns regarding the clarity of the last paragraph in the Introduction by underscoring its interdisciplinary nature and highlighting it as a non-trivial assimilation of concepts, ideas, as well as findings from different fields such as temporal learning, graph-based representations, weight space learning, hyperbolic geometry, network science and physics. We also highlight our belief that this work covers a research gap in representing deep neural network traces in hyperbolic spaces and could be used as a baseline for improving performance in future works, while still being a useful tool for studying the intrinsic evolution of geometry over time during network optimisation.

We have also reformatted the related works section on hyperbolic and temporal graph learning (Line 137-161) by making it free of acronyms of past methods and relying on explaining the concepts rather than unnecessarily focusing on the terms and abbreviations of the approaches.

Continued Below..

---

> ### Author Response · Authors · 2025-11-20
>
> Experimental Targeting Issues. Another important concern that was highlighted was that the aim, goal, or purpose of our experiments was not always clear. We have now directly addressed these concerns by explicitly stating the goal of the experiments.  We would also like to state them here as well, for the sake of clarity: We conduct 3 experiments within the paper,
>
> 1. Classifications of Images via INR Traces (Lines 409-412). Whether the temporal evolution of the parameter graph of the intrinsic neural representation (INR) of an image contains enough structure for a meta-network to infer the class of the underlying image.
> 2. Predicting DNN Generalisation from Weights (Lines 423-426). Whether the temporal evolution of the parameters of a model contains a predictive signal about its test accuracy.
> 3. Predicting the Sine Wave Frequency(Lines 468-472). To learn a meta-network encoder that maps trained MLP temporal parameter graphs to meaningful representations, while remaining invariant to neuron permutations.
>
> These experiments outline different tasks for testing the predictive performance of neural meta-networks. We add a temporal version by accounting for traces of parameter graphs to test the predictive performance of our model across different types of meta-learning tasks, to highlight that the model is capable of exhibiting strong predictive performance in relation to its Euclidean and non-temporal counterparts and offers faithful geometric representations that remain inherently explainable, even though the predictive performance may not always be state-of-the-art. We express this as a trade-off between predictive performance and geometric explainability, and now also highlight mitigation strategies for noisy performance across these experiments (Lines 507-516).
>
> Presentation Issues. On top of that, to reinforce the presentation and clarity of exposition of our approach and results, we now also discuss the implications of the results in detail, given the additional page limit (Now 10 Pages, previously 9) in the rebuttal phase, by introducing a comprehensive discussion section. In the discussion section (Line 477-539), we interpret the results, their alignment with our motivation. Also, we discuss the implications for explainability, what quantifiable measures (drift rate, radial separation measures, hub-centralities, etc.) could be extracted from this approach and how they can aid both interpretability and explainability. Additionally, we acknowledge the limitation that this work empirically studies only MLPs; we also discuss the theoretical roadmap towards extending the approach to other complex architectures like CNNs and Transformers in the discussion now (Lines 526-534). Fundamentally, the hyperbolic and temporal modules of the framework remain unaffected, with only changes required in converting these architectures into coherent parameter graph structures.
>
> 1. CNNs can be modelled as multi-graphs where edges connect input/output channels, with kernel spatial positions encoded as edge features.
> 2. Transformers induce subgraphs connecting query, key, and value channels.
>
> Since our hyperbolic encoder operates on the graph topology rather than specific tensor shapes, the core method is architecture-agnostic. We classify their empirical evaluation as a subsequent future work.
>
>
> Continued Below..

---

> ### Author Response · Authors · 2025-11-20
>
> Ablation Targeting Issues. Apart from that, almost all the reviewers requested us for an ablation study concerning the loss term, which is composed of 6 elements (Fermi-Dirac loss, ranking loss, temporal smoothness, scale regularizer, weight magnitude, and sign/polarity predictors) in our optimization setup. We understand their need regarding the component analysis and what impact each term could have. However, we would also like to point out that removing all the terms “naively” would collapse the purpose of the experiments, where the aim is to study the evolving geometry in negatively curved spaces. So, we always keep the Fermi-Dirac term that enables the geometric meta-network to learn the hyperbolic nature of edge connectivity, while lacking the binding for magnitude strengths, which emerges from the signed magnitude loss terms. We perform ablations on the Classification of Images from INR traces task on MNIST and FMNIST datasets in the following scenarios:
>
>
> 1. A full objective (all 6 loss terms):  This serves as the reference baseline to establish the upper bound of performance and geometric coherence. It tests the hypothesis that the synergy of all inductive biases, hierarchical connectivity (Fermi-Dirac), magnitude-distance coupling (Signed-Weight), and temporal continuity (Smoothness) produces the most stable and predictive representation of the training dynamics.
>
> 2. Without ranking loss terms: This isolates the importance of monotonicity. We aim to test whether strictly enforcing the ordering "larger weights ≅ smaller hyperbolic distances" is necessary for the downstream classification accuracy, or if it is primarily a tool for visual interpretability. This helps quantify how much the "crispness" of the geometric organisation contributes to the predictive signal.
>
> 3. Without a temporal regularizer and a ranking loss: This tests the temporal stability of the parameter graph embedding. By removing the smoothness constraint, we evaluate whether the model naturally learns a coherent trajectory or if the embeddings "jump" chaotically between checkpoints. This determines if the temporal kernel evolution alone is sufficient to maintain a smooth geometric path without explicit penalisation.
>
> 4. Without the Signed-Weight heads:  This is the critical test of the binding mechanism. We retain the Fermi-Dirac loss (to learn connectivity) but remove the explicit supervision on weight values. This asks the fundamental question: Can the geometry naturally capture the heavy-tailed distribution of weights solely through link prediction, or is explicit regression required to "bind" functional magnitude to hyperbolic distance?
>
> | Variant | L_FD | L_wrec | L_sign | L_α | L_rank | L_smooth |
> |---------|------|--------|--------|-----|--------|----------|
> | Full objective | ✓ | ✓ | ✓ | ✓ | ✓ | ✓ |
> | No ranking | ✓ | ✓ | ✓ | ✓ | ✗ | ✓ |
> | No ranking + no smoothness | ✓ | ✓ | ✓ | ✓ | ✗ | ✗ |
> | No signed head | ✓ | ✗ | ✗ | ✗ | ✓ | ✓ |
>
> **Ablation Summary**
>
> | Condition          | Metric     | MNIST  | Fashion-MNIST |
> |--------------------|------------|--------|----------------|
> | Full Objective     | τ          | 0.946  | 0.935          |
> |                    | Slope (b)  | -0.617 | -0.322         |
> | No Ranking Loss    | τ          | 0.811  | 0.797          |
> |                    | Slope (b)  | -0.473 | -0.189         |
> | No Rank/Smooth     | τ          | 0.765  | 0.764          |
> |                    | Slope (b)  | -0.365 | -0.061         |
> | No Signed Head     | τ          | 0.738  | 0.726          |
> |                    | Slope (b)  | -0.211 | +0.125         |
>
>
> Continued Below..

---

> > ### Author Response · Authors · 2025-11-20
> >
> > Continued from above..
> >
> > Finally, we now also provide a detailed Computational Cost Analysis in Appendix I to address the required resource requirements requested by the reviewers. We understand the following behaviour from our analysis:
> >
> > 1. Scaling with Architecture Complexity (Time): We demonstrate that the computational cost is dominated by the edge processing term $O(LHEd)$. This is critical when extending the framework to CNNs and Transformers, which are represented as denser parameter graphs with a significantly higher number of edges $E$. Because our method is effectively linear in the number of edges $E$, the computational cost scales predictably even for these heavier architectures, maintaining parity with standard Euclidean GNNs.
> > 2. Memory Efficiency (Space): By utilising recurrent kernel meta-evolution, we decouple memory usage from the trace length. The space complexity is $O(Nd + EH + Kh)$, where $K$ is the window size. This ensures that training remains memory-efficient even for long optimization traces ($T=100+$) of large models, as we avoid the prohibitive cost of storing the full history of embeddings.
> >
> >
> > In conclusion, we argue that these substantial revisions covering improved readability, rigorous component validation, and explicit cost evaluations have transformed the manuscript into a more accessible and empirically robust contribution. By clarifying the theoretical intuition and validating the geometric constraints, we hope to have effectively demonstrated the unique value of our framework in offering a principled and interpretable window into the training dynamics of neural networks that standard Euclidean methods cannot provide. We remain committed to engaging with any further questions during the discussion period, and thank you once again for guiding us toward a stronger submission.

---

### Meta-Review · Area_Chair_Wy6W · 2026-01-05

**Summary:**

This work aims to provide an interpretability-based training and analysis of deep neural networks based on ideas from hyperbolic geometry inference. The paper is mathematically well founded and the conceptual framework was identified by the reviewers as novel and interesting. There were a number of concerns and the authors spent significant efforts working to address these concerns. Of these, the primary concerns that I found most important were the issues of presentation clarity, and the interplay between interpretability and accuracy.

The issues of clarity were taken very seriously by the authors, with very significant edits to the text. I do think, reading through, that the new text can be followed more easily, however, there are still moments of density, such as the "Temporal learning in euclidean and hyperbolic spaces" section.

For interpretability vs. accuracy, the authors correctly indicate that building in interpretability often can smooth the function in a way that can hurt accuracy. A flip side is that appropriate applications of priors (in this case the projections into the hyperbolic space) might also help generalization. Regardless there remains a pretty significant gap. Additionally, the authors mention that the curvature parameter might be adjusted to lower the variance of the accuracy scores. This is an interesting assertion and felt like a missing piece given that the theory and model derivation/design was much stronger than the results.

Regardless, the paper itself presents an interesting and novel approach that centers on an important topic in modern day machine learning.

**Reviewer Concerns:**

The main concerns of the reviewers were:
 1) Clarity: Reviewers noted that the dense mathematical notation
 2) Clarity of the experiments: Similarly reviewers noted a lack of clarity as to the specific goals of each experiment.
 3) Clarity on how interpretability is achieved.
 4) Reviewers noted a lack of computational complexity analysis
 5) Experimental results: the reviewers noted minimal improvement (or lack thereof) over competing methods, along with high variance for some results
 6) Limitations on which architectures the methods can be extended to and how.

To address these, the authors heavily edited the text and provided new content including:
 1) Added ablation studies to study and validate each term in the loss function
 2) Edited text to be clearer on the paper's motivations and experimental goals. This includes extensive new text across the paper
 3) New computational cost analysis
 4) Explanations for the lower accuracy and high variance in the results

 The reviewers did a thorough job responding the concerns. I think that there is more to discuss on the interpretability vs. accuracy angle, as I found that response halfway convincing, and I do think that the authors should try to do another pass for clarity on the first few sections. As an aside, there is related work that intersects with the embedding nature of the work that the authors missed in the citations:
  - Gigante et al. Visualizing the phate of neural networks. NeurIPS, 32. 2019

**Reviewer Scores:**

The original scores for this submission were 6,4,4,4. Of these, before the cataclysm, Reviewers oHUv and fCBW raised their scores from 4-->6 and 6-->8, respectively. The remaining reviewers did not respond in time. Given the success in raising scores and the thoroughness of the responses, I would be surprised if at least one of them did not raise to a 6 as well, and I would estiamte that the final scores would have been 8,6,6,4 with a small possibility of 8,6,4,4, or 8,6,6,6.

---

### Decision · Program_Chairs · 2026-01-26

Accept (Poster)